# Identification of shared and disease-specific host gene–microbiome associations across human diseases using multi-omic integration

Sambhawa Priya[1,2], Michael B. Burns[3], Tonya Ward[4], Ruben A. T. Mars[5], Beth Adamowicz [1], Eric F. Lock[6], Purna C. Kashyap [5], Dan Knights[4,7] and Ran Blekhman [1,8] ✉

While gut microbiome and host gene regulation independently contribute to gastrointestinal disorders, it is unclear how the two may interact to influence host pathophysiology. Here we developed a machine learning-based framework to jointly analyse paired host transcriptomic ($n = 208$) and gut microbiome ($n = 208$) profiles from colonic mucosal samples of patients with colorectal cancer, inflammatory bowel disease and irritable bowel syndrome. We identified associations between gut microbes and host genes that depict shared as well as disease-specific patterns. We found that a common set of host genes and pathways implicated in gastrointestinal inflammation, gut barrier protection and energy metabolism are associated with disease-specific gut microbes. Additionally, we also found that mucosal gut microbes that have been implicated in all three diseases, such as *Streptococcus*, are associated with different host pathways in each disease, suggesting that similar microbes can affect host pathophysiology in a disease-specific manner through regulation of different host genes. Our framework can be applied to other diseases for the identification of host gene–microbiome associations that may influence disease outcomes.

The human gut microbiome plays a critical role in modulating human health and disease. Variations in the composition of the human gut microbiome have been associated with a wide variety of chronic diseases, including colorectal cancer (CRC), inflammatory bowel disease (IBD) and irritable bowel syndrome (IBS). For example, previous studies have reported an increase in the abundance of *Fusobacterium nucleatum* and *Parvimonas* in CRC[1,2], reduced abundance of *Faecalibacterium prausnitzii* and enrichment of enterotoxigenic *Bacteroides fragilis* in CRC and IBD[3–5], and over-representation of Enterobacteriaceae and *Streptococcus* in IBD and IBS[6–8]. In addition to the gut microbiome, dysregulation of host gene expression and pathways have also been implicated in these diseases. Researchers have reported disruption of Notch and WNT signalling pathways in CRC[9,10], activation of toll-like receptors (for example, TLR4) that induce NF-κB and TNF-α signalling pathways in IBD[11,12], and dysregulation of immune response and intestinal antibacterial gene expression in IBS[8,13]. While host transcription and gut microbiome have separately been identified as contributing factors to these gastrointestinal (GI) diseases, it is unclear how the two may associate to influence host pathophysiology[14].

Studies in model organisms have demonstrated that the modulation of host gene expression by the gut microbiome is a potential mechanism by which microbes can affect host physiology[15–18]. For example, in zebrafish, the gut microbiome negatively regulates the transcription factor hepatocyte nuclear factor 4, leading to host gene expression profiles associated with human IBD[18]. In mice, the gut microbiota can alter host epigenetic programming to modulate intestinal gene expression involved in immune and metabolic

processes[16,17]. Additionally, recent in vitro cell culture experiments have shown that specific gut microbes can modify the gene expression in interacting human colonic epithelial cells[19]. Given the evidence for crosstalk between the gut microbiome and host gene regulation, characterizing the interplay between the two factors is critical for unravelling their role in the pathogenesis of human intestinal diseases.

A few recent studies have investigated associations between the host transcriptome and gut microbiome in specific human gut disorders, including IBD, CRC and IBS. For example, studies examining microbiome–host gene relationships in IBD have identified mucosal microbiome associations with host transcripts enriched for immunoinflammatory pathways[20–22]. While investigating longitudinal host–microbiome dynamics in IBD, Lloyd-Price et al.[22] identified associations between expression of chemokine genes, including *CXCL6* and *DUOX2*, and abundance of gut microbes, including *Streptococcus* and Ruminococcaceae. Studies investigating the role of host gene–microbiome associations in CRC have found correlations between the abundance of pathogenic mucosal bacteria and expression of host genes implicated in gastrointestinal inflammation and tumorigenesis[23,24]. In IBS, host genes implicated in gut barrier function and peptidoglycan binding, such as *KIFC3* and *PGLYRP1*, are associated with microbial abundance of Peptostreptococcaceae and *Intestinibacter*[8]. While these studies have revealed important insights about host gene–microbiome crosstalk in GI diseases, they are limited in several aspects. For example, most studies have examined associations between a limited subset of host genes and gut microbes; for instance, by focusing only on differentially expressed genes[21,22,24],

[1]Department of Genetics, Cell Biology and Development, University of Minnesota, Minneapolis, MN, USA. [2]Bioinformatics and Computational Biology, University of Minnesota, Minneapolis, MN, USA. [3]Department of Biology, Loyola University Chicago, Chicago, IL, USA. [4]BioTechnology Institute, College of Biological Sciences, University of Minnesota, Minneapolis, MN, USA. [5]Division of Gastroenterology and Hepatology, Department of Internal Medicine, Mayo Clinic, Rochester, MN, USA. [6]Division of Biostatistics, School of Public Health, University of Minnesota, Minneapolis, MN, USA. [7]Department of Computer Science and Engineering, University of Minnesota, Minneapolis, MN, USA. [8]Department of Ecology, Evolution, and Behavior, University of Minnesota, Minneapolis, MN, USA. ✉e-mail: blekhman@umn.edu

genes associated with immune functions[13,23] or select microbes representing bacterial clusters or co-abundance groups[20,23], thus characterizing only a subset of potential associations. In addition, the identification of host gene–microbe associations is based on testing for pairwise correlation between every host gene and microbe using Spearman or Pearson correlation, thus ignoring the inherent multivariate properties of these datasets[21,22,24]. Additionally, most studies focus on examining associations in a single disease at a time; hence, common and unique patterns of host–microbiome associations across multiple disease states remain poorly characterized.

Here we comprehensively characterized associations between mucosal gene expression and microbiome composition in patients with colorectal cancer, inflammatory bowel disease and irritable bowel syndrome—three GI disorders in which both host gene regulation and gut microbiome have been implicated as contributing factors[1,6,8,10,13]. We developed and applied a machine learning framework that overcomes typical challenges in multi-omic integrations, including high-dimensionality, sparsity and multicollinearity, to identify biologically meaningful associations between gut microbes and host genes and pathways in each disease. We leveraged our framework to characterize disease-specific and shared host gene–microbiome associations across the three diseases that may facilitate insights into the molecular mechanisms underlying pathophysiology of these gastrointestinal diseases.

## Results

**Integrating host gene expression and gut microbiome abundance.** To study host–microbiome relationship across diseases, we used host gene expression (RNA-seq) data and gut microbiome abundance (16S rRNA sequencing) data generated from colonic mucosal biopsies obtained from patients with CRC, IBD and IBS (Fig. 1a). For each individual in our study, we obtained a pair of samples—a microbiome sample and a host gene expression sample. In total, across the three disease cohorts, our study included 208 such pairs of microbiome and host gene expression samples (416 samples in total; Supplementary Table 1). All datasets, except the host gene expression (RNA-seq) data for CRC, have been previously published as individual studies[3,8,22,25]. Detailed information on disease cohorts, samples, sequencing, quality control and data processing is available in Methods.

Previous studies have identified host gene–microbiome associations in human gut disorders, including CRC, IBD and IBS[8,22,23]. Thus, one might expect intestinal gene expression patterns and microbiome composition to be broadly correlated in these diseases. To test for such an overall association between host gene expression and gut microbiome composition, we performed Procrustes analysis using paired data for each disease cohort. Our analysis showed significant correspondence between host gene expression variation and gut microbiome composition across subjects in CRC (Monte Carlo $P$ value = 0.0001). However, Procrustes agreement is not significant in IBD (Monte Carlo $P$ value = 0.1) and IBS (Monte Carlo $P$ value = 0.42) (Fig. 1b and Methods). These results were verified using a Mantel test (Methods). This lack of significant overall correspondence between host transcriptome and gut microbiome across diseases might suggest that, rather than an overall association between the two, it is probable that only a subset of gut microbes is associated with a subset of host genes at the colonic epithelium[15,17]. Hence, we need integration approaches to characterize such host gene–microbiome associations.

To this end, we developed a machine learning-based multi-omic integration framework using sparse canonical correlation analysis (sparse CCA)[26,27] and lasso penalized regression[28]. We applied this approach to data from CRC, IBD and IBS for a comprehensive characterization of potentially biologically meaningful associations between gut microbiota and host genes and pathways across the three diseases (Fig. 1, Methods and Extended Data Fig. 1).

**Shared host pathways are associated with disease-specific gut microbes.** We hypothesized that host genes and gut microbial taxa involved in common biological functions would act in a coordinated fashion, and hence would have correlated expression and abundance patterns. To investigate this, we used sparse CCA to characterize group-level association between host transcriptome and gut microbiome in each of the three diseases[26,27]. We fit the sparse CCA model for each dataset to identify subsets of significantly correlated host genes and gut microbes, known as components (Methods and Supplementary Tables 2–4). We then performed pathway enrichment analysis on the set of host genes in each significant component to determine host pathways that associate with gut microbes in a disease. We identified 'shared' pathways, namely host pathways for which gene expression correlates with gut microbes across disease cohorts, and 'disease-specific' pathways, namely host pathways for which gene expression correlates with gut microbes in only one of the three disease cohorts (Fig. 2a, Fisher's exact test, Benjamini-Hochberg FDR < 0.1; Supplementary Table 5). For simplicity, we focused on the top five most significant shared and disease-specific pathways (Fig. 2a). We found three pathways shared across CRC, IBD and IBS that are known to regulate gastrointestinal tract inflammation, and gut barrier protection and repair. For example, oxidative phosphorylation, which is the process of energy metabolism in the mitochondria, is known to be dysregulated in IBD and CRC, and contributes to tumorigenesis and drug resistance in CRC[29–32]. We also found overlapping host pathways between disease pairs (see CRC & IBD, CRC & IBS, and IBD & IBS in Fig. 2a), including immunoregulatory pathways and cell-surface receptors such as integrin pathway, cell and focal adhesion, and proteasome.

In addition, we identified 102 disease-specific host pathways that are associated with gut microbes, including 52 CRC-specific, 25 IBD-specific and 25 IBS-specific pathways in our study cohorts (Supplementary Table 5 and Fig. 2a). While IBD-specific host pathways include A6B1/A6B4 integrin pathway and integrin beta-1 pathway that regulate leucocyte recruitment in GI inflammation[33], IBS-specific pathways include immune response pathways, including B cell receptor signalling pathway, and ribosome pathway.

To better understand the host gene–microbe associations that underlie common associations, we focused on the RAC1 pathway, where host gene expression is associated with microbiome composition in CRC, IBD and IBS. The RAC1 pathway is known to regulate immune response and intestinal mucosal repair, and has previously been implicated in IBD and CRC[34,35] (Fig. 2b). As expected, we observed some overlapping host genes for this shared pathway across the three diseases. However, the microbial taxa they are correlated with are disease-specific. In CRC, the RAC1 pathway is associated with oral bacterial taxa such as *Streptococcus*, *Synergistales* and *GN02*, where *Streptococcus* species are known to be associated with colorectal carcinogenesis[36,37]. In IBD, the RAC1 host pathway is associated with microbial taxa previously implicated in IBD, including *Granulicatella*[38,39], and *Clostridium sensu stricto 1*, a microbe associated with chronic enteropathy similar to IBD[40]. In IBS, this pathway is associated with bacteria such as *Bacteroides massiliensis* that has been shown to be prevalent in colitis[41], and *Bifidobacterium* and *Odoribacter* that are known to be depleted in IBS[42,43].

To investigate disease-specific associations, we considered unique host pathways for which host gene expression correlates with gut microbes only in one of the three diseases (Fig. 2c). For example, the Syndecan-1 pathway, which we found to be associated with gut microbial taxa only in CRC, has been previously shown to regulate the tumorigenic activity of cancer cells[44,45]. Host gene expression in this pathway is associated with microbial taxa such as *Parvimonas* and *Bacteroides fragilis* that are known to promote intestinal carcinogenesis and are considered biomarkers of CRC[1,46,47]. The integrin-1 pathway, a disease-specific host pathway in IBD, is found to be associated with *Peptostreptococcaceae*,

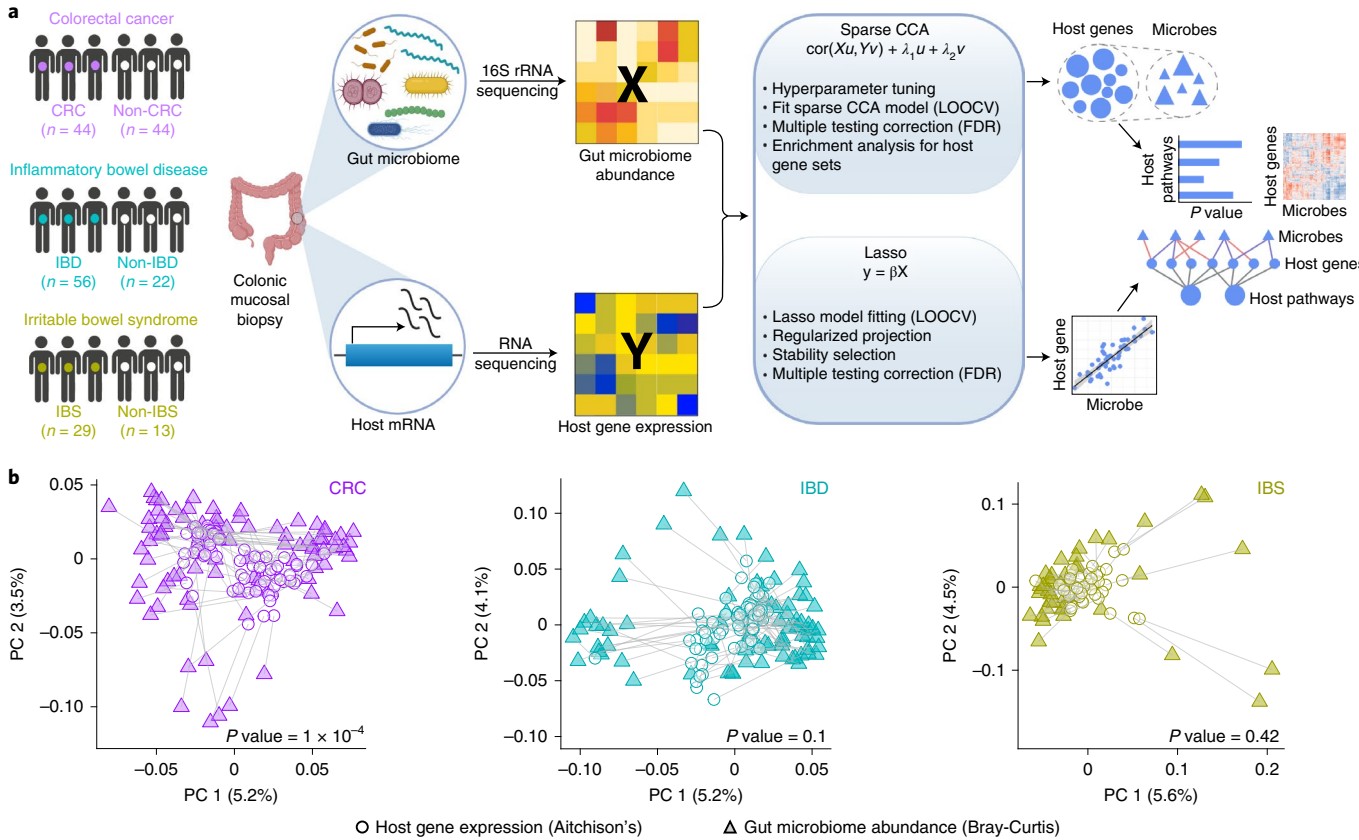

**Fig. 1 | Integrating host gene expression and gut microbiome abundance in CRC, IBD and IBS. a**, Overview of study design: colonic biopsy samples were collected from individuals in each disease cohort and for each sample, paired host transcriptomic (RNA-seq) data and gut microbiome abundance (16S rRNA) data were generated. The paired host transcriptomic and gut microbiome data were integrated using a machine learning-based framework to characterize associations between gut microbiota and host genes and pathways across the three diseases (left to right) (see Methods for details on the integration framework and mathematical notations). **b**, Procrustes analysis showing overall association between variation in host gene expression and gut microbiome composition in CRC, IBD and IBS (left to right). We used Aitchison's distance for host gene expression data (circles) and Bray-Curtis distance for gut microbiome data (triangles). Panel **a** was created using BioRender.com.

*Intestinibacter* and *Phascolarctobacterium*—microbial taxa that have been implicated in IBD by previous studies[48–50]. To assess similarities in host gene components across diseases, we identified a set of host genes that are common between components across the three diseases, and we found that these genes are enriched for immune response pathways in gut epithelium, including vascular endothelial growth factor (VEGF), complementation and coagulation cascades, and cytokine–cytokine receptor association (Fig. 2d, Fisher's exact test, Benjamini-Hochberg FDR < 0.1). While this set of host genes is associated with disease-specific groups of microbes, we also found overlapping microbes between IBD and IBS, such as Peptostreptococcaceae and *Intestinibacter*—taxa that are found in high abundance in gastrointestinal inflammation[48,50].

**Gut microbes are associated with individual host genes and pathways.** Previous studies have shown that specific microbial taxa can regulate expression of individual host genes[15,19]. Therefore, we explored associations between individual host genes and gut microbes in each disease. To do so, we used lasso penalized regression models to identify specific gut microbial taxa whose abundance is associated with the expression of a host gene[28]. We fit these models in a gene-wise manner, using the expression for each host gene as the response variable and the abundances of gut microbial taxa as predictors. We then applied stability selection to identify robust associations (Methods). Using this approach, we found 755, 1,295 and 441 significant and stability-selected host gene–taxa

associations in CRC, IBD and IBS, respectively (Fig. 3, FDR < 0.1). These represent associations between 745 host genes and 120 gut microbes in CRC (Supplementary Table 6), between 1,246 host genes and 56 gut microbes in IBD (Supplementary Table 7), and between 436 host genes and 102 gut microbes in IBS (Supplementary Table 8 and Fig. 3a). Examples of specific host gene–microbe associations can be found in Extended Data Fig. 2. Overall, we observed disease-specific patterns in host gene–taxa associations.

To characterize the biological functions represented by the host genes that associate with specific gut microbes, we applied enrichment analysis to the set of gut microbiota-associated host genes in each disease (Methods). This is complementary to our group-level approach (Fig. 2) in that these host pathways are enriched among individual host gene–microbe pairs. We identified 18 host pathways that are unique to each disease, including 4 CRC-specific, 9 IBD-specific and 5 IBS-specific pathways that associate with unique gut bacteria (Fig. 3b, Fisher's exact test, Benjamini-Hochberg FDR < 0.1; Supplementary Table 9 and Methods). The host pathways enriched among CRC-specific associations are known to modulate tumour growth, progression and metastasis in CRC, such as interleukin-10 signalling, signalling by *NOTCH1* in cancer, and regulation of MECP2 expression and activity[51–53]. The host pathways we identified as enriched among IBD-specific associations are known to be responsible for maintenance of gastric mucosa integrity, inflammatory response and host defence against invading pathogens, such as thrombin signalling through proteinase

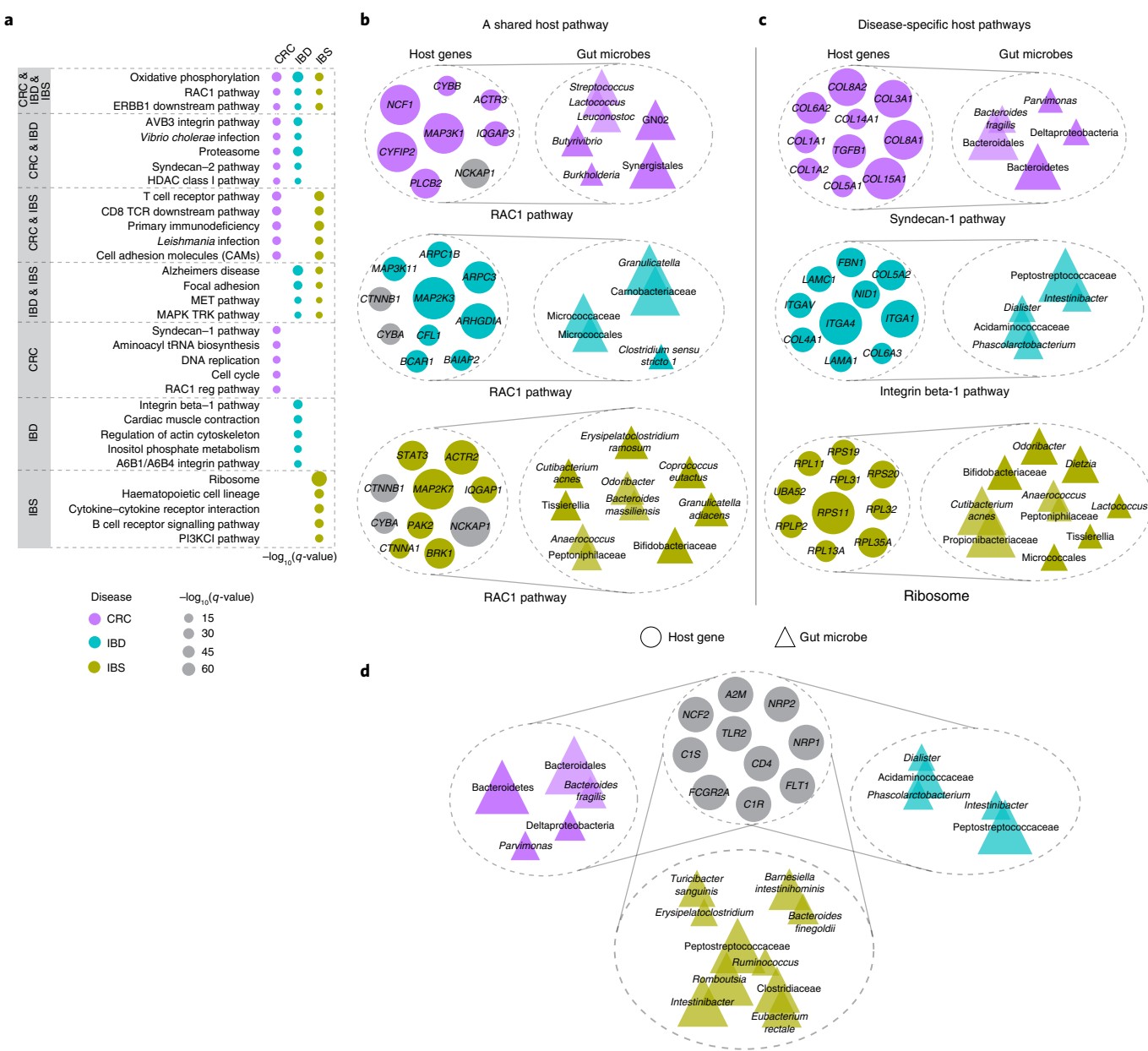

**Fig. 2 | Shared immunoregulatory and metabolic host pathways associate with disease-specific gut microbes across human diseases. a**, Host pathways enriched for sparse CCA gene sets associated with gut microbiome composition across diseases (FDR < 0.1). Dot size represents the significance of enrichment for each pathway, and dot colour denotes the disease cohort in which this pathway is significantly associated with microbiome composition. **b**, Association between microbial taxa in CRC, IBD and IBS (top to bottom) and host genes in the RAC1 pathway, a pathway for which host gene expression correlates with gut microbes across disease cohorts (a shared host pathway). The size of circles and triangles represents the absolute value of sparse CCA coefficients of genes and microbes, respectively. Microbial taxa belonging to a common taxonomic order are shown as overlapping triangles. Genes that are common between pathways or components across at least two disease cohorts are shown in grey. **c**, Association between the set of host genes in disease-specific host pathways (that is, host pathways for which gene expression correlates with gut microbes in only one of the disease cohorts) and groups of gut bacteria in CRC, IBD and IBS (top to bottom). **d**, A common set of host genes (grey circles) that are associated with disease-specific sets of microbes. These host genes are enriched for immunoregulatory and acute inflammatory response pathways.

activated receptors (PARs) and glucagon type ligand receptors[54,55]. For IBS-specific associations, the enriched host pathways identified here have been shown to regulate homoeostasis of intestinal tissue and proinflammatory mechanisms in IBS, such as sumoylation of DNA damage response and repair proteins, and arachidonic acid metabolism[56,57].

To characterize the potential mechanism of host gene–microbe associations, we further investigated the gut microbial taxa associated with host genes in these pathways (Fig. 3c–e). In CRC, we found that Anaerolineae and TM7—oral microbes that also inhabit the human gastrointestinal tract, and are known to promote oral and colorectal tumorigenesis[58-60]—are negatively correlated with host genes enriched in the tumour-promoting interleukin-10 signalling pathway, such *CXCL8* and *IL1RN* (Fig. 3c and Extended Data Fig. 2). *CXCL8* is known to be overexpressed in CRC, and *IL1RN* is centrally involved in immune and inflammatory response, and its polymorphisms are implicated in colorectal carcinogenesis[61,62]. Other host genes in interleukin-10 signalling, such as *CCR2*

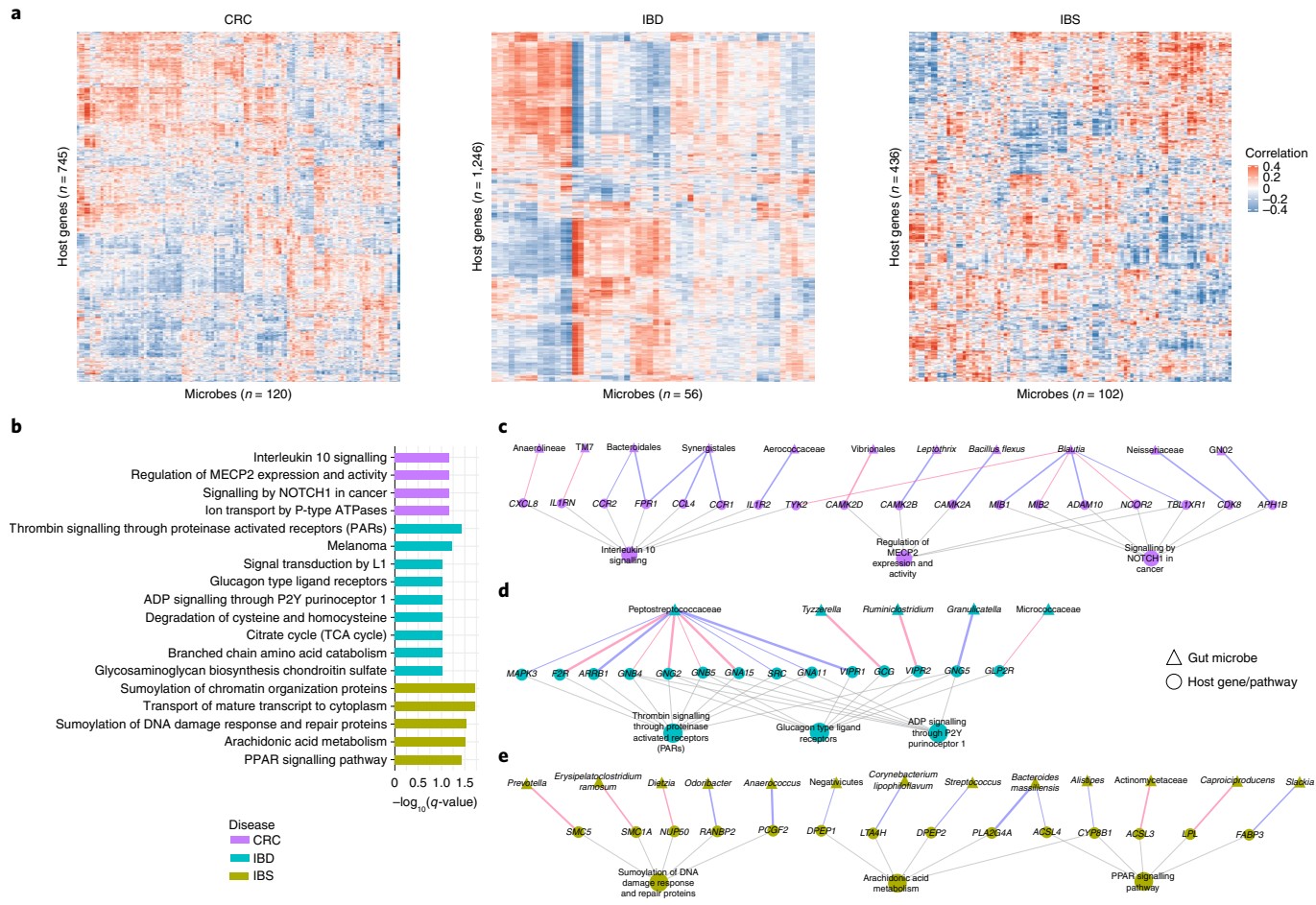

**Fig. 3 | Specific gut microbes are associated with individual host genes and pathways in each disease. a**, Heatmap showing the overall pattern of correlation between significant and stability-selected host genes (rows) and gut microbial taxa (columns) identified by the lasso model in CRC, IBD and IBS (FDR < 0.1). **b**, Host pathways enriched among genes that are correlated with specific gut microbes in CRC (purple), IBD (green) and IBS (yellow) (FDR < 0.1). **c-e**, Networks showing specific gut microbes correlated with specific host genes enriched for disease-specific host pathways in CRC (**c**), IBD (**d**) and IBS (**e**). Triangular nodes represent gut microbes, circular nodes represent host genes and pathways. Edge colour represents positive (blue) or negative (red) association, and edge width represents strength of association (Spearman rho). Grey edges represent host gene–pathway associations.

and *FPR1*, are positively correlated with Bacteroidales (Fig. 3c and Extended Data Fig. 2). *CCR2* and *FPR1* are overexpressed in colorectal tumours, while Bacteroidales are enriched in CRC and associated with tumorigenesis[63–65].

We observed that Peptostreptococcaceae, which is prevalent in patients with IBD[50], is associated with multiple host genes and pathways in IBD (Fig. 3d). For example, its abundance is positively correlated with the expression of host genes *MAPK3* and *VIPR1*, involved in thrombin signalling through proteinase activated receptors (PARs) and glucagon-type ligand receptors pathways, respectively. *MAPK3* is known to play a role in the progression and development of IBD, and *VIPR1* is overexpressed in inflamed mucosa[66,67]. In IBS-specific associations, we found the levels of *Prevotella*, which is known to be overrepresented in individuals with loose stool, to be negatively associated with expression of *SMC5*, which is involved in the sumoylation pathway[68,69] (Fig. 3e). We also found the expression of *PLA2G4A*, a host gene that plays an important role in arachidonic acid metabolism and modulates gut epithelial homoeostasis[70], to be positively correlated with the abundance of *Bacteroides massiliensis* in IBS, *B. massiliensis* being known to be prevalent in patients with gut malignancies[41] (Fig. 3e). Taken together, these findings demonstrate that associations between specific gut microbial taxa and specific host genes and pathways vary by disease state.

**Disease-specific gut microbe–host gene crosstalk.** To understand how gut microbes may associate with specific host genes across diseases, we explored the overlaps between host gene–microbe associations in CRC, IBD and IBS (Fig. 4a, lasso regression, Benjamini-Hochberg FDR < 0.1; Supplementary Table 10). We identified 'shared' gut microbes, namely gut microbes that associate with host genes in at least two diseases, and visualized their networks of association with host genes across diseases. We found three gut microbes, Peptostreptococcaceae, *Streptococcus* and *Staphylococcus*, whose abundance is correlated with host gene expression in all three diseases in our study cohorts (Fig. 4a, Network 1). Previous studies have revealed that Peptostreptococcaceae and *Streptococcus* spp. are found at elevated levels in CRC, IBD and IBS[8,37,50,71–74]. While traditionally considered nasal- or skin-associated bacteria, *Staphylococcus* spp. also colonize the human gastrointestinal tract and include opportunistic pathogens that can cause acute intestinal infections in patients with CRC and IBD, and are associated with increased risk of IBS and CRC[74–77]. We found that the abundance of Peptostreptococcaceae is positively correlated with the expression of host genes *PYGB* and *NCK2* in IBD, whereas it is negatively correlated with the expression of host gene *HAS2* in IBS. *PYGB* and *NCK2* are both upregulated in IBD, where *PYGB* is known to regulate the WNT/β-catenin pathway, and *NCK2* is involved in integrin and epidermal growth factor receptor signalling[78–80]. In contrast,

*HAS2* is known to have a protective effect on the colonic epithelium through regulation of intestinal homoeostasis and inflammation[81,82]. In CRC, we found that the abundance of Peptostreptococcaceae is negatively associated with the expression of *GAB1*, a host gene for which overexpression stimulates tumour growth in colon cancer cells[83]. *Streptococcus* also shows a disease-specific pattern of association with host gene expression in our study cohort. In CRC, its abundance is correlated with the expression of *RIPK4*, which regulates WNT signalling and the NF-κB pathway, and is upregulated in several cancer types, including colon cancer[84,85]. Similarly, in IBS, *Streptococcus* abundance is correlated with the expression of *DPEP2*, which is known to modulate macrophage inflammatory response[86] (Fig. 4a, Network 1).

Next, we visualized the networks of host gene–microbe associations for gut microbes that are associated with host genes in two diseases (Fig. 4a, Networks 2–4, lasso regression, Benjamini-Hochberg FDR < 0.1; Supplementary Table 10). We found 20 microbes for which abundance is associated with the expression of host genes in at least two diseases. Notably, the abundance of *Blautia*, a butyrate-producing beneficial microbe, is found to be negatively correlated with the expression of *RIPK3* in both CRC and IBD (Fig. 4a, Network 4; Extended Data Fig. 3). *RIPK3* promotes intestinal inflammation in IBD, and colon tumorigenesis[87,88]. Interestingly, in CRC, *Blautia* is also associated with *ZBP1* (Fig. 4a, Network 4), a host gene that recruits *RIPK3* to induce NF-κB activation, and regulates innate immune response to mediate host defence against tumours and pathogens[89,90].

Conversely, to explore how the same host genes may associate with different gut microbes across all diseases, we identified host genes that are associated with gut microbes in at least two diseases, and visualized their networks of association across diseases (Fig. 4b, lasso regression, FDR < 0.1; Supplementary Table 11). We identified 5 such host genes that associate with 4 gut microbes in CRC, 5 gut microbes in IBS and 4 gut microbes in IBD (Fig. 4b, Network 1; Supplementary Table 11). Of note, the expression of *PINK1*—a host gene that regulates mitochondrial homoeostasis and activates PI3-kinase/AKT signalling, contributing to intestinal inflammation in IBD and tumorigenesis[32,91]—is associated with the abundance of *Collinsella* in CRC, Peptostreptococcaceae in IBD and *Blautia* in IBS. Previous studies have found that *Collinsella* is increased in abundance in CRC[92], whereas *Blautia* has been found to be both positively and negatively correlated with IBS symptoms[8,43].

In addition, we identified 135 host genes for which expression is associated with abundance of microbial taxa in at least two of the three diseases, and visualized the network of the most significant associations (Fig. 4b, Networks 2–4, lasso regression, FDR < 0.1; Supplementary Table 11). We found that the host genes whose expression is correlated with gut microbes in both CRC and IBD are enriched for pathways involved in immune response, including natural killer cell mediated toxicity, *Leishmania* infection and leucocyte transendothelial migration (Fig. 4b, Network 4, Fisher's exact test, Benjamini-Hochberg FDR < 0.1). These host genes and the microbial taxa they associate with have been previously implicated in CRC and IBD. For example, expression of Annexin A1 or

*ANXA1*—a host gene known to regulate intestinal mucosal injury and repair, and found to be dysregulated in CRC and IBD[93,94]—is positively correlated with Bacteroidales in CRC, while negatively correlated with Peptostreptococcaceae in IBD (Fig. 4b, Network 4). *TLR4*—a host gene known to modulate inflammatory response in intestinal epithelium through recognition of bacterial lipopolysaccharide[95], and previously implicated in IBD and CRC[96,97]—is found to be associated with an oral microbe *GN02* in CRC[98], whereas in IBD it associates with Acidaminococcaceae—a gut microbe found to be increased in abundance in patients with Crohn's disease[99] (Fig. 4b, Network 4). Overall, our analysis shows that gut microbial taxa and host genes that are shared between associations across diseases depict disease-specific host–microbe crosstalk, thus suggesting that the mechanism of host gene–microbiome association might be specific to the disease.

## Discussion

While gut microbial communities and host gene expression have separately been implicated with human health and disease, the role of the association between gut microbes and host gene regulation in the pathogenesis of human gastrointestinal diseases remains largely unknown. Using a machine learning-based multi-omic integration framework, here we found both common and disease-specific interplay between gut microbes and host gene regulation that may contribute to the underlying pathophysiology of GI disorders, including CRC, IBD and IBS.

Previous studies have found common microbial signatures across CRC, IBD and IBS. For example, all three diseases exhibit an overrepresentation of Peptostreptococcaceae and *Streptococcus* spp.[8,37,50,72]. In addition, both CRC and IBD microbiomes are denoted by a loss of butyrate-producing gut bacteria, including *Blautia*, and an enrichment of enterotoxigenic *Bacteroides fragilis*[5,47,72]. In contrast to these microbiome similarities, host gene regulation shows distinct alterations across the three GI disorders; for example, unique antibacterial gene expression profile and disruption of the purine salvage pathway are specific to IBS, deregulation of proinflammatory IL-23/IL-17 signalling is unique to IBD, and prominent activation of oncogenic pathways such as Notch and WNT signalling is a hallmark of CRC[8,13,100,101]. Here we found that the same disease-related gut microbes can associate with different host genes and pathways in each disease. Thus, it is compelling to hypothesize that although diseases can be characterized by similar microbial perturbations, these microbes can impact disease-specific pathophysiological processes through association with different host genes in each disease. For example, we found that in CRC, *Streptococcus* is correlated with the expression of host genes that regulate WNT signalling and the NF-κB pathway, whereas in IBS *Streptococcus* is correlated with host genes that modulate macrophage inflammatory response, thus suggesting that this gut microbe may perturb distinct host pathways in CRC and IBS. Of course, since our results are based on correlational analysis, it is challenging to assess directionality. While it is possible that these disease-specific associations have a role in disease pathogenesis, it is also possible that the disease-transformed colonic mucosa renders it more conducive to the same microbial taxa.

**Fig. 4 | Disease-specific gut microbe–host gene crosstalk. a**, Associations for 'shared' gut microbes, namely microbes that are associated with host genes in at least two diseases. Centre: Venn diagram showing overlap between gut microbes associated with host genes in CRC, IBD and IBS. Counter-clockwise, left to right: networks showing host gene–microbe associations for gut microbes shared across associations in CRC, IBD and IBS (Network 1), CRC and IBS (Network 2), IBD and IBS (Network 3), and CRC and IBD (Network 4). **b**, Associations for 'shared' host genes, that is, genes that are associated with microbes in at least two diseases. Centre: Venn diagram showing overlap between host genes associated with gut microbes in CRC, IBD and IBS. Counter-clockwise, left to right: networks showing host gene–microbe associations for host genes shared across associations in CRC, IBD and IBS (Network 1), CRC and IBS (Network 2), IBD and IBS (Network 3), and CRC and IBD (Network 4). Circular nodes represent host genes, triangular nodes represent gut microbes. Coloured nodes represent specific diseases (purple, CRC; green, IBD; yellow, IBS), grey nodes represent gut microbes (**a**) and host genes (**b**) shared between associations across diseases. Edge colour represents positive (blue) or negative (red) association, and edge width represents strength of association (Spearman rho). All associations were determined at FDR < 0.1.

We also identified a common set of host genes and pathways that are associated with gut microbiome composition in all three diseases. These included pathways that regulate gastrointestinal inflammation, immune response and energy metabolism, and have been previously implicated in these diseases[30,102,103]. Our analysis shows that these common host genes and pathways correlate with

**a** Shared gut microbes

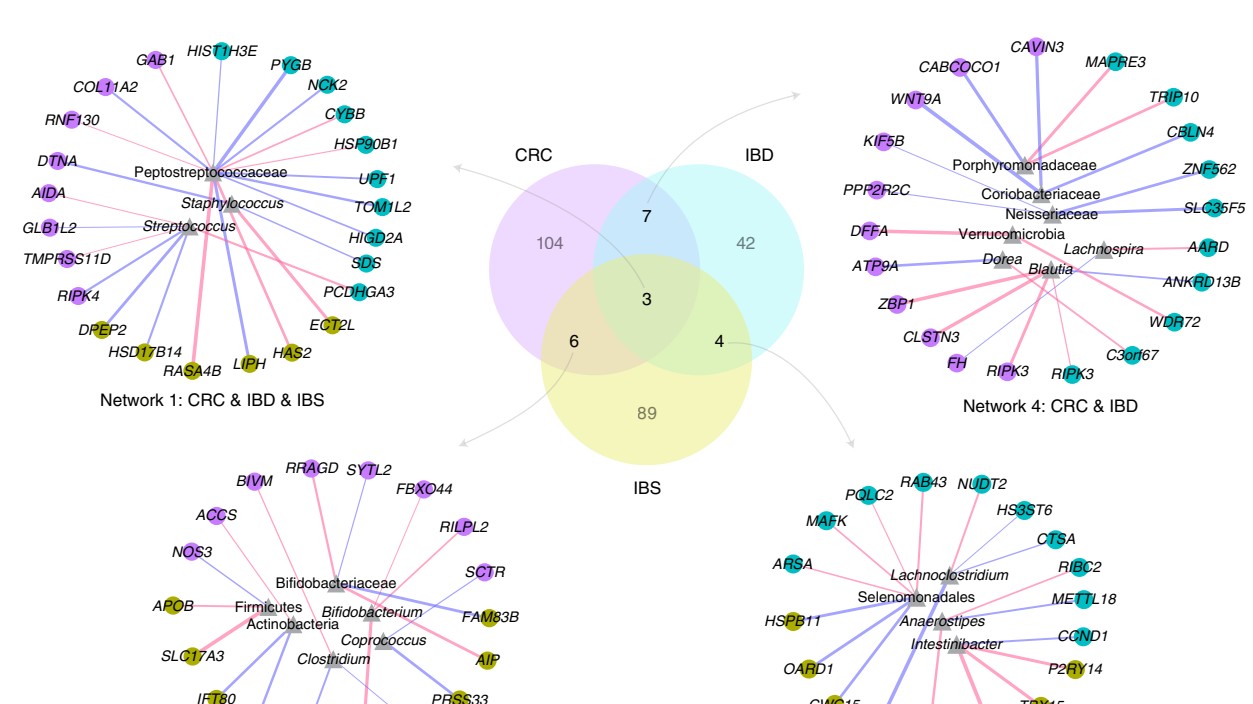

**b** Shared host genes

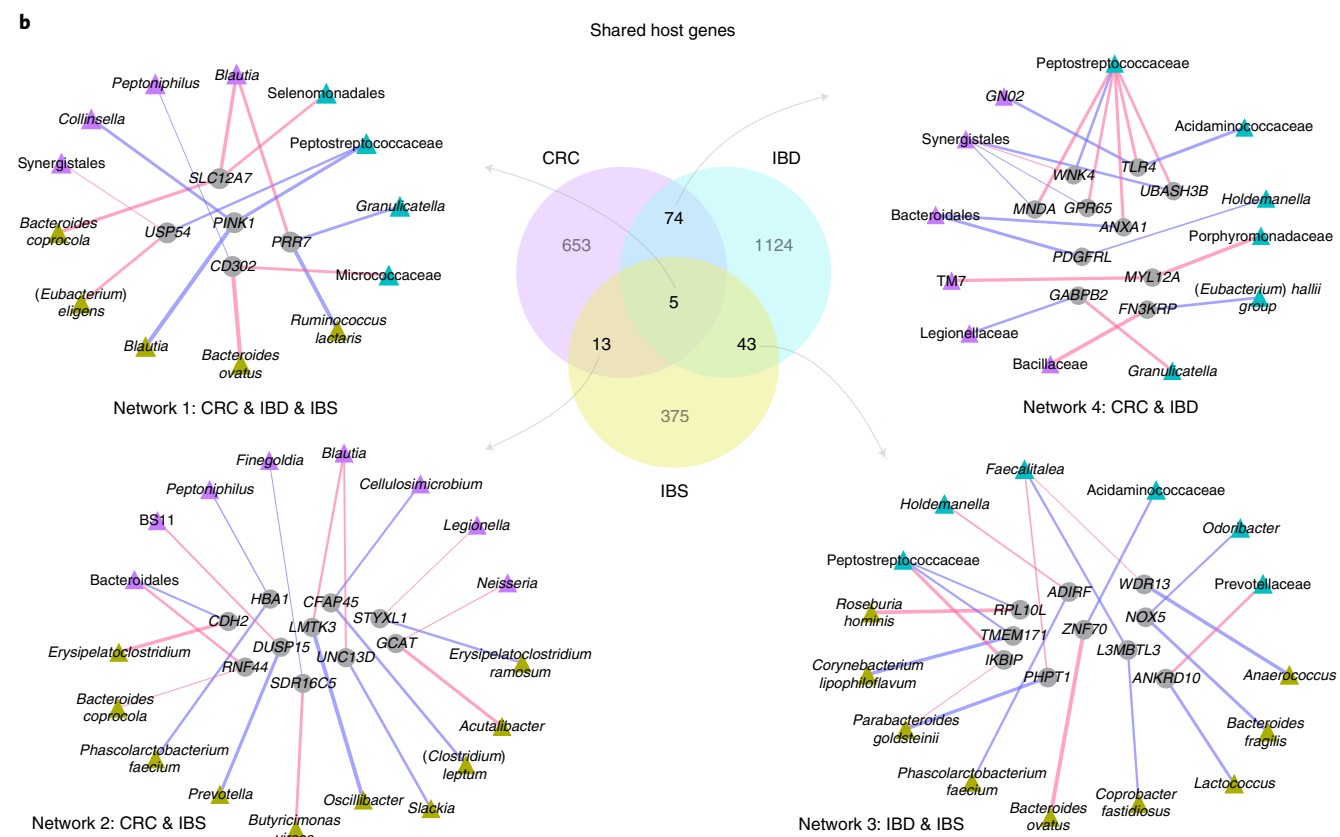

disease-specific gut microbes in CRC, IBD and IBS. For example, we found that the expression of host gene *PINK1*, which regulates the PI3-kinase/AKT signalling pathway[91], is associated with the abundance of *Collinsella* in CRC, Peptostreptococcaceae in IBD, and *Blautia* in IBS in our study. This suggests that in some cases, distinct gut microbes may modulate host genes and pathways that are commonly dysregulated across different gut pathologies. At the same time, we also found disease-specific host gene–microbe associations. For example, in CRC, the Syndecan-1 pathway, a host pathway that modulates tumour growth and progression, is correlated with microbial taxa such as *Parvimonas* and *Bacteroides fragilis* that are known to promote intestinal carcinogenesis[44,46,47]. These associations are not found in IBD or IBS, and are unique to CRC in our study cohort. The disease-specific pattern of host gene–microbe crosstalk suggests that gut microbes, either through direct interaction with host cells or through indirect interaction (for example, via production of specific metabolites), may regulate host gene expression differently in specific disease contexts.

Our study has several limitations. While we report the potential role of host gene–microbiome associations in the pathophysiology of GI disorders, our study identifies correlations, and we cannot directly infer causality from these results. Given the challenges associated with studying causal mechanisms in humans, future studies using cell culture or animal models would be useful in elucidating the causal role and directionality of associations between the gut microbiome and host gene regulation in these diseases[104]. Additionally, our analysis focused only on the taxonomic composition of the microbiome, and hence we could not characterize associations involving microbial genes and pathways. Lastly, there are several host and environmental variables that could potentially influence the microbiome and/or host gene expression, including age differences, sampling locations, diet, host genetics, treatment and medication history, which are not available across our disease cohorts. Thus, these factors are potential confounders that might influence our results.

Overall, our work demonstrates the power of integrating gut microbiome and host gene expression data to provide insights into their combined role in GI diseases, including CRC, IBD and IBS. Taken together, our results indicate that GI disorders are characterized by a complex network of associations between microbes and host genes. Although these associations can be disease-specific, we find cases where the same microbial taxon is associated with different host genes in each disease, and vice-versa: cases where the same host pathway is associated with different microbes in each disease. Although much effort in microbiome research has been directed towards identifying specific microbial taxa that are responsible for the pathogenesis of disease, our findings indicate that it is critical to incorporate host genomics data, as it can provide invaluable information on the potential mechanisms through which microbes can affect health. Our results represent an important step towards characterizing the association between gut microbiome and host gene regulation, and understanding the contribution of the microbiome to disease aetiology.

## Methods

**Overall study design, samples and data.** Overall, our study included 208 paired microbiome (16S rRNA) and host gene expression (RNA-seq) samples, which include 88 pairs of samples in the CRC cohort (44 tumour and 44 patient-matched normal)[3], 78 pairs of samples in the IBD cohort (56 patients and 22 controls)[22,25] and 42 pairs of samples in the IBS cohort (29 patients and 13 controls)[8] (Supplementary Tables 1 and 12–17). All datasets, except the host gene expression data for CRC, have been previously published as individual studies[3,8,22,25]. The original studies obtained written informed consent from study participants in each cohort. Details on randomization and blinding during data collection can be found in publications describing the original studies. No statistical methods were used to pre-determine sample sizes, but our sample sizes are similar to those reported in the previous publications[3,8,22,25]. Below, we describe in detail the sample collection, sequencing

and quality control for the CRC cohort host RNA-seq data, and summarize sample collection and data processing and acquisition for other datasets.

**CRC samples and data.** We used 88 pairs of gut microbiome and host gene expression samples from 44 patients, with primary tumour and normal tissue samples taken from each individual. The individuals in this cohort included 23 females and 21 males, with an average age of 65 years (median: 67, range: 17–91). Patient samples were characterized and described by Burns et al.[3]. Briefly, these de-identified samples were obtained from the University of Minnesota Biological Materials Procurement Network (Bionet). Tissue pairs were resected concurrently, rinsed with sterile water, flash frozen in liquid nitrogen and characterized by staff pathologists. Detailed cohort characteristics for this dataset are included in Supplementary Table 1.

*Host RNA-seq sequencing, alignment and quality control.* Total RNA was extracted using a previously established protocol[3,105]. Approximately 100 mg of flash-frozen tissue per sample were lysed by placing the tissue in 1 ml of Qiazol lysis reagent (Qiagen) and sonicating in a 65 °C water bath for 1–2 h. Nucleic acids were purified from the lysates using the Qiagen AllPrep DNA/RNA mini kit (Qiagen), quantified using a Nanodrop 2000 spectrophotometer (Thermo Fisher) and submitted for RNA sequencing to the University of Minnesota Genomics Center. Total eukaryotic RNA isolates were quantified using a fluorimetric RiboGreen assay, and once the samples passed the initial QC step ($\geq$1 µg and RNA integrity number $\geq$ 8), they were converted to Illumina sequencing libraries using Illumina's TruSeq stranded total RNA library prep (for details, see www.illumina.com). Truseq libraries were hybridized to a paired-end flow cell and individual fragments were clonally amplified by bridge amplification on the Illumina cBot. Once clustering was complete, the flow cell was loaded onto the HiSeq 2500 and sequenced using Illumina's SBS chemistry. Base call (.bcl) files for each cycle of sequencing were generated by Illumina Real Time Analysis (RTA) software. Primary analysis and index de-multiplexing were performed using Illumina's bcl2fastq v2.20.0.422, which outputs the demultiplexed FASTQ files.

A quality check of raw sequence FASTQ files was performed using FastQC software (version 0.11.5)[106]. Quality trimming was performed to remove sequence adaptors and low-quality bases using Trimmomatic with 3 bp sliding window trimming from the 3' end requiring minimum Q16 (phred33)[107]. FastQC was run on the resulting trimmed files to ensure good quality of sequences. The paired-end reads were mapped to NCBI v38 *H. sapiens* reference genome using HISAT2[108], resulting in an average alignment rate of 87.11% overall for 88 samples. We obtained a range of read counts between 14,365,657 and 31,530,487 aligned reads per sample, with an average of 22,475,688.2 and 22,697,605.5 aligned reads per sample. SAMtools was used for sorting and indexing the aligned bam files. After alignment, the 'Subread' package (version 1.4.6) within the 'featureCounts' programme was used to generate the transcript abundance file[109] (Extended Data Fig. 4).

*16S rRNA sequencing data.* The microbiome dataset used in this study was generated and published previously[3]. Briefly, total DNA was isolated from the flash-frozen tissue samples and their associated microbiomes by adapting an established nucleic acid extraction protocol[105]. DNA isolated from colon samples was quantified by quantitative PCR (qPCR), and the V5–V6 regions of the 16S rRNA gene were PCR amplified and sequenced using the Illumina MiSeq (v3 Kit) with 2 × 250 bp paired-end protocol. The forward and reverse reads were merged and trimmed using USEARCH v7[110]. The merged and filtered reads were used to pick operational taxonomic units (OTUs) with QIIME v.1.7.0[111]. We used the unnormalized and unfiltered OTU table in tab-delimited format, representing mucosal microbiome data from 44 tumour and 44 patient-matched colon tissue samples. We describe the steps for preprocessing microbiome data for integration analysis below.

**IBD samples and data.** We used previously generated and published host gene expression (RNA-seq) and mucosal gut microbiome (16S rRNA) data for the IBD cohort generated as part of the HMP2 project[22,25]. These include data from colonic biopsy samples collected from 78 individuals, including 56 individuals with IBD and 22 individuals without IBD ('non-IBD' in HMP2). Out of 56 IBD patients, 34 patients had Crohn's disease (CD) and 22 patients had ulcerative colitis (UC). The individuals in this cohort included 38 females and 40 males. Age at the time of sample collection is not reported in the metadata file available for this cohort (http://ibdmdb.org). The detailed protocol for sample collection and processing is described at http://ibdmdb.org/protocols. Briefly, biopsies were collected during the initial screening colonoscopy and stored in RNAlater for molecular data generation (host and microbial, stored at −20 °C). DNA and RNA were extracted from RNAlater-preserved biopsies using the AllPrep DNA/RNA universal kit from Qiagen. For microbiome profiling, bacterial genomic DNA was extracted from the total mass of the biopsied specimens using the MoBIO PowerLyzer tissue and cells DNA isolation kit. The 16S rDNA V4 region was PCR amplified and sequenced in the Illumina MiSeq platform using the 2 × 250 bp paired-end protocol. Read pairs were demultiplexed and merged using USEARCH v7.0.1090 and clustered into OTUs using the UPARSE algorithm[110,112]. For host gene expression data, mRNA

was extracted from biopsy samples, followed by RNA-seq library preparation using a variant of Illumina TruSeq stranded mRNA sample preparation kit. Sequencing was performed according to the manufacturer's protocols using either the HiSeq 2000 or HiSeq 2500 with 101 bp paired-end reads. Data were analysed using the Broad Picard Pipeline.

Detailed cohort characteristics for this dataset are included in Supplementary Table 1. We downloaded metadata, host RNA-seq data and microbiome data for these samples from http://ibdmdb.org in July 2018. We downloaded the unnormalized and unfiltered OTU table and host transcript read counts files in tab-delimited format. We describe the filtering and preprocessing steps for host gene expression and microbiome data below.

**IBS samples and data.** We used previously generated and published host gene expression (RNA-seq) and mucosal gut microbiome (16S rRNA) data for the IBS cohort[8]. These include data from colonic biopsy samples collected from 42 individuals, including 29 individuals with IBS and 13 healthy individuals (non-IBS). The individuals in this cohort included 31 females and 11 males, with an average age of 38 years (median: 35, range: 20–63). These samples were collected at Mayo Clinic Rochester and are described in detail by Mars et al.[8]. Briefly, for the microbiome data, DNA was extracted from biopsy sections using the QIAGEN PowerSoil kit (QIAGEN). The V4 region of the 16S rRNA gene was amplified, followed by paired-end 2×250 bp sequencing on an Illumina MiSeq. Trimming of adaptors, quality control and merging of reads were performed using Shi7[113]. Amplicon sequences were aligned to the 16S rRNA genes using BURST[114]. For host gene expression data, mRNA was extracted from biopsy samples, followed by RNA-Seq library preparation using the Illumina TruSeq RNA Library Prep Kit v2. Sequencing was performed on an Illumina HiSeq 2000 with 101 bp paired-end reads. Gene expression counts were obtained using the MAP-RSeq v.2.0.0 that consists of alignment with TopHat 2.0.12 against the human hg19 genome build, and gene counts with the Subread package 1.4.4[115–117].

Detailed cohort characteristics for this dataset are included in Supplementary Table 1. We obtained the unnormalized and unfiltered OTU table and host transcript read count files in tab-delimited format via personal communication with authors of the paper[8]. For some individuals, samples were collected at two timepoints. For these cases, we averaged the gene expression levels and microbiome abundance measurements across the two timepoints. This is supported by a recent study showing that 'omics' methods are more accurate when using averages over multiple sampling timepoints[118]. We describe the filtering and preprocessing steps for host gene expression and microbiome data below.

**Preprocessing host gene expression data.** For host gene expression data for each disease cohort, we used the 'biomaRt' R package (version 2.37.4) to only keep data for protein-coding genes[119]. We filtered out lowly expressed genes to retain genes that are expressed in at least half of the samples in each disease cohort. We performed variance stabilizing transformation using the R package 'DESeq2' (version 1.14.1) on the filtered gene expression read count data[120]. We filtered out genes with low variance, using 25% quantile of variance across samples in each disease cohort as cut-off. Performing these steps for RNA-seq data for each disease cohort separately resulted in a unique host gene expression matrix per disease for downstream analysis, including 12,513 genes in the CRC dataset, 11,985 genes in IBD dataset and 12,429 genes in IBS dataset.

**Preprocessing microbiome data.** We performed the following steps for microbiome data from each disease cohort separately. First, sequences that were classified as either having originated from Archaea, chloroplasts, known contaminants originating from laboratory reagents or kits, and soil- or water-associated environmental contaminants were removed from the OTU table as described earlier[121]. Next, we summarized the OTU table at the species (if present), genus, family, order, class and phylum taxonomic levels, and performed prevalence and abundance-based filtering to retain taxa found at 0.001 relative abundance in at least 10% of the samples.

To allow for identification of associations at any taxonomic level without repeating the analyses at each taxonomic rank, we combined summarized taxa matrices at different ranks into a combined taxa matrix. This approach could potentially lead to multi-counting of reads within a taxonomic group, leading to addition of correlated features in the taxa abundance matrix. This issue was mitigated by our penalization approach using lasso with stability selection (see Methods: 'Lasso regression analysis' and 'Stability selection for the lasso model'). Specifically, instead of picking multiple correlated microbial taxa from a given taxonomic clade, this approach only selects the microbial taxon out of a group of correlated taxa for which the abundance is most robustly associated with the expression of a host gene[28,122]. This approach allowed us to identify signals found at any taxonomic level and avoid missing potentially relevant associations by limiting the analysis to a single taxonomic level. At the same time, given the large number of features in high-dimensional datasets such as gene expression and microbiome data, our approach circumvents the computationally intensive analysis that would be required if each taxonomic level was analysed separately.

To account for compositionality effects in microbiome datasets, we tested two different approaches for performing centered log ratio (CLR) transformation

on our taxonomic data for each disease: (1) we concatenated the summarized taxa matrices (count data) into a combined taxa matrix, and then applied CLR transform on the combined matrix, (2) we CLR-transformed each taxon rank, and then concatenated them into a combined matrix. We verified whether these two transformation approaches were correlated with each other. To do so, we compared the taxa abundance profiles resulting from the two transformation approaches, and found that the two profiles are significantly correlated (*P* value < 0.05), with an average Pearson's correlation of 0.92 and an average Spearman correlation of 0.87 across samples in a dataset (see Extended Data Fig. 5 for example correlations between the taxa profiles resulting from the two transformations on a few randomly selected samples from the CRC microbiome dataset). This concordance between the taxa profiles resulting from the two transformation approaches implies that the transformation approach is unlikely to impact downstream results. The first approach generated a taxa profile that is compositionally coherent and has a uniform transformation across taxa in a dataset, whereas the second approach resulted in a taxa profile with multimodal distribution (corresponding to composition at each taxonomic rank) that might bias the variable selection by lasso approach. Hence, we adopted the first approach for transforming our taxonomic data.

As a result, we obtained a taxonomic abundance matrix for each disease cohort, which included 235 taxa in the CRC dataset, 121 taxa in the IBD dataset and 238 taxa in the IBS dataset. We observed that the number of unique taxonomic groups found in the IBD dataset was 40% of those found in the CRC and IBS datasets, thus implying that the IBD dataset has lower bacterial diversity than the other two disease datasets. This observation is consistent with previous studies that have shown reduced bacterial diversity in gut mucosal microbiome in patients with IBD compared with individuals without this disease[123–125], including the HMP2 study that generated and described the IBD dataset used in our study[22,25]. In addition, previous studies that generated and characterized the CRC and IBS datasets used in our work have reported increased microbial diversity in these conditions[3,8].

**Integrating host gene expression and gut microbiome data across diseases.** Our study includes three different disease cohorts with disparate protocols for sample collection, handling, preparation and sequencing. Previous studies have shown that differences in data generation protocols, including sample collection, storage, DNA/mRNA extraction, PCR amplification and sequencing can lead to potential batch effects regardless of the data processing pipeline used[126–128]. Studies have shown that even when using the same data curation pipeline, biases in the data generation pipeline and batch effects still influence the assignment of taxonomic composition and gene expression profiles[127,129]. Statistical approaches to correct for batch effects have been proposed for gene expression and microbiome datasets; however, most of these approaches are relevant to testing differences between cases and controls, and do not apply to integrative analyses[126,128].

Previous studies have also shown that different clustering approaches, such as operational taxonomic units (OTUs), zero-radius OTUs (zOTU) and amplicon sequence variants (ASVs), and specific pipeline settings may not have major influence on taxonomic classification compared with experimental factors such as choice of sequencing primer[130]. To check this in our dataset, we re-analysed a few samples from some of our disease dataset using the DADA2 pipeline[131], which uses ASV clustering, and compared results to the OTU clustering that we applied to process the data used in our study. We found that the estimated taxa profiles are correlated between the two approaches at different taxonomic levels. For example, in a few CRC samples, we found that taxa profiles obtained using OTU clustering and ASV clustering are significantly correlated at different taxonomic levels, including at the phylum (Spearman rho = 0.89, *P* value = 0.0068), class (Spearman rho = 0.68, *P* value = 0.029) and genus (Spearman rho = 0.6, *P* value = 0.088) levels. Despite these fairly correlated taxonomic profiles, it is hard to assess the overall bias due to differences in data processing on the downstream analyses. Additionally, it is difficult to disentangle and compare the bias due to differences in data processing pipelines with those due to differences in experimental factors, as the latter cannot be quantified in our study. These combined differences could potentially influence the assignment of taxonomic composition and gene expression profiles across disease cohorts, which may in turn impact downstream integration analyses. While it is difficult to fully eliminate these biases, we tried to minimize the overall batch effect across disease cohorts in our study by adopting a meta-analysis approach, where we performed our integration analysis and compared disease to control samples within each cohort separately, and combined the results across cohorts at the last analysis step. While meta-analysis approaches have disadvantages, such as reduced statistical power, they have been extensively used to minimize batch effects when integrating genomic data from multiple studies[128], and have recently proven useful in microbiome studies[132,133].

Another potential issue is the difference in sample size across disease cohorts and between the case and control groups within each cohort. These sample size differences can result in differences in statistical power when applying sparse CCA and lasso regression. This can impact the number of host gene–microbe associations and pathways identified in each disease cohort, and the number of overlapping associations and pathways identified across cohorts. We attempted to minimize this effect by applying a differential enrichment analysis that is more robust to different levels of statistical power due to sample size. Additionally, our

analysis focused only on the taxonomic composition of the microbiome, hence we could not characterize associations involving microbial genes and pathways. We note that it might be challenging to generalize some of the results found here, as microbiome profiles may vary across individuals and diseases. Hence, investigating the functional repertoire of microbial changes and its association with host genomic data would be a promising future direction.

**Procrustes analysis and Mantel test.** To assess overall correspondence between host gene regulation and gut microbiome composition in CRC, IBD and IBS, we performed Procrustes analysis in R using the 'vegan' package (version 2.4-5)[134]. For each disease cohort, we used Aitchison's distance on host gene expression data and Bray-Curtis distance on gut microbiome data as input to the Procrustes analysis[135]. The significance of rotation agreement was obtained using the 'protest()' function with 9,999 permutations. We also applied a Mantel test to verify overall correlation between dissimilarity matrices of host gene expression (Aitchison's distance) and gut microbiome abundance (Bray-Curtis distance) in each disease cohort using the vegan package (version 2.4-5) in R. We also used the Mantel test to verify the overall correspondence between paired data for each disease cohort, and found a similar pattern of significance of agreement as Procrustes analysis for CRC ($P$ value $= 0.0026$), IBD ($P$ value $= 0.2597$) and IBS ($P$ value $= 0.9525$). Significance was tested using 9,999 permutations.

**Overview of integration framework.** We developed a machine learning framework for integrating multi-omic high-dimensional datasets, such as host gene expression and gut microbiome abundance. Our integration approach has two parts: (1) sparse CCA[26,27] for identifying groups of host genes that associate with groups of gut microbial taxa to characterize pathway-level associations and (2) lasso penalized regression[28] for identifying specific associations between individual host genes and gut microbial taxa (Fig. 1a and Extended Data Fig. 1). We describe both methods in detail below. We applied our integration analysis to paired host gene expression data and gut microbiome data for each disease cohort separately to avoid any potential batch effects. For each disease cohort dataset, we conducted the integration analysis separately for the patient data (that is, CRC, IBD and IBS) and corresponding control data (non-CRC, non-IBD and non-IBS, respectively), and considered only associations that were found in patients but not in controls. We standardized and normalized the data in all host gene expression and microbiome datasets before the application of statistical methods to satisfy the distribution requirements of the statistical models.

**Sparse CCA.** We used sparse CCA to identify group-level correlations between paired host gene expression and gut microbiome data in each disease cohort. CCA identifies linear projection of two sets of observations into shared latent space that maximizes correlation between the two datasets[136]. Sparse CCA is adapted from CCA for high-dimensional settings to incorporate feature selection by utilizing $L1$ or lasso penalty in CCA[26]. The objective function of sparse CCA can be expressed as follows:

$$\text{maximize}_{u,v} u^T X^T Y v \text{ subject to } u^T X^T X u \leq 1, \; v^T Y^T Y v \leq 1, \; ||u||_1 \leq \lambda_1, ||v||_1 \leq \lambda_2$$

where, $X$ and $Y$ denote two data matrices with the same number of samples but different number of features (representing gut microbiome taxonomic composition data and host gene expression data, respectively); $u$ and $v$ are canonical loading vectors of $X$ and $Y$, respectively; $\lambda_1$ and $\lambda_2$ control lasso penalties of $u$ and $v$, respectively; $T$ denotes the transpose of a matrix.

For each disease cohort, we separately applied sparse CCA using the R (version 3.3.3) package 'PMA' (version 1.1), with gut microbiome taxonomic composition and host gene expression as two sets of variables to be correlated[137]. Below, we describe details on hyperparameter tuning, fitting sparse CCA models, enrichment analysis and visualization of sparse CCA output.

**Hyperparameter tuning and fitting for sparse CCA model.** We performed hyperparameter tuning to identify the sparsity penalty parameters for gut microbiome abundance ($\lambda_1$) and host gene expression ($\lambda_2$) data. Since the permutation search provided in the PMA package only performs a one-dimensional search in the tuning parameter space, we implemented a grid-search approach using leave-one-out cross-validation in R (version 3.3.3) for hyperparameter tuning. We selected penalty parameters that had the highest correlation under cross-validation. Using this approach, we identified $\lambda_1$ as 0.15 and $\lambda_2$ as 0.2 for CRC data, $\lambda_1$ as 0.177 and $\lambda_2$ as 0.333 for IBD data, and $\lambda_1$ as 0.4 and $\lambda_2$ as 0.1 for IBS data.

After identifying sparsity parameters, we fit the sparse CCA model to obtain subsets of correlated host genes and gut microbes, known as components. Each sparse CCA component includes non-zero weights (or canonical loadings) on gut microbes, and non-zero weights on a subset of host genes correlated with those gut microbes to capture joint variation in the two sets of observations. We computed the first 10 sparse CCA components for each disease cohort, performing a separate computation for case and control samples. Sparse CCA components were computed iteratively, informed by previously computed components, thus resulting in uncorrelated components[138].

**Significance of correlation for sparse CCA components.** We computed the significance of each pair of canonical variables (or a component) using the leave-one-out cross-validation approach in R (version 3.3.3). For a given component, we first used the penalty parameters determined above to compute the sparse CCA output, with one sample held out. We then computed the scores for the held-out sample, that is, we computed $\text{score}X_i = X_i u_{-i}$ and $\text{score}Y_i = Y_i v_{-i}$, where $i$ is the held-out sample, $X_i$ and $Y_i$ denote the values for the $i$th sample of the input data matrices $X$ and $Y$, and $u_{-i}$ and $v_{-i}$ are the canonical loadings estimated from the sparse CCA computation without the $i$th sample. We repeated this $n$ times, where $n$ is the total number of samples in the data, to obtain the vector of held-out scores. To assess the true strength of association and its significance, we used 'cor.test()' on the scores computed for the held-out samples. We corrected the $P$ values for multiple hypothesis testing using the Benjamini-Hochberg (FDR) method within each disease cohort, and determined significant components at FDR $< 0.1$. Note that here we are testing for significance at the component level, that is, for a group of host genes correlated with a group of gut microbes, rather than the significance of the individual features selected, which depends on the level of sparsity penalization.

Using this approach, we identified 7 significant components in CRC, with an average of 828 host genes and 8 gut microbes; 4 significant components in IBD, with an average of 2,095 host genes and 6 gut microbes; and 6 significant components in IBS, with an average of 577 host genes and 61 gut microbes (FDR $< 0.1$, Supplementary Tables 2–4).

**Enrichment analysis for sparse CCA.** To characterize host pathways enriched for the set of host genes associated with microbes in each component, we implemented an enrichment analysis in R (version 3.3.3). We implemented Fisher's exact test to assess pathway enrichment, where we used the set of host genes input to the sparse CCA analysis as background genes, and the set of host genes in a component as the genes of interest. We used the Kyoto Encyclopedia of Genes and Genomes (KEGG) and Pathway Interaction Database (PID) gene sets from the MsigDB canonical pathways collection[139,140]. To avoid pathways that are too large to provide any specific biological insights or too small to provide adequate statistical power, we excluded any pathway with either (1) fewer than 25 genes, (2) more than 300 genes or (3) fewer than 5 genes that overlapped between the genes of interest and the pathway. We combined the set of enriched host pathways for all significant components for a given disease dataset, corrected for multiple hypothesis testing within each disease cohort using the Benjamini-Hochberg (FDR) approach and determined significant host pathways at FDR $< 0.1$. For a given pathway, we assigned the component that gave the highest significance for this pathway. This analysis was performed separately for case and control data for each disease. To identify case-specific host pathways, we used a two-part approach: (1) we identified pathways that were only significantly enriched in cases (FDR $< 0.1$) but not in controls and (2) we identified pathways that were significant in both the cases and controls at FDR $< 0.1$, and performed differential enrichment analysis for pathways in cases versus controls as described below. We retained pathways that were significantly enriched at FDR $< 0.1$ and differentially enriched in cases versus controls at FDR $< 0.2$. Finally, we combined the pathways from parts (1) and (2) to obtain case-specific pathways at FDR $< 0.1$.

**Differential enrichment analysis of pathways.** We performed differential enrichment of pathways in cases versus controls by implementing a comparative log odds-ratio approach in R[141,142]. To do so, we first computed the $z$-score for the odds ratio for the $i$th pathway in cases:

$$z_{i,\text{case}} = \log(\delta_i)/\text{SE}(\delta_i),$$

where $\delta_i$ is the odds-ratio for the $i$th pathway in cases, and SE $(\delta_i)$ is the standard error for the $i$th pathway in cases, which is computed using the four elements, $n_1$ to $n_4$, of the $2 \times 2$ contingency table used in the enrichment analysis for the $i$th pathway as follows:

$$\text{SE}(\delta_i) = \sqrt{1/n_1 + 1/n_2 + 1/n_3 + 1/n_4}.$$

Similarly, we computed $z_{i,\text{ctrl}}$ for the same pathway in the controls. Next, we computed a comparative log odds-ratio for the $i$th pathway overlapping between cases and controls as follows:

$$z_{i,\text{ case}-\text{ctrl}} = \frac{\log(\delta_{i,\text{case}}) - \log(\delta_{i,\text{ctrl}})}{\text{SE}(\delta_{i,\text{case,ctrl}})}.$$

The greater the value of $z_{i,\text{case}-\text{ctrl}}$, the greater the odds that a pathway is differentially enriched in case versus control than by chance. $P$ values were inferred assuming normal approximations, and corrected for multiple hypothesis testing using the Benjamini-Hochberg (FDR) approach.

**Visualizing disease-specific and shared host pathways and components from sparse CCA.** To determine 'shared' host pathways (that is, host pathways for which gene expression correlates with gut microbes across all three disease cohorts) and 'disease-specific' host pathways (that is, host pathways for which gene expression

correlates with gut microbes in only one of the three disease cohorts), we computed overlaps between significant case-specific host pathways determined above across the three disease cohorts. Given the overlap across the curated gene sets from MsigDB, we controlled for redundancy across pathways for visualization purposes. To do this, we identified similar pathways on the basis of their relative overlap in terms of the set of genes using an overlap coefficient. The overlap coefficient between two pathways is defined as the number of common genes between the pathways divided by the number of genes in the pathway with fewer genes. Specifically, the overlap coefficient is represented as follows:

$$\text{overlap}(X, Y) = \frac{|X \cap Y|}{\min\left(|X|, |Y|\right)}$$

For the top 15 most significant host pathways (FDR < 0.1) discovered for each shared and disease-specific set (Supplementary Table 18), we computed pairwise similarity between pathways as overlap coefficients and used a maximum allowed similarity score of 0.5 as a cut-off. Using the pairs of pathways that satisfied the cut-off, we computed the connected components to identify clusters of overlapping pathways. For visualization purposes, we selected a representative pathway from each connected component, prioritizing the pathway with the highest number of genes (Fig. 2a and Supplementary Table 4). We visualized host pathway enrichment using the R package 'ggplot2' (version 3.2.1).

For visualizing components corresponding to selected host pathways or common host genes across diseases (Fig. 2b–d), we ordered host genes and taxa by their absolute coefficients in the component, and selected the top 10 host genes and taxa for representation. If multiple taxa originating from the same lineage occurred in a component, we selected the one with the highest coefficient to reduce redundancy, thus representing the taxa with the largest contribution from a given lineage. The size of host genes and gut microbial taxa were scaled by the absolute value of their corresponding coefficients in a given component. All sparse CCA components were visualized using Cytoscape (version 3.5.1)[143].

**Lasso regression analysis.** We used lasso penalized regression to identify specific associations between individual host genes and gut microbial taxa within each disease cohort[28]. We implemented a gene-wise model using expression for each host gene as response and abundances of microbiome taxa as predictors, to identify microbial taxa that are correlated with a host gene. An ordinary least squares (OLS) regression is not suitable for this task since OLS results in unstable solutions under high-dimensional settings or when $p \gg n$, that is, the number of predictors $p$ is much higher than the number of samples $n$. Additionally, we expected the abundance of only a few microbial taxa to correlate with the expression of each host gene. To address this, we used lasso regression, which is similar to multivariate OLS, except that it uses shrinkage or regularization to perform variable selection, thus picking only a few taxa that associate with a host gene's expression.

To account for other factors that can impact host gene expression or microbiome composition, each model also included covariates in the predictor matrix (that is, microbiome abundance table) for gender (male or female), disease subtype for IBD (Crohn's disease or ulcerative colitis), disease-subtype for IBS (constipation (IBS-C) or diarrhoea (IBS-D)).

The lasso model estimates the lasso regression coefficient $\hat{\beta}$ by minimizing the following:

$$\sum_{i=1}^{n} \left( y_i - \beta_0 - \sum_{j=1}^{p} \beta_j x_{ij} \right)^2 + \lambda \sum_{j=1}^{p} |\beta|,$$

where $n$ is the number of samples, $p$ is the number of predictors (taxa and other covariates), $1 \leq i \leq n$, $1 \leq j \leq p$, $y$ is the response (host gene expression), $x$ is the predictor (taxa abundance and other covariates) and $\lambda$ is the tuning parameter, $\lambda \geq 0$.

In addition to minimizing the residual sum of squares (first term in the equation), lasso minimizes the $l_1$ norm of the coefficients (second term in the equation), which has an effect of forcing some of the coefficients to zero as the value of the tuning parameter $\lambda$ increases. Thus, lasso performs feature or variable selection that leads to sparse models.

We implemented a lasso regression framework using the R (version 3.3.3) package glmnet (version 2.0-13), which uses cyclical coordinate descent to compute a regularization path[144]. Our framework executes a lasso regression for each host gene's expression as response and abundances of microbial taxa and values of other covariates as predictors. We used leave-one-out cross-validation to estimate the tuning parameter $\lambda$, which was used to fit the final model on a given disease dataset.

We then performed inference for the lasso model using a regularized projection approach known as desparsified lasso. The desparsified lasso uses the asymptotic normality of a bias-corrected version of the lasso estimator to obtain 95% confidence intervals and $P$ values for the coefficient of each predictor (microbe) associated with a given host gene[145]. We used the R package 'hdi' (version 0.1-7) that implements the desparsified lasso approach for estimation of confidence intervals and hypothesis testing in high-dimensional and sparse settings[145,146]. We

then corrected for multiple hypothesis testing using the Benjamini-Hochberg (FDR) method.

**Stability selection for the lasso model.** Since the lasso model is sensitive to small variations of the predictor variable, we used stability selection to pick out robust microbes associated with a host gene[122]. Stability selection is a resampling-based method that can be combined with different variable selection procedures in high-dimensional settings, including lasso. Briefly, stability selection with lasso proceeded as follows:

Step 1. Select a random subset of the data.

Step 2. Fit the lasso model with a randomly perturbed penalty term in the neighbourhood of the 'best' penalty $\lambda$. Record the set of selected variables (microbes).

Step 3. Repeat steps (1) and (2) $K$ times.

Step 4. Compute the frequency of selection $f_i$ per variable (microbe) across all trials.

Step 5. Select the variables (microbes) that are selected with a frequency of at least $f_{\text{thr}}$, a pre-specified threshold value. Thus, we selected a set of stable variables (microbes) such that $f_i \geq f_{\text{thr}}$.

The overall idea is that, if the same variables (microbes) are repeatedly selected when the parameters are perturbed, then they are robust variables. Stability selection also controls for family-wise error rate, thus controlling for false positives in addition to the FDR approach mentioned above[122]. In our analysis, we used the R package 'stabs' (version 0.6-3) to perform stability selection[147]. Specifically, we used the following parameters in the process described above: in Step 1, a random subset of size $n/2$ of data was selected, where $n$ is total number of samples; in Step 3, $K = 100$; and in Step 5, $f_{\text{thr}} = 0.6$, that is, a predictor (microbe) selected in at least 60% of the fitted models is considered stable. The choice of these parameters are in accordance with the proposal of stability selection by Meinshausen & Bühlmann[122].

Finally, we performed an intersection between associations identified by stability selection here and associations identified at FDR < 0.1 by the lasso model described above. We filtered out any significant and stability-selected host gene–gender and host gene–disease subtype (applicable to IBD and IBS) associations from the output to retain significant and stability-selected host gene–microbe associations at FDR < 0.1.

**Parallel execution of lasso analysis on supercomputing nodes.** We leveraged supercomputing nodes to implement a parallel processing framework to allow scalable computation for the high-dimensional datasets. We implemented a parallel framework for executing the gene-wise lasso analysis, where we parallelized execution of lasso models on host genes across multiple nodes and cores on a compute cluster from Minnesota Supercomputing Institute. Our framework scalably computes gene-wise models for over 12,000 host genes, where each model includes hundreds of microbial taxa as covariates. We used job arrays to parallelize our analysis on multiple nodes on the cluster. Additionally, we used the R packages 'doParallel' (version 1.0.15) and 'foreach' (version 1.4.7) to run parallel processes on multiple cores of each compute node. Our parallel processing framework using 5 compute nodes took an average of 5 h 30 min per disease cohort.

**Enrichment analysis for lasso output.** To characterize biological functions for the host genes that were found to be associated with specific gut microbes in a disease cohort by the lasso framework, we implemented an enrichment analysis in R (version 3.3.3) using Fisher's exact test. We used the set of expressed genes input to the lasso analysis as the background genes, and the set of host genes associated with gut microbes in patient samples as genes of interest. We used the KEGG, PID and REACTOME gene sets from the MsigDB canonical pathways collection[139,140]. To avoid pathways that were too large to provide any specific biological insights or too small to provide adequate statistical power, we excluded from our analysis any pathways with more than 85 genes, fewer than 10 genes, or fewer than 5 genes that overlapped between the pathway and the genes of interest. Out of 1,881 host pathways, these criteria filtered out 297 pathways for being too large, 299 pathways for being too small and an average of 1,186 pathways for not having sufficient overlap with the genes of interest, resulting in an average of 99 pathways that were tested for enrichment in each disease cohort. We are aware that this approach may filter out some potential pathways of interest from our analysis. However, even if included, these pathways are unlikely to yield significant associations due to lack of statistical power. The $P$ values obtained from Fisher's exact test were adjusted for multiple testing using the Benjamini-Hochberg (FDR) approach.

**Comparing case versus control associations and pathways.** Using lasso regression and stability selection, we identified associations for cases and controls with non-zero lasso regression coefficients for each disease cohort. To identify associations that were found only in cases but not in controls within a disease cohort, we checked for any potential overlap between case and control associations, without subsetting associations using any $P$ value or FDR cut-off. We found no overlapping host gene–microbe associations between cases and controls in any disease cohort, which could be driven by underlying biological differences between case and control conditions within each cohort. In addition, downsampling the cases to match the controls also did not yield any overlapping associations. Thus,

we conclude that the designation of host gene–microbe associations as specific to cases and controls is robust to the significance cut-off and sample size differences in our study cohort.

Another potential approach to identify case-specific associations would be to use an interaction term between the independent variable (that is, microbial taxa) and disease status in the lasso model, and determine associations that are significant in cases but not in controls. Incorporating such an interaction within the lasso framework is not straightforward, and various approaches have been proposed (for an overview, see Lim & Hastie[148]). However, these approaches do not explicitly allow for different sparsity structures in cases and controls (that is, where an effect is present for cases but not for controls, or present for controls but not for cases), which is crucial to our interpretation and subsequent analyses. Moreover, this approach is computationally challenging for our implementation for the following reason: for each disease cohort, we fit over 12,300 gene-wise lasso models on average, where each model uses the expression for a host gene as response and the abundance of about 200 microbial taxa on average as predictors/independent variables. Adding an interaction term between each predictor (that is, microbial taxon) and disease status will lead to doubling the number of predictors in each gene's model, resulting in about 400 predictors per model, and including over $12,300 \times 200 = 2,460,000$ additional terms for assessment of model fit per disease cohort. Hence, we did not include interaction terms in our models.

Using enrichment analysis for host genes associated with specific gut microbes, we identified host pathways for cases and controls in each disease cohort. To account for case versus control comparison at the pathway level, we performed the following analysis: (1) from the results of our enrichment analysis, we retained the set of host pathways that were tested for enrichment in both cases and controls within each disease cohort without using any P value or FDR cut-off and (2) we tested for differential enrichment of these pathways between case and control groups using a comparative log odds-ratio approach (described above). For IBD and IBS cohorts, we found no pathways enriched in both case and control groups in step (1) above, implying that host pathways enriched in cases are indeed case-specific in these cohorts. In CRC, we found two host pathways that were common in both cases and controls in step (1); however, they were not differentially enriched in cases versus controls at FDR < 0.1 (step 2), implying that they were not necessarily specific to cases. Hence, we filtered out these two pathways from consideration for case-specific pathways.

In addition to accounting for overlaps between cases and controls at the association level and at the pathway level, we also accounted for any overlaps between host genes that were associated with microbes in cases and controls. We only used case-specific host genes, that is, host genes that were found to be associated with microbes in cases but not in controls, to perform enrichment analysis to determine case-specific pathways in each disease cohort. Using this approach, we identified 18 host pathways that are unique to each disease, including 4 CRC-specific, 9 IBD-specific and 5 IBS-specific pathways that associate with unique gut bacteria (FDR < 0.1, Supplementary Table 9).

To identify any loss of function in disease (that is, associations that are found in controls but not in cases), we performed analysis at pathway level to determine functional trends in control associations compared to case associations. To do so, we first determined any host pathways that were found enriched only in controls but not in cases at FDR < 0.1. In IBD and IBS, no pathways were found to be enriched in controls at FDR < 0.1, so we could not identify any functional trends in control associations for these two disease cohorts. In CRC, top 10 control-specific pathways at FDR < 0.1 are related to transcriptional regulation, rRNA expression, DNA methylation and other such general cellular function categories that did not provide any useful insight on loss of function in disease. Additionally, as mentioned above, we did not find any pathways that were differentially enriched in controls vs cases in any disease cohort. Hence, given that we did not find any specific functional trends in the control associations in CRC, IBD and IBS cohorts, we only focused on case-specific associations and pathways across diseases in this study.

**Identification and visualization of taxa and genes that are shared or distinct across associations in diseases.** To visualize association patterns for gut microbes and host genes that are shared between associations across diseases, we examined common microbes/genes between association-pairs across disease cohorts. Given the difference between the number of host genes and gut microbial taxa identified across diseases, we first determined overlaps in host genes and microbial taxa in input datasets and gene–taxa associations identified across disease cohorts. To do this, we used a two-step process:

(1)  To examine common and distinct features in input datasets, we considered the host genes and taxa that were used as input to the lasso integration pipeline, and calculated the overlap in these host genes and taxa across cohorts (Supplementary Tables 19 and 20). We computed the pairwise overlap as an overlap coefficient, which is a measure of similarity between two sets and is defined here as the number of common genes (or taxa) between the two disease datasets divided by the number of genes (or taxa) in the dataset with fewer genes (or taxa). Overall, we found that an average of 84% of host genes are commonly expressed between diseases (Supplementary Table 19), while the remaining 16% of host genes might be expressed in one disease

cohort, but not in the other two. On the other hand, only about 37% of input gut microbial taxa overlapped between diseases (Supplementary Table 20), indicating that a majority of taxa were specific to each disease cohort. This is consistent with previous research that have shown high dissimilarity between disease-associated microbial communities[132,149].

(2)  Next, we calculated the overlap between the set of host genes found associated with gut microbes only in one of the disease cohorts (that is, disease-specific genes in our identified associations) and the set of host genes used as input in the other two diseases (Supplementary Table 21). Similarly, we calculated the overlap between gut microbial taxa found associated with host genes only in one disease (that is, disease-specific taxa in our identified associations) and the set of taxa used as input in the other two diseases (Supplementary Table 22). Overall, we found similar trends as described above. We found that an average of 80% of disease-specific host genes in the identified associations were also included as input in the other two diseases (Supplementary Table 21). The remaining 20% of disease-specific host genes were probably not expressed in the other two conditions. On the contrary, we found that on average, only 12% of disease-specific taxa in the identified associations were included as input taxa in the other two diseases (Supplementary Table 22). This is in line with our observation of disease-specific trends for gut microbial taxa used as input to the integration pipeline, as described above.

Given these patterns, we considered all the host genes and taxa that were identified after preprocessing the input dataset, which allowed us to identify disease-specific as well as overlapping host gene/taxa between associations across diseases.

To visualize associations for gut microbes that are associated with host gene expression in multiple diseases (Fig. 4a), we identified common microbes between all possible overlaps between diseases (Fig. 4a, Networks 1–4), host genes that associate with these common microbes in each disease, and all associations involving these microbes and genes in each disease (FDR < 0.1). Next, we grouped gene–taxa associations identified per disease by shared taxa, sorted them by FDR value and picked the top gene–taxa association per shared taxon until we obtained at most 10 associations per disease (FDR < 0.1, Supplementary Table 10).

Similarly, for visualizing associations for host genes that are associated with gut microbes in multiple diseases (Fig. 4b), we identified shared host genes between all possible overlaps between diseases (Fig. 4b, Networks 1–4), and host gene–taxa associations per disease for these host genes shared across diseases. We sorted the associations by FDR-adjusted p-values, i.e. q-values (ordered first by q-value in CRC associations, followed by q-value in IBD associations and finally, by q-value in IBS associations, depending on the overlapping set under consideration). We picked the top 10 genes from this merged output and identified at most the top 10 associations involving these genes in each disease for the overlapping set under consideration (FDR < 0.1, Supplementary Table 11). Since lasso gives biased estimates of the coefficients, we used Spearman correlation coefficient (rho) to depict strength of association for visualizing host gene–taxa associations. All the associations in Fig. 4 were visualized using Cytoscape v3.5.1, where shared features are in grey and disease-specific features are in disease-specific colours[143].

**Notes on our approach and comparison to previous studies.** While the disease cohorts used in our study have been previously published[3,8,25,28], to the best of our knowledge, our study appears to be the first to perform a comprehensive characterization of host gene–microbiome crosstalk within and across these disease cohorts. In the previous study that described the CRC cohort, Burns et al.[3] characterized the tumour-associated microbiome, and how it varies in composition compared to the microbiome of adjacent matched normal colon tissue. For example, they reported loss in abundance of multiple taxa within the order Bacteroidales in tumour-associated microbiota compared with normal samples. Here we found that Bacteroidales is associated with host genes *CCR2* and *FPR1*, which are part of the tumour-associated IL-10 signalling pathway. The previous study that described the IBD cohort compared IBD versus non-IBD samples, and found differences in transcriptional activity and abundance among taxa belonging to class Clostridia, and dysregulation of immune-related host pathways in disease state[25,28]. Our integrative analysis for the same cohort revealed associations between immunoinflammatory pathways and members within class Clostridia, including Peptostreptococcaceae and *Clostridium sensu stricto 1*. In the original study describing the IBS cohort, Mars et al.[8] found overrepresentation of *Streptococcus* species in patients with IBS compared with healthy individuals, and identified associations between faecal microbes, such as Peptostreptococcaceae, with host genes implicated in peptidoglycan binding. Our analyses revealed several important associations between closely interacting tissue-adherent microbiome and host genes and pathways in IBS, including associations between *Streptococcus* and host genes that modulate macrophage inflammatory response, and between Peptostreptococcaceae and host pathways that regulate intestinal homoeostasis and inflammation.

An important contribution of our work is a machine learning-based integrative framework for characterization of host gene–microbe associations across human diseases. Although few recent studies have investigated associations between host transcriptome and gut microbiome in human gut disorders, our analysis uses a unique analytical technique that has several advantages[21–24]. First, as opposed

to analyses that rely on calculating pairwise correlations between features (for example, Dayama et al.[24]), our approach does not require restricting the data to a predetermined subset of taxa or genes of interest. In addition, compared to Procrustes analysis, which is commonly used for finding overall correspondence between paired datasets, our approach does not only detect overall association, but can also find specific associations between gut microbial taxa and host genes (using lasso) and pathways (using sparse CCA), shedding light on potential biological mechanisms of association. Furthermore, our approach can be applied to other types of multi-omic dataset, including microbial metabolomic and metagenomic data[8]. Lastly, our analysis incorporates data from several diseases, identifying commonalities across conditions as well as disease-specific patterns.

Our study uncovered key insights at the systems level; for example, we found that gut microbes that have been associated with all three diseases, such as *Streptococcus*, associate with different host genes in each disease, suggesting that the same microbial taxon can contribute to different health outcomes by potentially regulating the expression of different host genes in the colon. We also identified numerous specific hypotheses in the form of disease-specific associations; for example, we found that: Bacteroidales is associated with host genes *CCR2* and *FPR1* in the IL-10 signalling host pathway in colorectal cancer; Peptostreptococcaceae is associated with *MAPK3* and *VIPR1* that are part of G protein-coupled receptors pathways in inflammatory bowel disease; and *Bacteroides massiliensis* is associated with the host gene *PLA2G4A*, a member of the prostaglandin biosynthesis pathway, in irritable bowel syndrome.

**Ethics statement.** For the colorectal cancer cohort, all research conformed to the Helsinki Declaration and was approved by the University of Minnesota Institutional Review Board, protocol 1310E44403. For the inflammatory bowel disease and irritable bowel syndrome cohorts, ethical approval is described in their respective publications[8,22,25].

**Reporting Summary.** Further information on research design is available in the Nature Research Reporting Summary linked to this article.

## Data availability

Raw data for host RNA-seq for the CRC cohort are available on the NCBI Sequence Read Archive (SRA) under BioProject ID PRJNA816986. Raw data for previously published 16S rRNA sequencing for the CRC cohort can be accessed at PRJNA284355[3]. Raw data for previously published 16S rRNA sequencing and host RNA-seq for the IBD cohort can be accessed at PRJNA398089 and GSE111889, respectively[22,25]. Raw data for 16S rRNA sequencing and host RNA-seq for the IBS cohort can be accessed at PRJEB37924 and GSE146853, respectively[8]. Processed data tables for host transcriptomics and microbiome data for each disease cohort have been included as supplementary tables (Supplementary Tables 12–17).

## Code availability

Code used for integration analyses performed in the paper is available at https://github.com/blekhmanlab/host_gene_microbiome_interactions. We have also included a tutorial for our integration pipeline at https://github.com/blekhmanlab/host_gene_microbiome_interactions/tree/main/Tutorial.

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

## Acknowledgements

We thank the IBD HMP2 consortium for making the dataset publicly available; the Blekhman Lab members for their comments and suggestions on the manuscript; W. Wang and G. Al-Ghalith for their feedback. This work was supported by NIH grant R35-GM128716 (to R.B.), a Minnesota Partnership for Biotechnology and Genomics grant (to R.B.), a University of Minnesota Doctoral Dissertation Fellowship (to S.P.), NIH grant R01-GM130622 (to E.F.L.) and R01-DK114007 (to P.C.K.). This work was carried out in part by resources provided by the Minnesota Supercomputing Institute.

## Author contributions

S.P. and R.B. conceived, designed and supervised the study. M.B.B. and R.B. collected samples and produced data for the CRC cohort. R.A.T.M., T.W., D.K. and P.C.K. collected samples and generated data for the IBS cohort. S.P. performed data analysis with contributions from R.A.T.M., T.W. and M.B.B. S.P. and R.B. wrote the manuscript, with contributions from M.B.B., R.A.T.M., T.W., B.A., E.F.L., P.C.K. and D.K.

## Competing interests

D.K. is a Senior Scientific Advisor and T.W. is the Director of Bioinformatics at Diversigen, a company involved in the commercialization of microbiome analysis. P.C.K. is an ad hoc consultant for Pendulum Therapeutics, IP group inc., Novome Biotechnologies, and Intrinsic Medicine. The other authors declare no competing interests.

## Additional information

**Extended data** is available for this paper at https://doi.org/10.1038/s41564-022-01121-z.

**Correspondence and requests for materials** should be addressed to Ran Blekhman.

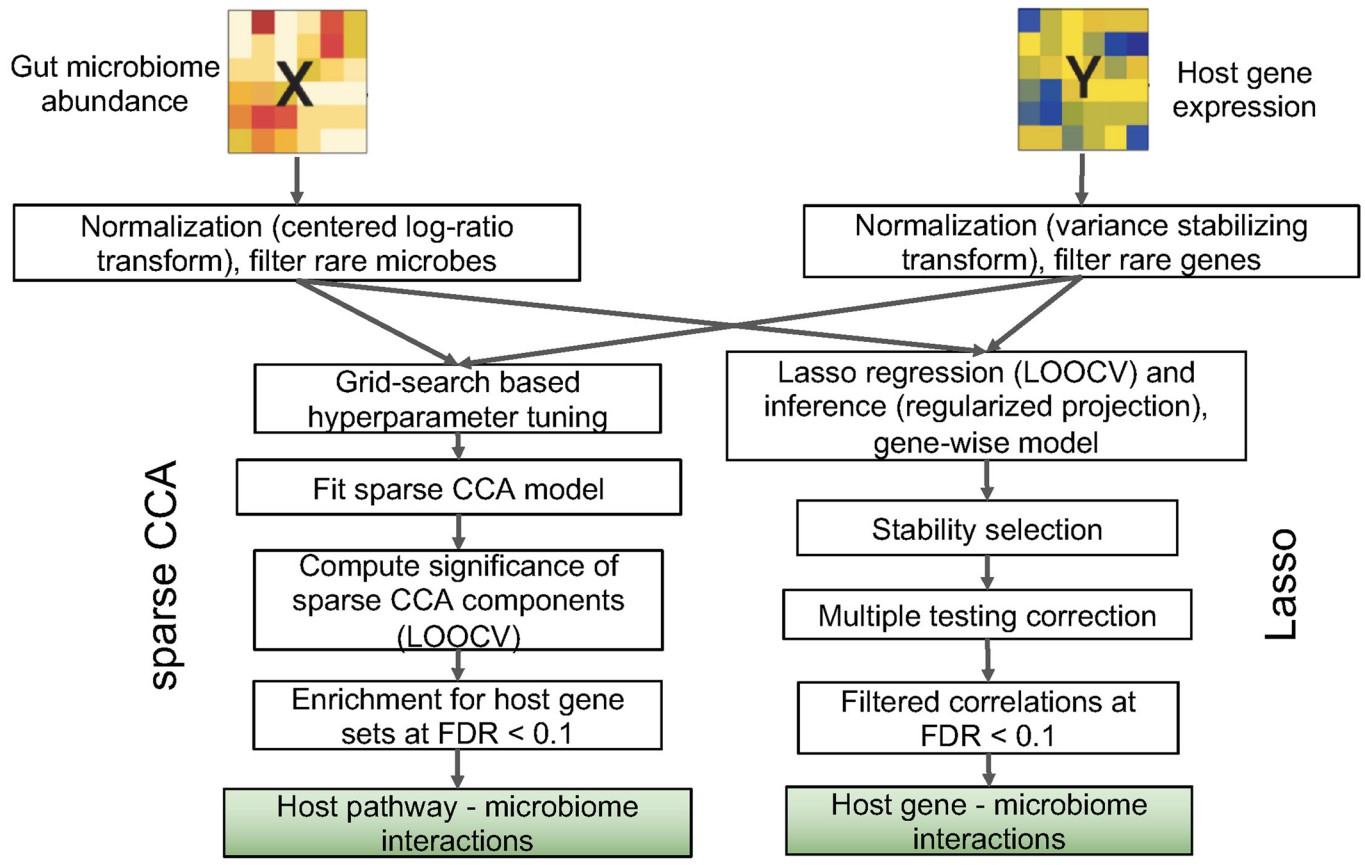

**Extended Data Fig. 1 | Overview of host gene-microbiome integration pipeline.** Steps for integrating gut microbiome abundance and host gene expression data using sparse CCA and lasso approaches.

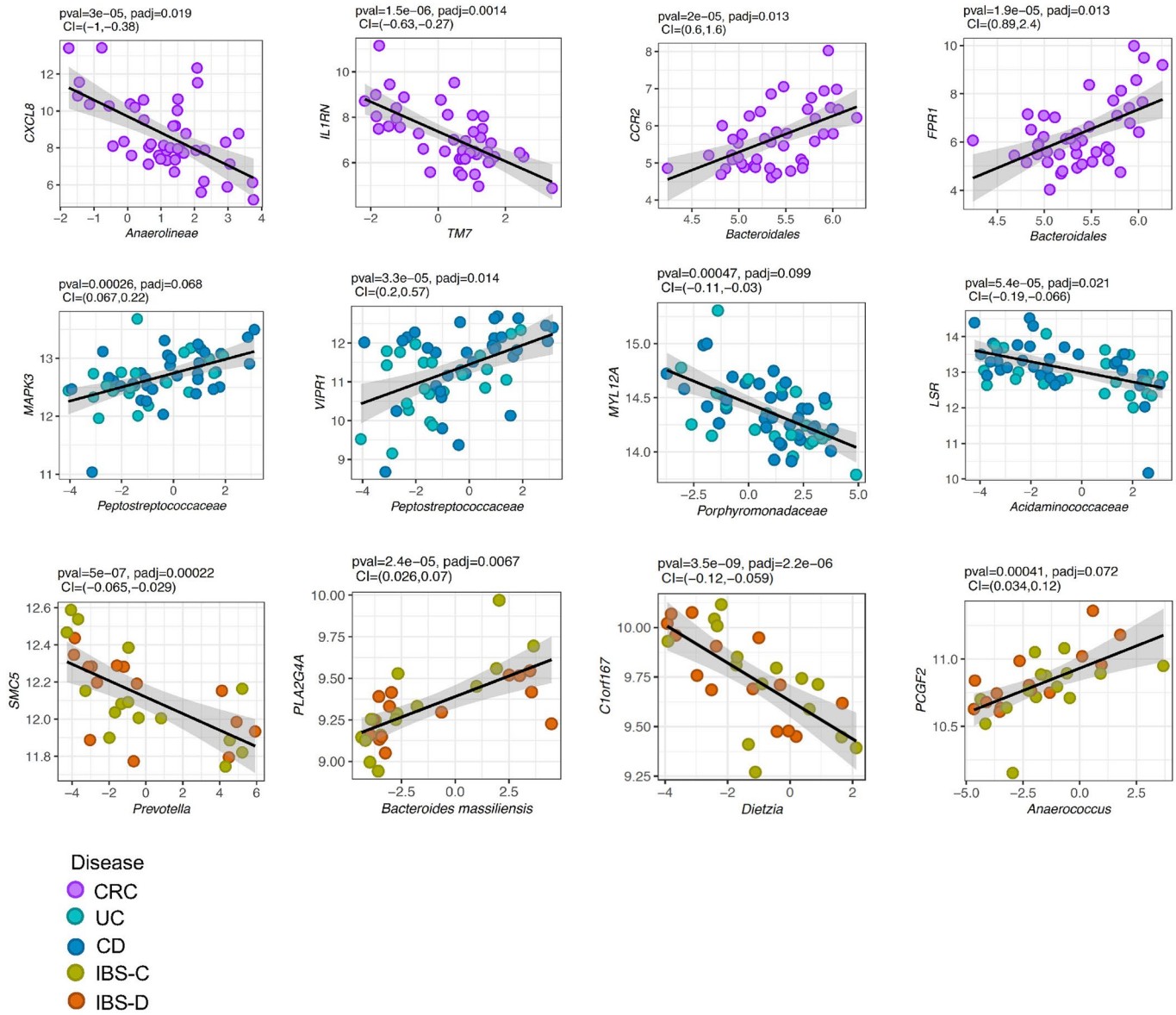

**Extended Data Fig. 2 | Examples of specific associations between gut microbial taxa and host genes in colorectal cancer (CRC), Inflammatory Bowel Disease (IBD), and Irritable Bowel Syndrome (IBS).** The top row shows specific associations for CRC (n = 44), the middle row shows the specific associations for IBD (n = 56), and the bottom row shows the specific associations for IBS (n = 29). The x-axis represents normalized abundance of microbial taxa, and the y-axis represents normalized expression of the host gene. The black line represents the line of best fit using a linear model and the grey shaded area represents 95% confidence interval. Desparsified lasso was used to obtain 95% confidence intervals (CI) and p-values (pval) for the association. P-values were adjusted for multiple comparisons using Benjamini-Hochberg (FDR) method (padj). CRC: colorectal cancer, UC: Ulcerative colitis, CD: Crohn's Diseases, IBS-C: Irritable bowel syndrome - constipation, IBS-D: Irritable bowel syndrome - diarrhoea.

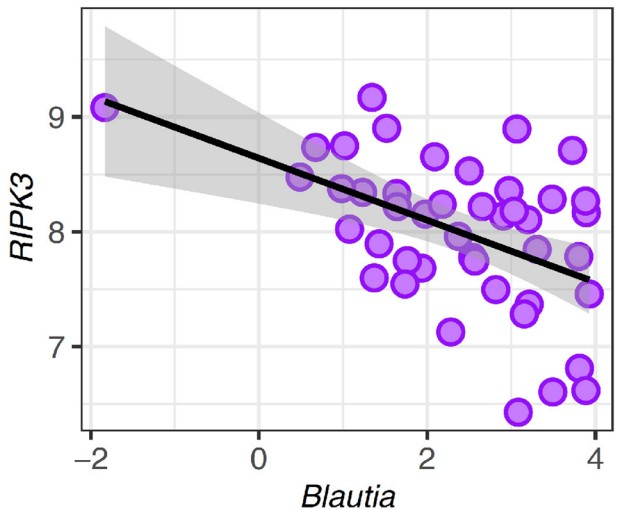

pval=1.3e−05, padj=0.0096
CI=(−0.42,−0.16)

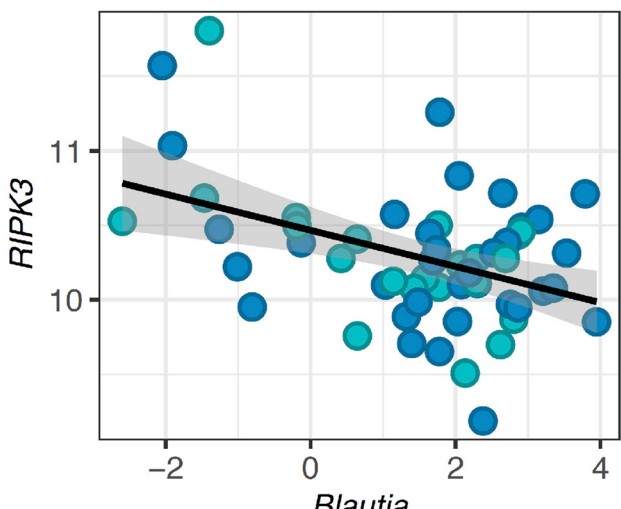

pval=0.00013, padj=0.041
CI=(−0.22,−0.071)

Disease
○ CRC
○ UC
○ CD

**Extended Data Fig. 3 | A common host gene-microbe association between colorectal cancer (CRC) and Inflammatory Bowel Disease (IBD).** The association for CRC (n = 44) is shown on the left, and for IBD (n = 56) is shown on the right. The black line represents the line of best fit using a linear model and the grey shaded area represents 95% confidence interval. Desparsified lasso was used to obtain 95% confidence intervals (CI) and p-values (pval) for the association. P-values were adjusted for multiple comparisons using Benjamini-Hochberg (FDR) method (padj). CRC: colorectal cancer, UC: Ulcerative colitis, CD: Crohn's Diseases.

A

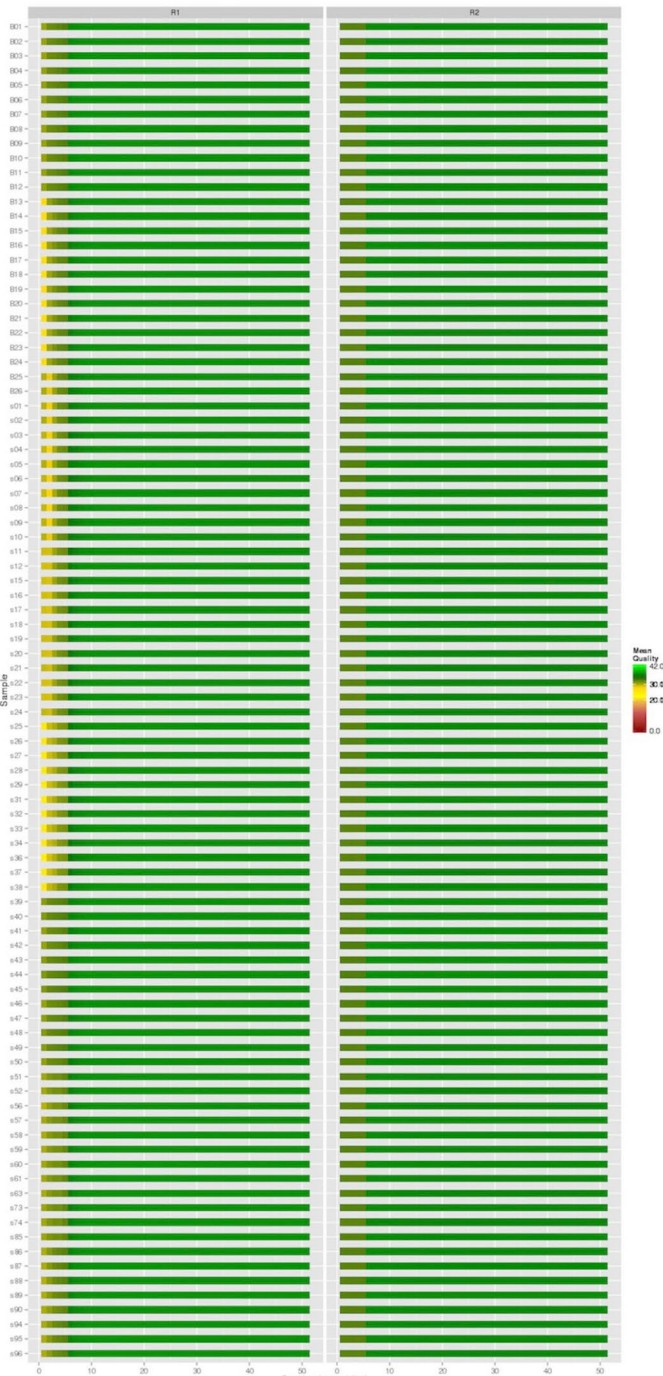

B

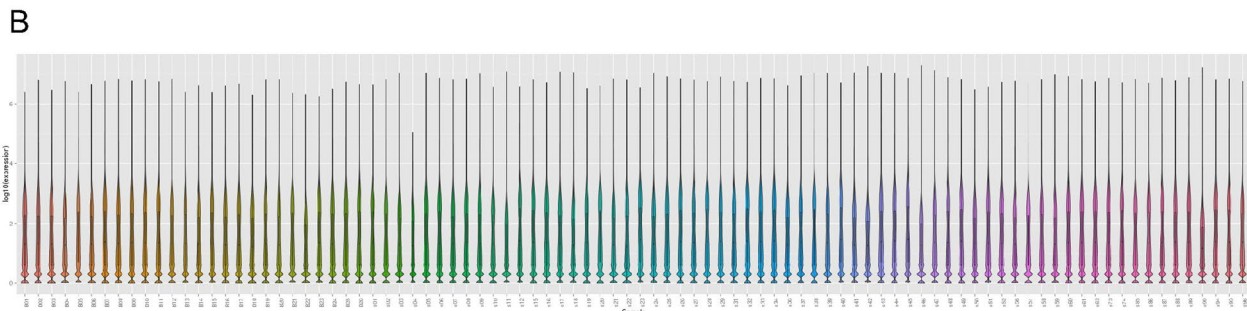

**Extended Data Fig. 4 | Quality control and transcript quantification of host RNA-seq data for colorectal cancer samples.** A. Average phred score for forward (R1) and reverse (R2) reads output by FastQC. B. Distribution of host gene expression (log10(expression) value) for each sample quantified by Subread package.

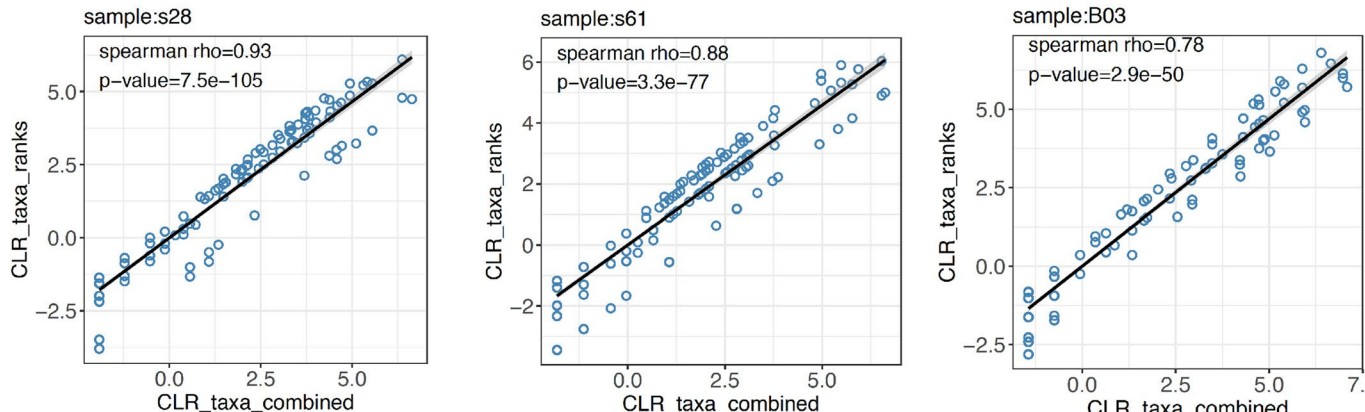

**Extended Data Fig. 5 | Scatterplots showing correlation between taxa profiles generated from two transformation approaches across three random samples from colorectal cancer cohort.** The x-axis shows taxa profile resulting from one approach, where we summarize taxa ranks, combine summarized rank matrices, and CLR transform the combined matrix (CLR_taxa_combined), and the y-axis represents taxa profiles resulting from the second approach, where we summarize taxa ranks, CLR transform each taxa rank, and combine the CLR-transformed taxa ranks (CLR_taxa_ranks). The black line represents the line of best fit using a linear model. Spearman's rho and associated two-tailed p-values are shown.

# nature research

# Reporting Summary

Nature Research wishes to improve the reproducibility of the work that we publish. This form provides structure for consistency and transparency in reporting. For further information on Nature Research policies, see our Editorial Policies and the Editorial Policy Checklist.

## Statistics

For all statistical analyses, confirm that the following items are present in the figure legend, table legend, main text, or Methods section.

| n/a | Confirmed | |
|---|---|---|
| ☐ | ☒ | The exact sample size (*n*) for each experimental group/condition, given as a discrete number and unit of measurement |
| ☐ | ☒ | A statement on whether measurements were taken from distinct samples or whether the same sample was measured repeatedly |
| ☐ | ☒ | The statistical test(s) used AND whether they are one- or two-sided *Only common tests should be described solely by name; describe more complex techniques in the Methods section.* |
| ☐ | ☒ | A description of all covariates tested |
| ☐ | ☒ | A description of any assumptions or corrections, such as tests of normality and adjustment for multiple comparisons |
| ☐ | ☒ | A full description of the statistical parameters including central tendency (e.g. means) or other basic estimates (e.g. regression coefficient) AND variation (e.g. standard deviation) or associated estimates of uncertainty (e.g. confidence intervals) |
| ☐ | ☒ | For null hypothesis testing, the test statistic (e.g. *F*, *t*, *r*) with confidence intervals, effect sizes, degrees of freedom and *P* value noted *Give P values as exact values whenever suitable.* |
| ☒ | ☐ | For Bayesian analysis, information on the choice of priors and Markov chain Monte Carlo settings |
| ☒ | ☐ | For hierarchical and complex designs, identification of the appropriate level for tests and full reporting of outcomes |
| ☐ | ☒ | Estimates of effect sizes (e.g. Cohen's *d*, Pearson's *r*), indicating how they were calculated |

*Our web collection on statistics for biologists contains articles on many of the points above.*

## Software and code

Policy information about availability of computer code

| Data collection | No software was used in data collection. |
|---|---|
| Data analysis | All the code for analysis was written in R (version 3.3.3). The plots were generated in R using ggplot2 (version 3.2.1). Descriptions for data analyses used in the paper are described in detail in the Methods section. Code used for analyses in the paper is available at https://github.com/blekhmanlab/host_gene_microbiome_interactions. Here are the software/packages (with version number) used in our analyses: <br><br> - FastQC (version 0.11.5) <br> - Subread (version 1.4.6) <br> - biomaRt (version 2.37.4) <br> - DESeq2 (version 1.14.1) <br> - vegan (version 2.4-5) <br> - glmnet (version 2.0-13) <br> - PMA (version 1.1) <br> - hdi (version 0.1-7) <br> - stabs (version 0.6-3) <br> - ggplot2 (version 3.2.1) <br> - Cytoscape (version 3.5.1) |

For manuscripts utilizing custom algorithms or software that are central to the research but not yet described in published literature, software must be made available to editors and reviewers. We strongly encourage code deposition in a community repository (e.g. GitHub). See the Nature Research guidelines for submitting code & software for further information.

## Data

Policy information about availability of data

All manuscripts must include a data availability statement. This statement should provide the following information, where applicable:
- Accession codes, unique identifiers, or web links for publicly available datasets
- A list of figures that have associated raw data
- A description of any restrictions on data availability

Raw data for host RNA-seq for CRC cohort is available on the NCBI Sequence Read Archive (SRA) under BioProject ID: PRJNA816986. Raw data for previously published 16S rRNA sequencing for the CRC cohort can be accessed at PRJNA2843553 [1]. Raw data for previously published 16S rRNA sequencing and host RNA-seq for the IBD cohort can be accessed at PRJNA398089 and GSE111889, respectively [2,3]. Raw data for 16S rRNA sequencing and host RNA-seq for the IBS cohort can be accessed at PRJEB37924 and GSE146853, respectively [4]. Processed data tables for host transcriptomics and microbiome data for each disease cohort have been included as supplemental tables (Supplementary Tables S12–S17). We used the KEGG, PID, and REACTOME gene sets from MsigDB canonical pathways collection [5].

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

# Field-specific reporting

Please select the one below that is the best fit for your research. If you are not sure, read the appropriate sections before making your selection.

☒ Life sciences          ☐ Behavioural & social sciences          ☐ Ecological, evolutionary & environmental sciences

For a reference copy of the document with all sections, see nature.com/documents/nr-reporting-summary-flat.pdf

# Life sciences study design

All studies must disclose on these points even when the disclosure is negative.

| | |
|---|---|
| Sample size | We have used all publicly available data that includes both microbiome profiles and gene expression quantification from gut mucosal samples. Although no sample size calculation was performed, our results indicate that the sample size was sufficient to identify statistically significant patterns in the data. |
| Data exclusions | No data was excluded from the analyses. |
| Replication | For study reproducibility, we have made publicly available all the data and code underlying the analysis and results presented (see above and in the manuscript for the Data Availability and Code availability). Replication was only performed as part of this study examining results that are shared across disease cohorts. There is no replication of the results within each disease cohort, for the reason that for each disease cohort, only data from a single study was available, and thus replication was not possible. |
| Randomization | The participants were allocated within each cohort (IBD, IBS, and CRC) based on their disease status (either disease or non-disease control). We controlled for relevant covariates based on available metadata on factors that could impact gene expression or microbiome composition. Specifically, we included covariates for gender (male or female) in all three disease cohorts, disease-subtype for IBD (Crohn's Disease or ulcerative colitis), and disease-subtype for IBS (constipation (IBS-C) or diarrhea (IBS-D)). |
| Blinding | Since we used publicly available, previously collected data, information on blinding during data collected can be found in the respective manuscripts describing the primary datasets. In our study, blinding was not relevant since this study is not a trial and did not assess patient outcomes; since this is a hypothesis-generating study, we did not have specific expectations for results that could bias our data analysis or interpretation. |

# Reporting for specific materials, systems and methods

We require information from authors about some types of materials, experimental systems and methods used in many studies. Here, indicate whether each material, system or method listed is relevant to your study. If you are not sure if a list item applies to your research, read the appropriate section before selecting a response.

## Materials & experimental systems

| n/a | Involved in the study |
|-----|------------------------|
| ☒ ☐ | Antibodies |
| ☒ ☐ | Eukaryotic cell lines |
| ☒ ☐ | Palaeontology and archaeology |
| ☒ ☐ | Animals and other organisms |
| ☐ ☒ | Human research participants |
| ☒ ☐ | Clinical data |
| ☒ ☐ | Dual use research of concern |

## Methods

| n/a | Involved in the study |
|-----|------------------------|
| ☒ ☐ | ChIP-seq |
| ☒ ☐ | Flow cytometry |
| ☒ ☐ | MRI-based neuroimaging |

# Human research participants

Policy information about studies involving human research participants

| | |
|---|---|
| Population characteristics | The study did not involve any newly collected samples, only previously collected, published datasets [1-4]. The CRC cohort comprised of 44 patients with colorectal cancer, including 23 females and 21 males, with an average age of 65 years (median: 67, range: 17–91) [1]. The IBD cohort comprised of 78 individuals, including 56 individuals with IBD, and 22 individuals without IBD ("non-IBD" in HMP2) [2,3]. Out of 56 IBD patients, 34 patients had Crohn's disease (CD) and 22 patients had ulcerative colitis (UC). The individuals in this cohort included 38 females and 40 males. Age at the time of sample collection is not reported in the metadata file available for this cohort (http://ibdmdb.org). The IBS cohort is comprised of 42 individuals, including 29 individuals with IBS, and 13 healthy individuals (non-IBS) [4]. The individuals in this cohort included 31 females and 11 males, with an average age of 38 years (median: 35, range: 20–63). The original studies obtained written informed consent from study participants in each cohort.<br><br>References:<br>1. Burns, M. B. et al. Virulence genes are a signature of the microbiome in the colorectal tumor microenvironment. Genome Med. 7, 55 (2015).<br>2. Lloyd-Price, J. et al. Multi-omics of the gut microbial ecosystem in inflammatory bowel diseases. Nature 569, 655–662 (2019).<br>3. Integrative HMP (iHMP) Research Network Consortium. The Integrative Human Microbiome Project. Nature 569, 641–648 (2019).<br>4. Mars, R. A. T. et al. Longitudinal Multi-omics Reveals Subset-Specific Mechanisms Underlying Irritable Bowel Syndrome. Cell 184, 1460-1473 (2020). |
| Recruitment | The study did not involve any newly collected samples, only previously collected, published datasets. Information on recruitment of participants is available in the publications describing the primary samples:<br><br>- Burns, M. B. et al. Virulence genes are a signature of the microbiome in the colorectal tumor microenvironment. Genome Med. 7, 55 (2015).<br>- Mars, R. A. T. et al. Longitudinal Multi-omics Reveals Subset-Specific Mechanisms Underlying Irritable Bowel Syndrome. Cell 184, 1460-1473 (2020).<br>- Lloyd-Price, J. et al. Multi-omics of the gut microbial ecosystem in inflammatory bowel diseases. Nature 569, 655–662 (2019).<br>- Integrative HMP (iHMP) Research Network Consortium. The Integrative Human Microbiome Project. Nature 569, 641–648 (2019). |
| Ethics oversight | For the colorectal cancer cohort, all research conformed to the Helsinki Declaration and was approved by the University of Minnesota Institutional Review Board, protocol 1310E44403. For the inflammatory bowel disease and irritable bowel syndrome cohorts, ethical approval is described in their respective publications [1,2,3].<br><br>References:<br>1. Mars, R. A. T. et al. Longitudinal Multi-omics Reveals Subset-Specific Mechanisms Underlying Irritable Bowel Syndrome. Cell 184, 1460-1473 (2020).<br>2. Lloyd-Price, J. et al. Multi-omics of the gut microbial ecosystem in inflammatory bowel diseases. Nature 569, 655–662 (2019).<br>3. Integrative HMP (iHMP) Research Network Consortium. The Integrative Human Microbiome Project. Nature 569, 641–648 (2019). |

Note that full information on the approval of the study protocol must also be provided in the manuscript.

