## [Peer Review File · Nature Microbiology]

Peer Review Information

Journal: Nature Microbiology

Manuscript Title: Identification of shared and disease-specific host gene-microbiome associations across human diseases using multi-omic integration

Corresponding author name(s): Dr Ran Blekhman

Editorial Notes:

Redactions – unpublished data	Parts of this Peer Review File have been redacted as indicated to maintain the confidentiality of unpublished data.
Redactions – confidential patient information	Parts of this Peer Review File have been redacted as indicated to maintain patient confidentiality.
Redactions – published data	Parts of this Peer Review File have been redacted as indicated to remove third-party material.
Redactions – reviewer opt-out	Parts of this Peer Review File have been redacted as indicated as we could not obtain permission to publish the reports of reviewer no. XX .
Reviewer comments in marked-up manuscript	In their review of the [first/second/third/...] version of this manuscript, reviewer no. XX added their comments to the manuscript file. These comments, excluding minor textual revisions, have been copied into this Peer Review File.

Reviewer Comments & Decisions:

Decision Letter, initial version:
--

20th April 2021

Dear Ran,

Thank you for your patience while your manuscript "Shared and disease-specific host gene-microbiome interactions across human diseases" was under peer-review at Nature Microbiology. It has now been seen by 3 referees, whose expertise and comments you will find at the end of this email.

Although they find your work of some potential interest, they have raised a number of concerns that will need to be addressed before we can consider publication of the work in Nature Microbiology.

In particular, referees #1 and #3 ask that you use “associate” rather than “interact”, and referee #2 asks that you tone down overstatements throughout. Referees #1 and #3 also note that sample size and therefore statistical power varies across disease groups, and referee #2 notes that data processing also varied across datasets. Please clarify how this was accounted for and add a discussion to the text. Referee #1 has suggestions for alternative methods for some analyses, referee #2 asks for additional code details including a tutorial for the provided scripts, which should be added to GitHub, and referee #3 has several requests for more information regarding the datasets and cohorts used, and suggests improvements for the figures, including figure 2, which we encourage.

We will overrule the request from referee #2 to include a proof of principle experiment to validate your findings.

Should further experimental data allow you to address these criticisms, we would be happy to look at a revised manuscript.

Please include a data availability statement as a separate section after Methods but before references, under the heading "Data Availability". This section should inform readers about the availability of the data used to support the conclusions of your study. This information includes accession codes to public repositories (data banks for protein, DNA or RNA sequences, microarray, proteomics data etc...), references to source data published alongside the paper, unique identifiers such as URLs to data repository entries, or data set DOIs, and any other statement about data availability. At a minimum, you should include the following statement: “The data that support the findings of this study are available from the corresponding author upon request”, mentioning any restrictions on availability. If DOIs are provided, we also strongly encourage including these in the Reference list (authors, title, publisher (repository name), identifier, year). For more guidance on how to write this section please see: <http://www.nature.com/authors/policies/data/data-availability-statements-data-citations.pdf>

* Include a “Response to referees” document detailing, point-by-point, how you addressed each referee comment. If no action was taken to address a point, you must provide a compelling argument. This response will be sent back to the referees along with the revised manuscript.

* If you have not done so already we suggest that you begin to revise your manuscript so that it conforms to our Article format instructions at <http://www.nature.com/nmicrobiol/info/final->

submission. Refer also to any guidelines provided in this letter.

[Redacted]

Note: This url links to your confidential homepage and associated information about manuscripts you may have submitted or be reviewing for us. If you wish to forward this e-mail to co-authors, please delete this link to your homepage first.

Nature Microbiology is committed to improving transparency in authorship. As part of our efforts in this direction, we are now requesting that all authors identified as 'corresponding author' on published papers create and link their Open Researcher and Contributor Identifier (ORCID) with their account on the Manuscript Tracking System (MTS), prior to acceptance. This applies to primary research papers only. ORCID helps the scientific community achieve unambiguous attribution of all scholarly contributions. You can create and link your ORCID from the home page of the MTS by clicking on 'Modify my Springer Nature account'. For more information please visit www.springernature.com/orcid.

If you wish to submit a suitably revised manuscript we would hope to receive it within 6 months. If you cannot send it within this time, please let us know. We will be happy to consider your revision, even if a similar study has been accepted for publication at Nature Microbiology or published elsewhere (up to a maximum of 6 months).

With best wishes,

[Redacted]

Reviewer Expertise:

Referee #1: gut microbiome, computational biology
 Referee #2: gut microbiome, machine learning, GI disease
 Referee #3: microbiome, computational biology

Reviewer Comments:

Reviewer #1 (Remarks to the Author):

Summary:

The authors present a well-written manuscript that brings together host gene expression (bulk RNA-seq of gut biopsy tissue) and 16S amplicon sequencing of the gut microbiome. This data set is exciting, in that it combines two types of data that are rarely integrated and it spans different case-control cohorts that represent three distinct disease states: CRC, IBD, and IBS (i.e. results can be compared across diseases for consistencies or differences). This work provides exciting new insights into associations between host gene expression and the ecological composition of the gut across three gut-relevant pathologies. The meta-analysis element of this work is especially cool, to see what associations are shared or distinct across diseases.

While I'm enthusiastic about the data set and reported results, I have some methodological concerns and concerns about interpretation that should be addressed. Specifically, the individual subgroups analyzed vary in size and can be quite small (13 healthy controls in the case of IBS data set). I had some concerns about statistical power and comparing 'significant' associations identified across groups of varying sizes.

Specific Comments:

Throughout the manuscript, you use the term 'interact' when describing correlations/associations between host genes and microbial abundances. I don't think this is semantically appropriate due to the implicit causality inherent to this term. I know different fields can use terms differently. In ecology or molecular genetics, 'interaction' usually implies a direct (and directed) interaction between entities (e.g. predator+ prey or transcription factor/repressor + gene expression). In linear regression, 'interaction' implies something like: $[y \sim \beta_1(x) + \beta_2(z) + \beta_3(x*z) + \text{intercept} + \text{residuals}]$, where β_3 is the 'interaction' term. I don't think you can claim any of these interpretations. I'd suggest sticking to the term 'association' throughout.

lines 90-98: I understand the authors' point about having a multi-variate approach, but they might overstate it a bit (seems somewhat unnecessary, and therefore a tad distracting). They are using established methods (i.e. CCA & LASSO). Univariate tests with multi-test corrections, like Pearson or Spearman, are valid ways to go (pros and cons to any approach). For example, LASSO has the limitation of being a linear method (but it has a nice advantage in being fairly interpretable, compared to other ML approaches). The authors mention statistical power, but they end up running separate tests to assess the 'significance' of the various LASSO coefficients...so it probably ends up being somewhat similar to univariate+correction for power (although you are getting rid of a lot of the colinear stuff with the L1 penalty, so maybe it's a bit better). Anyway, I think the overall approach is fine -- I'd just present it as it stands, without needing to justify it.

Line 147: Based on my own (anecdotal) experience, I've found that Procrustes can sometimes give unreasonably low and less consistent p-values, when compared against a Mantel test. I think it's fine here, but you may want to double-check against Mantel.

lines 167-171: I have some concerns about how your control vs. case analysis was conducted. From what I read, it seems like you identify your gene-microbe associations within the healthy cohort and within the disease cohort separately, and then exclude associations found in the healthy cohort, correct? My concern arises from the different sizes of these cohorts in IBD and IBS. Your statistical power is changing across these groups, and if you're using some kind of p-value cutoff for picking associations then you'll be comparing statistical apples to oranges. Did you account for this somehow? Coefficients are usually much more robust to differences in sample sizes than p-values (e.g. correlation coefficients). I'm not sure how exactly this relates to LASSO coefficients due to the hyperparameter tuning and regularization steps... But this makes me a tad queasy. If I was doing this in a classic GLM, I'd add an interaction term between the independent variable and disease status and identify significant interaction effects (i.e. association is significant in one population but not in the other, or the association reverses sign). How flexible is your LASSO framework? Can you add an interaction term and model cases and controls together, rather than splitting them up? I feel like this would be more robust.

I see that your FDR q-value thresholds vary throughout the manuscript. This makes the interpretation of the results a bit harder. Why not stick to a consistent threshold?

lines 398-422: I found this section difficult to read. For good reason -- it's hard to intuitively explain the 'shared microbes' and 'shared genes' stuff. But I think this section could be improved. For example, the statement 'shared host genes' was initially confusing to me -- aren't all host genes shared? Shared microbes made more sense to me because, I thought, we each maybe share ~30% of the species in our guts. However, the things being shared are not the genes or the microbes, but their membership in gene-microbe association pairs. Please try to smooth out the exposition here -- I think it'll really help with the flow of the text.

line 441: Highlighting the use of the term 'interaction' again (it appears frequently, but making a point here to repeat my prior comment). I think it has a causal implication here that isn't appropriate. Please remove this term throughout.

lines 670-677: You included abundance info summarized at multiple taxonomic levels, all in the same abundance matrix. I see that you CLR transformed these profiles? I'm not sure if this is totally appropriate. These profiles can't really be considered relative abundances anymore, because you're multi-counting reads from taxa (i.e. you have species abundances, but you also have the abundance of the genus and the reads mapping to that species also contribute to the genus abundance). Why not calculate relative abundances for each taxonomic rank (i.e. relative frequencies that all sum to 1.0) and CLR transform each of these ranks separately, and then combine the profiles?

lines 738-743: I think I'm ok with this, but it's pretty ad hoc. And going back to my prior comment, this kind of univariate p-value calculation kinda defeats your prior argument about the power of a multi-variate approach (if by 'power' you are interested in identifying individual 'significant' features).

line 913: How exactly did you remove gender and disease sub-type associations?

line 939: How valuable is it to present more (potentially spurious) features by using a less stringent 0.2 cutoff? Why not just pick a consistent cutoff across the manuscript?

Reviewer #2 (Remarks to the Author):

Priya et al. present a computational approach for analysing host-gene and microbiome interactions and apply this approach to three datasets (CRC, IBD, IBS). While the underlying idea is interesting, I have several concerns:

Major concerns:

- Data processing was not done with the same pipeline. Previously generated host transcript read count files and OTU tables were used from the original publications of the data. Studies have shown that the same dataset processed with different bioinformatic pipelines result in very different outcomes. While the authors analyse the datasets separately to avoid batch effects (due to differences in sample collection, DNA extraction, library preparation) this is a bias that could be avoided by reprocessing the data. Also, the published datasets should be briefly described in the methods section with details on sample collection and data processing.
- The authors state that their aim is to facilitate new insights into the molecular mechanisms for different disease and that their analysis presents the power of integrating gut microbiome and host gene expression data to provide disease insights. The authors did a very thorough comparison with other published studies, but which specific novel hypotheses were generated? What type of validation experiments would the authors suggest based on groups of microbes and groups of genes that correlate in abundance/expression? Could a proof-of-principle experiment be added to validate their approach?
- For the analysis they “considered only associations that were found in patients and not in controls”. What about loss of function in disease?
- The identification of generalisable disease-specific characteristics is challenging on such small datasets with less than 100 participants per disease. The microbiome is highly variable making the replication of disease-associated species even across large cohort studies with hundreds of samples challenging. Hence investigating functional implications of microbial shifts may be a more promising avenue. Conclusions such as “these associations are not found in IBD or IBS, and are unique to CRC” are too general/overstating the findings and it should be clearly stated that this refers to this specific cohort analysis.
- The authors describe the use of supercomputing nodes for their analysis. Did the analysis of these relatively small dataset require that much compute power? How will this be scaled to hundreds of samples? Or is the purely a feature provided as part of their method? Some information on how long the cohort analysis took should be provided.
- It’s great that the authors made the code available on github, however, a description of the different scripts and a tutorial is missing.
- What is their theory behind the disease-specific host-microbe crosstalk? Difference in host genetics?

Minor concerns:

- The authors should clarify in the manuscript (not just the methods section) that most of the presented data is published data and also compare their analysis to the original findings.
- Different FDR threshold were used at various places.
- Why did the IBD dataset only have 121 taxa? Both the CRC and IBS dataset had twice as many taxa.

Reviewer #3 (Remarks to the Author):

In this manuscript by Priya et al., the relationship between microbiome composition and host gene expression is examined in colonic mucosal biopsy samples. In particular, case control studies of three separate diseases were conducted (colorectal cancer, inflammatory bowel disease, and irritable bowel syndrome), with RNA-seq being used to measure host gene expression and 16S rRNA gene sequencing being used to measure microbiome taxonomic composition. Sparse canonical correlation analysis and clr-lasso were applied to the data to identify microbe-host gene associations, and those were compared across diseases. Overall, this is a really impressive study: methods are suitable and very well described, results are informative, and the manuscript is very well-written, with many connections to previous literature to put results in context. A study like this can serve as a framework for others interested in combining these two types of datasets while conserving power to detect associations.

Minor comments:

1. Results are difficult to interpret when multiple levels of bacterial taxonomy are analyzed at the same time, rather than level by level. For example, in Figure 2, both Micrococcaceae and Micrococcales are highlighted under the IBD RAC1 pathway panel in B. Is this largely the same signal, as these groups are nested, or are there distinct signals at different taxonomic levels? It may be more interpretable for the reader to focus on one level for the main text (maybe OTU or genus level), with other levels falling to the supplement.
2. Page 4, line 136/methods: More description of the case/control design for each of the three disease types listed would be beneficial for the reader. What were the criteria used to "match" case with control? For some of the diseases (IBD and IBS), patients and controls are described as "pairs", but there are unequal case and control sample sizes.
3. Are the sample characteristics largely the same across the disease cohorts, or are there differences that might be confounded with "disease-specific configurations" of the microbiome and host gene expression? For example, if there were age differences, sex biases, different sampling locations, or treatments that might also influence gene expression or the microbiome, those would be worth clarifying for the reader and/or mentioning as a caveat to interpretation of the study.
4. Related to sample sizes mentioned above, the three diseases have different sample sizes, which seem to match with the number of associations detected in each individual disease and the overlaps between them (CRC and IBD have the most samples and the most identified associations (including overlap), while IBS has fewer (including fewer overlaps). Some discussion of this effect of sample size would be useful for the reader, as well as if the unbalanced case/control designs has any effect on the power of applying a lasso.
5. In addition to sample size, the number of detected human genes and taxa differ between disease sets. For the comparisons of overlapping vs. disease specific taxa/genes, were "core" sets of taxa and gene examined that were held constant between disease set?
6. p8, line 264: I suggest using "associate" rather than "interact", as "interact" could be interpreted by some as a direct, physical interaction (which it might not be).
7. p28: How many host pathways were filtered out for being too small, too large, or having not many genes that overlap? Are there potential pathways of interest that aren't being examined here, but could be interesting?
8. Figure 2/p50: Overall, I find the visualizations extremely effective for this manuscript, especially considering the many dimensions this data is explored in. The one exception is with the sCCA result visualizations in parts B/C of Figure 2 and the supplemental figure on page 50. It's hard to tell which genes are shared across disease subtypes, because host genes are listed in a different orientation in each subtype. I suggest having the entire host pathway illustrated (maybe in gray) and then coloring in the significant host genes within disease-subtype. That would preserve the order, allowing for better highlighting of the similarities and differences for the reader.

Author Rebuttal to Initial comments

Reviewer #1 (Remarks to the Author):

Summary:

The authors present a well-written manuscript that brings together host gene expression (bulk RNA-seq of gut biopsy tissue) and 16S amplicon sequencing of the gut microbiome. This data set is exciting, in that it combines two types of data that are rarely integrated and it spans different case-control cohorts that represent three distinct disease states: CRC, IBD, and IBS (i.e. results can be compared across diseases for consistencies or differences). This work provides exciting new insights into associations between host gene expression and the ecological composition of the gut across three gut-relevant pathologies. The meta-analysis element of this work is especially cool, to see what associations are shared or distinct across diseases.

While I'm enthusiastic about the data set and reported results, I have some methodological concerns and concerns about interpretation that should be addressed. Specifically, the individual subgroups analyzed vary in size and can be quite small (13 healthy controls in the case of IBS data set). I had some concerns about statistical power and comparing 'significant' associations identified across groups of varying sizes.

Response:

We thank the reviewer for their positive feedback and appreciate the constructive suggestions and concerns regarding comparisons between case and control subgroups that vary in size. To address this, we have now included multiple new analyses and modifications in the manuscript. These new analyses are described in detail in response to **Comment 4** (pages 5-9 of the current document), and edits in the revised manuscript are on **pages 29, 33, and 34**.

Specific Comments:

Reviewer 1, Comment 1

Throughout the manuscript, you use the term 'interact' when describing correlations/associations between host genes and microbial abundances. I don't think this is semantically appropriate due to the implicit causality inherent to this term. I know different fields can use terms differently. In ecology or molecular genetics, 'interaction' usually implies a direct (and directed) interaction between entities (e.g. predator+ prey or transcription factor/repressor + gene expression). In linear regression, 'interaction' implies something like: $[y \sim \beta_1(x) + \beta_2(z) + \beta_3(x*z) + \text{intercept} + \text{residuals}]$, where β_3 is the 'interaction' term. I don't think you can claim any of these interpretations. I'd suggest sticking to the term 'association' throughout.

Response:

We thank the reviewer for this suggestion. We agree that the term “interaction” can be misleading, and, as suggested by the reviewer, we have replaced all occurrences of “interact” with “associate” throughout the manuscript. Similarly, we have replaced all occurrences of “interaction” with “association” everywhere, including in the title of the manuscript, which is now “Shared and disease-specific host gene-microbiome associations across human diseases”. This has resulted in a total of 83 changes throughout the manuscript.

Reviewer 1, Comment 2

lines 90-98: I understand the authors' point about having a multi-variate approach, but they might overstate it a bit (seems somewhat unnecessary, and therefore a tad distracting). They are using established methods (i.e. CCA & LASSO). Univariate tests with multi-test corrections, like Pearson or Spearman, are valid ways to go (pros and cons to any approach). For example, LASSO has the limitation of being a linear method (but it has a nice advantage in being fairly interpretable, compared to other ML approaches). The authors mention statistical power, but they end up running separate tests to assess the 'significance' of the various LASSO coefficients...so it probably ends up being somewhat similar to univariate+correction for power (although you are getting rid of a lot of the colinear stuff with the L1 penalty, so maybe it's a bit better). Anyway, I think the overall approach is fine -- I'd just present it as it stands, without needing to justify it.

Response:

This is an important comment. We agree; while our multivariate analysis using the lasso approach discards a number of collinear features using the L1 penalty, our approach for assessing the significance of lasso coefficients and correcting for multiple hypothesis testing is similar to univariate methods with respect to statistical power. To address this, we have now edited following statements about the statistical power of our approach (**Introduction, page 3**):

“For example, to boost statistical power, most studies have examined interactions between a limited subset of host genes and gut microbes ...” changed to “For example, most studies have examined associations between a limited subset of host genes and gut microbes ...”

We have also removed the following sentence from paper (**Introduction, page 3**):

“This approach may also decrease statistical power to detect biologically meaningful associations due to the large number of statistical tests performed”.

We have modified this sentence as well (**Discussion, page 17**):

“... our approach does not require restricting the data to a predetermined subset of taxa or genes of interest to increase statistical power.” changed to “... our approach does not require restricting the data to a predetermined subset of taxa or genes of interest”.

Reviewer 1, Comment 3

Line 147: Based on my own (anecdotal) experience, I've found that Procrustes can sometimes give unreasonably low and less consistent p-values, when compared against a Mantel test. I think it's fine here, but you may want to double-check against Mantel.

Response:

In addition to Procrustes analysis, we have now included a Mantel test to double-check the correlation between the dissimilarity matrices of host gene expression and gut microbiome data in each disease cohort. We find that the overall pattern of correspondence between paired data across the disease cohorts using the Mantel test remained the same as with Procrustes analysis. We have added the following in the manuscript:

Results, page 5:

“We also used the Mantel test to verify the overall correspondence between paired data for each disease cohort, and found a similar pattern of significance of agreement as Procrustes analysis for CRC (p-value = 0.0026), IBD (p-value = 0.2597), and IBS (p-value = 0.9525, see Methods)”.

Methods, page 26:

“We also applied Mantel test to verify overall correlation between dissimilarity matrices of host gene expression (Aitchison's distance) and gut microbiome abundance (Bray Curtis distance) in each disease cohort using the *vegan* package (version 2.4-5) in R. Significance was tested using 9,999 permutations”.

Reviewer 1, Comment 4

lines 167-171: I have some concerns about how your control vs. case analysis was conducted. From what I read, it seems like you identify your gene-microbe associations within the healthy cohort and within the disease cohort separately, and then exclude associations found in the healthy cohort, correct? My concern arises from the different sizes of these cohorts in IBD and IBS. Your statistical power is changing across these groups, and if you're using some kind of p-value cutoff for picking associations then you'll be comparing statistical apples to oranges. Did you account for this somehow? Coefficients are usually much more robust to differences in sample sizes than p-values (e.g. correlation coefficients). I'm not sure how exactly this relates to LASSO coefficients due to the hyperparameter tuning and regularization steps... But this makes me a tad queasy. If I was doing this in a classic GLM, I'd add an interaction term between the independent variable and disease status and identify significant interaction effects (i.e. association is significant in one population but not in the other, or the association reverses sign).

How flexible is your LASSO framework? Can you add an interaction term and model cases and controls together, rather than splitting them up? I feel like this would be more robust.

Response:

We thank the reviewer for this observation. This is an important issue, and we performed the following analyses to address this:

(i) To identify any potential overlap between case and control associations, after lasso regression and stability selection, we compared all identified case and control associations with non-zero lasso regression coefficients without employing any significance cutoff (p-value or FDR) within the CRC, IBD, and IBS cohorts. Reassuringly, we found no overlapping host gene-microbe associations between cases and controls in any disease cohort. In addition, downsampling the cases to match the size of the corresponding control group also did not yield any overlapping associations. Thus, we conclude that the designation of host gene-microbe associations as specific to cases and controls is robust to the significance cutoff and sample size differences in our study cohort. We have now updated the manuscript to incorporate these analyses in identification of patient-specific associations (**Methods, page 33**):

“Using lasso regression and stability selection, we identified associations for cases and controls with non-zero lasso regression coefficients for each disease cohort. To identify associations that were found only in cases and not in controls within a disease cohort, we checked for any potential overlap between case and control associations, without subsetting associations using any p-value or FDR cutoff. We found no overlapping host gene-microbe associations between cases and controls in any disease cohort, which could be driven by underlying biological differences between case and control conditions within each cohort. In addition, downsampling the cases to match the controls also did not yield any overlapping associations. Thus, we conclude that the designation of host gene-microbe associations as specific to cases and controls is robust to the significance cutoff and sample size differences in our study cohort”.

(ii) To further account for case versus control comparison in the lasso approach by using a method that is more robust to different levels of power due to sample size, we did the following analysis at the pathway level: a) first, from the results of our enrichment analysis, we retained the set of host pathways that were tested for enrichment in both cases and controls within each disease cohort without using any p-value or FDR cutoff, and, b) next, we tested for differential enrichment of these pathways between case and control groups using a comparative log odds-ratio approach^{1,2}. This assessment of differential enrichment is analogous to testing for an interaction at the pathway level, because we are explicitly testing whether the levels of enrichment differ between cases and controls. Moreover, because the associated standard error accounts for the relative number of associated (or “significant”) genes in the case and control cohorts, it is more robust to the different levels of power due to sample size. In the IBD and IBS cohorts, we found no common pathways between both case and control groups in step (a) above, implying that host pathways enriched in cases are indeed case-specific in these cohorts.

In CRC, we found two common host pathways in both cases and controls in step (a) above, however their differential enrichment in step (b) was not significant at FDR < 0.1, implying that they were not necessarily specific to cases. Hence, we filtered out these two pathways from the list of case-specific pathways. Thus, we accounted for case versus control comparison at pathway level. A full description of this analysis has been incorporated in the manuscript (**Methods, pages 29 and 34**):

Methods, page 29:

“Differential enrichment analysis

We performed differential enrichment of pathways in cases versus controls by implementing a comparative log odds-ratio approach in R^{185,186}. To do so, we first computed the z-score for the odds ratio for i -th pathway in cases:

$$z_{i,case} = \log(\delta_i)/SE(\delta_i)$$

where, δ_i is the odds-ratio for i -th pathway in cases, and $SE(\delta_i)$ is the standard error for i -th pathway in cases, which is computed using the four elements, n_1 to n_4 , of the 2x2 contingency table used in the enrichment analysis for the i -th pathway as follows:

$$SE(\delta_i) = \sqrt{1/n_1 + 1/n_2 + 1/n_3 + 1/n_4}$$

Similarly, we computed $z_{i,ctrl}$ for the same pathway in the controls. Next, we compute a comparative log odds-ratio for i -th pathway overlapping between cases and controls as follows:

$$z_{i,case-ctrl} = \frac{\log(\delta_{i,case}) - \log(\delta_{i,ctrl})}{SE(\delta_{i,case,ctrl})}$$

The greater the value of $z_{i,case-ctrl}$, the greater the odds a pathway is differentially enriched in case versus control than by chance. P-values were inferred assuming normal approximations, and corrected for multiple hypothesis testing using Benjamini-Hochberg (FDR) approach”.

Methods, page 34:

“Using enrichment analysis for host genes associated with specific gut microbes, we identified host pathways for cases and controls in each disease cohort. To account for case versus control comparison at the pathway level, we performed the following analysis: 1) first, from the results of our enrichment analysis, we retained the set of host pathways that were tested for enrichment in both cases and controls within each disease

cohort without using any p-value or FDR cutoff, and, 2) next, we tested for differential enrichment of these pathways between case and control groups using a comparative log odds-ratio approach (described above). For IBD and IBS cohorts, we found no pathways enriched in both case and control groups in step 1 above, hence implying that host pathways enriched in cases are indeed case-specific in these cohorts. In CRC, we found two host pathways that were common in both cases and controls in step 1, however they were not differentially enriched in cases versus controls at $FDR < 0.1$ (step 2), implying that they were not necessarily specific to cases. Hence, we filtered out these two pathways from consideration for case-specific pathways”.

(iii) In addition to accounting for overlaps between cases and controls at the association level (as in i) and at the pathway level (as in ii), we also accounted for any overlaps between host genes that were associated with microbes in cases and controls. To do so, we only used case-specific host genes, i.e. host genes that were found to be associated with microbes in cases but not in controls, to perform enrichment analysis to determine case-specific pathways in each disease cohort. We have now updated the manuscript text to incorporate this (**Methods, page 34**):

“In addition to accounting for overlaps between cases and controls at the association level and at the pathway level, we also accounted for any overlaps between host genes that were associated with microbes in cases and controls. We only used case-specific host genes, i.e. host genes that were found to be associated with microbes in cases but not in controls, to perform enrichment analysis to determine case-specific pathways in each disease cohort.”

(iv) We have looked into using an interaction term between the independent variables (i.e. microbial taxa) and disease status in the lasso model. Incorporating such an interaction within the lasso framework is not straightforward, and various approaches have been proposed (for an overview, please see Lim and Hastie et al. ³). However, these approaches do not explicitly allow for different sparsity structure in cases and controls (i.e., where an effect is present for cases but not for controls, or present for controls but not for cases), which is crucial to our interpretation and subsequent analyses. Moreover, this approach is computationally arduous for our implementation of lasso analysis due to the following reason: for each disease cohort, we fit over 12,300 gene-wise lasso models on average, where each model uses the expression for a host gene as response and the abundance of about 200 microbial taxa on average as predictors/independent variables. Adding an interaction term between each predictor (i.e. microbial taxon) and disease status will lead to doubling the number of predictors in each gene’s model, resulting in about 400 predictors per model, and including over $12,300 \times 200 = 2,460,000$ additional terms for assessment of model fit per disease cohort, which is computationally challenging. Hence, we did not include interaction terms in our model. We have now added this in manuscript text to clarify (**Methods, page 33**):

“Another potential approach to identify case-specific associations would be to use an interaction term between the independent variable (i.e. microbial taxa) and disease

status in the lasso model, and determine associations that are significant in cases but not in controls. Incorporating such an interaction within the lasso framework is not straightforward, and various approaches have been proposed (for an overview, please see Lim and Hastie et al. ¹⁹³). However, these approaches do not explicitly allow for different sparsity structure in cases and controls (i.e., where an effect is present for cases but not for controls, or present for controls but not for cases), which is crucial to our interpretation and subsequent analyses. Moreover, this approach is computationally challenging for our implementation for the following reason: for each disease cohort, we fit over 12,300 gene-wise lasso models on average, where each model uses the expression for a host gene as response and the abundance of about 200 microbial taxa on average as predictors/independent variables. Adding an interaction term between each predictor (i.e. microbial taxon) and disease status will lead to doubling the number of predictors in each gene's model, resulting in about 400 predictors per model, and including over $12,300 \times 200 = 2,460,000$ additional terms for assessment of model fit per disease cohort. Hence, we did not include interaction terms in our model".

Reviewer 1, Comment 5

I see that your FDR q-value thresholds vary throughout the manuscript. This makes the interpretation of the results a bit harder. Why not stick to a consistent threshold?

Response:

We agree, and have now updated the manuscript to report all results at a consistent FDR q-value threshold of 0.1. Specifically, we have now updated our results to report host pathways enriched among genes that associate with unique gut microbes at FDR q-value cutoff of 0.1 (instead of the previously used FDR cutoff of 0.2). (**Results, page 10**):

"We identified 18 host pathways that are unique to each disease, including 4 CRC-specific, 9 IBD-specific, and 5 IBS-specific pathways that associate with unique gut bacteria (Figure 3B, Fisher's exact test, Benjamini-Hochberg FDR < 0.1, Supplementary Table S9, see Methods)".

We have also updated **Figure 3B** to visualize host pathways at FDR < 0.1 (**Results, page 9**):

We have also modified the method section and Supplementary Table S9 accordingly (**Methods, page 34**):

“Using this approach, we identified 18 host pathways that are unique to each disease, including 4 CRC-specific, 9 IBD-specific, and 5 IBS-specific pathways that associate with unique gut bacteria (FDR < 0.1, Supplementary Table S9)”.

We have deleted the following sentence (**Methods, previously page 28**):

“Here, we used a more relaxed FDR threshold of 0.2 to present a larger number of biologically relevant host pathways”.

Reviewer 1, Comment 6

lines 398-422: I found this section difficult to read. For good reason -- it's hard to intuitively explain the 'shared microbes' and 'shared genes' stuff. But I think this section could be improved. For example, the statement 'shared host genes' was initially confusing to me -- aren't all host genes shared? Shared microbes made more sense to me because, I thought, we each maybe share ~30% of the species in our guts. However, the things being shared are not the genes or the microbes, but their membership in gene-microbe association pairs. Please try to smooth out the exposition here -- I think it'll really help with the flow of the text.

Response:

We thank the reviewer for pointing this out, and agree that the usage of “shared” terminology is confusing here. We have now replaced occurrences of this term with a more intuitive description throughout this Results section to clarify that we are referring to host genes/gut microbes that are common across host gene-microbe association pairs. Here are our edits for the specific sentences highlighted by the reviewer:

Results, page 14:

Replaced “To elucidate potential host gene-microbe interactions for gut microbes shared between diseases, we visualized ...” with “Next, we visualized the networks of host gene-microbe associations for gut microbes that are associated with host genes in two diseases ...”

“Conversely, to explore how the same host genes may associate with different gut microbes across all diseases, we identified host genes that are associated with gut microbes in at least two diseases, and visualized their networks of association across diseases (**Figure 4B**, lasso regression, FDR < 0.1, Supplementary Table S11). We identified 5 such host genes that associate with 4 gut microbes in CRC, 5 gut microbes in IBS, and 4 gut microbes in IBD ...”

In addition, we made the following edits in the manuscript:

Results, page 13 (Legend for Figure 4):

“**Figure 4.** Disease-specific gut microbe-host gene crosstalk. **A.** Associations for *shared* gut microbes, namely microbes that are associated with host genes in at least two diseases; (*center*) Venn diagram showing overlap between gut microbes associated with host genes in CRC, IBD and IBS, (*counter-clockwise*) networks showing host gene-microbe associations for gut microbes shared across associations in CRC, IBD and IBS (**Network 1**), in CRC and IBS (**Network 2**), in IBD and IBS (**Network 3**), and in CRC and IBD (**Network 4**). **B.** Associations for *shared* host genes, i.e. genes that are associated with microbes in at least two diseases; (*center*) Venn diagram showing

overlap between host genes associated with gut microbes in CRC, IBD, and IBS, (*counter-clockwise*) networks showing host gene-microbe associations for host genes shared across associations in CRC, IBD and IBS (**Network 1**), in CRC and IBS (**Network 2**), in IBD and IBS (**Network 3**), and in CRC and IBD (**Network 4**). Circular nodes represent host genes, triangular nodes represent gut microbes. Colored nodes represent specific diseases (purple: CRC, green: IBD, yellow: IBS), grey nodes represent gut microbes (A) and host genes (B) shared between associations across diseases. Edge color represents positive (blue) or negative (red) association, and edge width represents strength of association (spearman rho). All associations were determined at $FDR < 0.1$.”

Results, page 13:

“To understand how gut microbes may associate with specific host genes across diseases, we explored the overlaps between host gene-microbe associations in CRC, IBD, and IBS (**Figure 4A**, lasso regression, Benjamini-Hochberg $FDR < 0.1$, Supplementary Table S10). We identified *shared* gut microbes, namely gut microbes that associate with host genes in at least two diseases, and visualized their networks of association with host genes across diseases. We found three gut microbes, Peptostreptococcaceae, *Streptococcus*, and *Staphylococcus*, whose abundance is correlated with host gene expression in all three diseases in our study cohorts ...”

Results, page 14:

Replaced “Some notable associations for these shared host genes include host genes and taxa previously implicated in CRC and IBD” with “These host genes and the microbial taxa they associate with have been previously implicated in CRC and IBD”.

Results, page 15:

Replaced “Overall, our analysis shows that shared gut microbial taxa and shared host genes depict disease-specific host-microbe crosstalk...” with “Overall, our analysis shows that gut microbial taxa and host genes that are shared between associations across diseases depict disease-specific host-microbe crosstalk...”

Reviewer 1, Comment 7

line 441: Highlighting the use of the term 'interaction' again (it appears frequently, but making a point here to repeat my prior comment). I think it has a causal implication here that isn't appropriate. Please remove this term throughout.

Response:

We agree, and as described in response to **Comment 1** above, we have replaced occurrences of “interact” and “interaction” with “associate” and “association”, respectively, throughout the manuscript, which resulted in a total of 83 replacements. Our edit for this specific sentence is as follows (**Results, page 15**):

“whereas in IBD, it interacts with Acidaminococcaceae” changed to “whereas in IBD, it associates with Acidaminococcaceae”

Reviewer 1, Comment 8

lines 670-677: You included abundance info summarized at multiple taxonomic levels, all in the same abundance matrix. I see that you CLR transformed these profiles? I'm not sure if this is totally appropriate. These profiles can't really be considered relative abundances anymore, because you're multi-counting reads from taxa (i.e. you have species abundances, but you also have the abundance of the genus and the reads mapping to that species also contribute to the genus abundance). Why not calculate relative abundances for each taxonomic rank (i.e. relative frequencies that all sum to 1.0) and CLR transform each of these ranks separately, and then combine the profiles?

Response:

This is an important point. To clarify, for each disease dataset, we summarized taxa at different taxonomic ranks by keeping read counts (not relative abundance)⁴⁻⁶, concatenated these summarized rank matrices into a combined taxa matrix, and applied CLR transform on the combined matrix to obtain the taxa matrix used in our integration analyses. This approach of applying CLR transformation on the combined taxa matrix results in a matrix that is compositionally coherent and has a uniform transformation across taxa, whereas applying CLR transform on each taxonomic rank separately and then combining them would result in a matrix with multimodal distribution (corresponding to composition at each taxonomic rank) that can potentially bias the variable selection by lasso approach. In addition, we hypothesize that the choice of transformation approach would have little effect on the resulting taxa profiles. To test this hypothesis, we examined the correlation between these two approaches of transformation. We generated the taxa profiles based on the two transformation approaches: (i) CLR transform on combined taxa matrix, and (ii) CLR transform each taxa rank separately and combine them. We found that the profiles are significantly correlated (p-value < 0.05) with an average Pearson's correlation of 0.92, and an average Spearman correlation of 0.87 across samples in a dataset. A new Figure S5 included below shows scatter plots depicting the correlation between the taxa profiles resulting from these two transformation approaches on a few randomly selected samples from the CRC microbiome dataset. This concordance between the taxa profiles resulting from the two transformation approaches implies that the results from our downstream analyses are unlikely to be affected by which of the two transformation approaches is used. We have now updated the manuscript to clarify this as follows (**Methods, page 24**):

“To account for compositionality effects in microbiome datasets, we tested two different approaches for performing centered log ratio (CLR) transformation on our taxonomic data for each disease: i) we concatenated the summarized taxa matrices (count data) into a combined taxa matrix, and then applied CLR transform on the combined matrix, ii) we CLR transformed each taxa rank, and then concatenated them into a combined matrix. We verified whether these two transformation approaches are correlated with each other. To do so, we compared the taxa abundance profiles resulting from the two transformation approaches, and found that the two profiles are significantly correlated (p -value < 0.05) with an average Pearson’s correlation of 0.92, and an average Spearman correlation of 0.87 across samples in a dataset (see Figure S5 for example correlations between the taxa profiles resulting from the two transformations on a few randomly selected samples from the CRC microbiome dataset). This concordance between the taxa profiles resulting from the two transformation approaches implies that the transformation approach is unlikely to impact downstream results. The first approach generates a taxa profile that is compositionally coherent and has a uniform transformation across taxa in a dataset, whereas the second approach results in a taxa profile with multimodal distribution (corresponding to composition at each taxonomic rank) that might bias the variable selection by lasso approach. Hence, we adopted the first approach for transforming our taxonomic data”.

Supplementary Figure S5: Scatterplots showing correlation between taxa profiles generated from two transformation approaches across three random samples from CRC disease cohort. The x-axis shows taxa profile resulting from one approach, where we summarize taxa ranks, combine summarized rank matrices, and CLR transform the combined matrix (CLR_taxa_combined), and the y-axis represents taxa profiles resulting from the second approach, where we summarize taxa ranks, CLR transform each taxa rank, and combine the CLR-transformed taxa ranks (CLR_taxa_ranks).

Reviewer 1, Comment 9

lines 738-743: I think I'm ok with this, but it's pretty ad hoc. And going back to my prior comment, this kind of univariate p-value calculation kinda defeats your prior argument about the power of a multi-variate approach (if by 'power' you are interested in identifying individual 'significant' features).

Response:

We thank the reviewer for this comment, and agree with the general statement. If we understand correctly, the reviewer is referring to our approach of assessing the significance of sparse CCA components. We have used a cross-validation approach to test for significance of each pair of canonical variables (or a component). Witten et al., whose work is the basis of our implementation of sparse CCA analysis, have used a similar approach using training set/test set for assessing the robustness of penalized canonical variates⁷. As mentioned in our response to **Comment 2** above, we agree with the reviewer's point regarding statistical power of identifying individual significant features (please see **Comment 2** above for full details and updates to manuscript text). We want to clarify that in the case of sparse CCA, we are testing significance at component level, i.e. for a group of host genes correlated with a group of gut microbes. We are not testing the significance of the individual features selected, which depends on the level of sparsity penalization. Hence, when a component is selected, we cannot claim that every individual gene/taxa belonging to that component is significant at $FDR < 0.1$. However, the validity of results at the pathway level would be robust to how the gene sets were obtained; for example, if they are all Type I errors, we would not expect significant enrichment of specific pathways. We have now added the following text in the manuscript to clarify this (**Methods, page 28**):

“Note that here we are testing for significance at the component level, i.e. for a group of host genes correlated with a group of gut microbes, rather than the significance of the individual features selected, which depends on the level of sparsity penalization”.

Reviewer 1, Comment 10

line 913: How exactly did you remove gender and disease sub-type associations?

Response:

Since our lasso model includes gender and disease-subtype as covariates along with gut microbiome abundance in the predictor matrix (page 31), our model output includes host gene-gender and host gene – disease-subtype associations, in addition to host gene-microbe associations. We used this output to filter out any significant host gene-gender and host gene – disease-subtype (applicable to IBD and IBS) associations ($FDR < 0.1$) to only retain host gene-microbe associations for each disease cohort. We have modified the text in manuscript to clarify this as follows (**Methods, page 32**):

“We filtered out any significant and stability-selected host gene-gender and host gene – disease-subtype (applicable to IBD and IBS) associations from the output to retain significant and stability-selected host gene-microbe associations at $FDR < 0.1$ ”.

Reviewer 1, Comment 11

line 939: How valuable is it to present more (potentially spurious) features by using a less stringent 0.2 cutoff? Why not just pick a consistent cutoff across the manuscript?

Response:

We thank the reviewer for pointing this out. We now adjusted this to report all results at a consistent FDR q-value cutoff of 0.1 throughout the manuscript. Please see a full description of these changes in our response to **Comment 5**; briefly, we have now updated our results to report host pathways enriched at $FDR < 0.1$ (instead of $FDR < 0.2$). This resulted in updates in **Figure 3B**, and multiple changes in Results and Methods sections. Specifically, we have now deleted the sentence highlighted here by the reviewer in the Methods section (previously line **939**):

“Here, we used a more relaxed FDR threshold of 0.2 to present a larger number of biologically relevant host pathways”.

Reviewer #2 (Remarks to the Author):

Priya et al. present a computational approach for analysing host-gene and microbiome interactions and apply this approach to three datasets (CRC, IBD, IBS). While the underlying idea is interesting, I have several concerns:

Major concerns:

Reviewer 2, Comment 1

Data processing was not done with the same pipeline. Previously generated host transcript read count files and OTU tables were used from the original publications of the data. Studies have shown that the same dataset processed with different bioinformatic pipelines result in very different outcomes. While the authors analyse the datasets separately to avoid batch effects (due to differences in sample collection, DNA extraction, library preparation) this is a bias that could be avoided by reprocessing the data. Also, the published datasets should be briefly described in the methods section with details on sample collection and data processing.

Response:

We thank the reviewer for this comment. We agree that processing by different bioinformatics pipelines can lead to different results; however, it is important to note that our study included data from three different disease cohorts -- data that was generated using different protocols for sample collection, handling, preparation, and sequencing. Previous studies have shown that batch effects introduced due to differences in data generation protocols, including sample collection, storage, DNA/mRNA extraction, PCR amplification, and sequencing cannot be

avoided by adopting a uniform data processing pipeline^{8–10}. Studies have shown that even when using the same data curation pipeline, biases in the data generation pipeline and batch effects still influence the assignment of taxonomic composition and gene expression profiles^{9,11}. Statistical approaches to correct for batch effects have been proposed for gene expression and microbiome datasets, however most of these approaches are relevant to testing differences between cases and controls, and do not apply to integrative analyses^{8,10}. One way to mitigate batch effects across multiple datasets is to use meta-analyses techniques, where each dataset is analysed separately followed by comparison across results from each analysis. While these meta-analysis approaches have disadvantages, like reduced statistical power, they have been extensively used to minimize batch effects when integrating genomic data from multiple studies¹⁰, and have recently been useful in microbiome studies^{12,13}. Therefore, we chose to perform our data analysis within each disease cohort, and compare the results across cohorts at the last analysis step. We have now incorporated these details in the manuscript to clarify our approach further (**Methods, page 25**):

“Integrating host gene expression and gut microbiome data across diseases

Our study includes three different disease cohorts with disparate protocols for sample collection, handling, preparation, and sequencing. Previous studies have shown that differences in data generation protocols, including sample collection, storage, DNA/mRNA extraction, PCR amplification, and sequencing can lead to potential batch effects regardless of the data processing pipeline used^{150–152}. Studies have shown that even when using the same data curation pipeline, biases in the data generation pipeline and batch effects still influence the assignment of taxonomic composition and gene expression profiles^{151,176}. Statistical approaches to correct for batch effects have been proposed for gene expression and microbiome datasets, however most of these approaches are relevant to testing differences between cases and controls, and do not apply to integrative analyses^{150,152}. One way to mitigate batch effects across multiple datasets is to use meta-analyses techniques, where each dataset is analyzed separately followed by comparison across results from each analysis. While these meta-analysis approaches have disadvantages, like reduced statistical power, they have been extensively used to minimize batch effects when integrating genomic data from multiple studies¹⁵², and have recently proven useful in microbiome studies^{154,155}. Therefore, we chose to perform our data analysis within each disease cohort, and compare the results across cohorts at the last analysis step”.

Previous studies have also shown that different clustering approaches, such as operational taxonomic units (OTUs), zero-radius OTUs (zOTU), and amplicon sequence variants (ASVs), and specific pipeline settings have minor influences on taxonomic classification compared to experimental factors such as choice of sequencing primer¹⁴. To check this in our dataset, we re-analyzed a few samples from some of our disease dataset using the DADA2 pipeline, which uses ASV clustering, and compared to the OTU clustering that was used to process the data used in our study. We found that the estimated taxa profiles are correlated between the two approaches at different taxonomic levels. For example, in a few CRC samples, we found that

taxa profiles obtained using OTU clustering and ASV clustering are significantly correlated at different taxonomic levels, including at phylum (Spearman rho = 0.89, p-value = 0.0068), class (Spearman rho = 0.68, p-value = 0.029), and genus (Spearman rho = 0.6, p-value = 0.088) levels. This implies that different processing pipelines may not alter the taxonomic composition drastically to influence the downstream analyses. We have now included the details of this analysis in the manuscript (**Methods, Page 26**), and also added a discussion in the manuscript about the effect of different processing pipelines and how we accounted for potential batch effects across disease cohorts (**Discussion, Page 18**):

Methods, Page 25:

“Previous studies have also shown that different clustering approaches, such as operational taxonomic units (OTUs), zero-radius OTUs (zOTU), and amplicon sequence variants (ASVs), and specific pipeline settings have minor influences on taxonomic classification compared to experimental factors such as choice of sequencing primer¹⁵³. To check this in our dataset, we re-analyzed a few samples from some of our disease dataset using the DADA2 pipeline¹⁷⁷, which uses ASV clustering, and compared to the OTU clustering that was used to process the data used in our study. We found that the estimated taxa profiles are correlated between the two approaches at different taxonomic levels. For example, in a few CRC samples, we found that taxa profiles obtained using OTU clustering and ASV clustering are significantly correlated at different taxonomic levels, including at phylum (Spearman rho = 0.89, p-value = 0.0068), class (Spearman rho = 0.68, p-value = 0.029), and genus (Spearman rho = 0.6, p-value = 0.088) levels. This implies that different processing pipelines may not alter the taxonomic composition drastically to influence the downstream analyses”.

Discussion, Page 18:

“Another caveat of our study is that it includes three different disease cohorts with disparate protocols for sample collection, preparation, sequencing, and data processing. Although this may lead to batch effects (see e.g.^{150–152}), studies have shown that different clustering approaches, such as operational taxonomic units (OTUs), zero-radius OTUs (zOTU), and amplicon sequence variants (ASVs), and specific pipeline settings have minor influences on taxonomic classification compared to experimental factors such as the choice of sequencing primer¹⁵³. Indeed, a re-analysis of representative samples in our dataset using a different pipeline found that the estimated taxonomic profiles are correlated between the two approaches at different taxonomic levels (see Methods). This implies that different processing pipelines may have a minor effect on the taxonomic composition and are unlikely to influence the downstream analyses. In addition, to mitigate overall batch effects across disease cohorts in our study, we adopted a meta-analysis approach, where we performed our integration analysis within each disease cohort separately, and compared the results across cohorts at the last analysis step. While meta-analysis approaches have disadvantages, like reduced statistical power, they have been extensively used to minimize batch effects when

integrating genomic data from multiple studies ¹⁵², and have recently proven useful in microbiome studies ^{154,155}”

As per reviewer’s suggestion, we have now updated the methods section in the manuscript to include additional descriptions for the published datasets used in our study, including details on sample collection and data processing for each dataset (**Methods, pages 21–23**):

Methods, page 21 (CRC samples and data):

“We used 88 pairs of gut microbiome and host gene expression samples from 44 patients, with primary tumor and normal tissue samples taken from each individual. These samples were characterized and described in a previous study ³. Briefly, these de-identified samples were obtained from the University of Minnesota Biological Materials Procurement Network (Bionet). Tissue pairs were resected concurrently, rinsed with sterile water, flash frozen in liquid nitrogen, and characterized by staff pathologists. Detailed cohort characteristics for this dataset are included in Supplementary Table S1”.

Methods, page 22:

“16S rRNA sequencing data. The microbiome dataset used in this study was generated and published previously ³. Briefly, total DNA was isolated from the flash-frozen tissue samples and their associated microbiomes by adapting an established nucleic acid extraction protocol ¹⁵⁶. DNA isolated from colon samples was quantified by quantitative PCR (qPCR), V5-V6 regions of the 16S rRNA gene were PCR amplified and sequenced using the Illumina MiSeq (v3 Kit) with 2 × 250 bp paired-end protocol. The forward and reverse reads were merged and trimmed using USEARCH v7 ¹⁶¹. The merged and filtered reads were used to pick operational taxonomic units (OTUs) with QIIME v.1.7.0¹⁶². We used the unnormalized and unfiltered OTU table in tab-delimited format, representing mucosal microbiome data from 44 tumor and 44 patient-matched colon tissue samples. We describe the steps for preprocessing microbiome data for integration analysis below”.

For the previously unpublished host gene expression data for CRC, we have extensively described the RNA-seq protocol used, including details on mRNA extraction, library preparation, sequencing, alignment, and quality control (see **Methods, page 21**).

Methods, page 22 (IBD samples and data):

“We used previously generated and published host gene expression (RNA-seq) and mucosal gut microbiome (16S rRNA) data for the IBD cohort generated as part of the HMP2 project ^{25,28}. These include data from colonic biopsy samples collected from 78 individuals, including 56 individuals with IBD, and 22 individuals without IBD (“non-IBD” in HMP2). Out of 56 IBD patients, 34 patients had Crohn’s disease (CD) and 22 patients

had ulcerative colitis (UC). Detailed protocol for sample collection and processing is described at <http://ibdmdb.org/protocols>. Briefly, biopsies were collected during the initial screening colonoscopy and stored in RNAlater for molecular data generation (host and microbial, stored at -20°C). DNA and RNA were extracted from RNAlater-preserved biopsies using the AllPrep DNA/RNA Universal Kit from Qiagen. For microbiome profiling, bacterial genomic DNA was extracted from the total mass of the biopsied specimens using the MoBIO PowerLyzer Tissue and Cells DNA isolation kit, 16S rDNA V4 region was PCR amplified and sequenced in the Illumina MiSeq platform using the 2×250 bp paired-end protocol. Read pairs were demultiplexed and merged using USEARCH v7.0.1090 and clustered into OTUs using UPARSE algorithm^{161,163}. For host gene expression data, mRNA was extracted from biopsy samples, followed by RNA-seq library preparation using a variant of Illumina TruSeq Stranded mRNA Sample Preparation Kit. Sequencing was performed according to the manufacturer's protocols using either the HiSeq 2000 or HiSeq 2500 with 101bp paired-end reads. Data was analyzed using the Broad Picard Pipeline.

Detailed cohort characteristics for this dataset are included in Supplementary Table S1. We downloaded metadata, host RNA-seq data, and microbiome data for these samples from <http://ibdmdb.org> in July 2018. We downloaded the unnormalized and unfiltered OTU table and host transcript read counts files in tab-delimited format. We describe the filtering and preprocessing steps for host gene expression and microbiome data below”.

Methods, page 23 (IBS samples and data):

“We used previously generated and published host gene expression (RNA-seq) and mucosal gut microbiome (16S rRNA) data for the IBS cohort⁸. These include data from colonic biopsy samples collected from 42 individuals, including 29 individuals with IBS, and 13 healthy individuals (non-IBS). These samples were collected at Mayo Clinic Rochester, and described in detail by Mars et al.⁸. Briefly, for the microbiome data, DNA was extracted from biopsy sections using the QIAGEN PowerSoil kit (QIAGEN, Germantown, MD, USA). The V4 region of the 16S rRNA gene was amplified, followed by paired-end 2x250 bp sequencing on an Illumina MiSeq. Trimming of adaptors, quality control, and merging of reads was performed using Shi7¹⁶⁴. Amplicon sequences were aligned to the 16S rRNA genes using BURST¹⁶⁵. For host gene expression data, mRNA was extracted from biopsy samples, followed by RNA-Seq library preparation using the Illumina TruSeq RNA Library Prep Kit v2. Sequencing was performed on an Illumina HiSeq-2000 with 101bp paired-end reads. Gene expression counts were obtained using the MAP-RSeq v.2.0.0 that consists of alignment with TopHat 2.0.12 against the human hg19 genome build, and gene counts with the Subread package 1.4.4¹⁶⁶⁻¹⁶⁸.”

Detailed cohort characteristics for this dataset are included in Supplementary Table S1. We obtained the unnormalized and unfiltered OTU table and host transcript read count files in tab-delimited format via personal communication with authors of the paper⁸. For

some individuals, samples were collected at two time points. For these cases, we averaged the gene expression levels and microbiome abundance measurements across the two time points. This is supported by a recent study showing that “omics” methods are more accurate when using averages over multiple sampling time points¹⁶⁹. We describe the filtering and preprocessing steps for host gene expression and microbiome data below”.

Reviewer 2, Comment 2

The authors state that their aim is to facilitate new insights into the molecular mechanisms for different disease and that their analysis presents the power of integrating gut microbiome and host gene expression data to provide disease insights. The authors did a very thorough comparison with other published studies, but which specific novel hypotheses were generated? What type of validation experiments would the authors suggest based on groups of microbes and groups of genes that correlate in abundance/expression? Could a proof-of-principle experiment be added to validate their approach?

Response:

This is an important point -- we agree that this was not described well in the previous version of the manuscript, and have now edited the manuscript to clarify the novel hypotheses and potential validation experiments. Briefly, we believe that many novel hypotheses, in the form of specific host gene-microbe associations, are presented in the results. We have now added the following text to the manuscript to clarify this (**Discussion, page 19**):

“Our study uncovered new insights at the systems level; for example, we found that gut microbes that have been associated with all three diseases, such as *Streptococcus*, associate with different host genes in each disease, suggesting that the same microbial taxon can contribute to different health outcomes through potentially regulating the expression of different host genes in the colon. We also identified numerous specific hypotheses in the form of disease-specific associations; for example, we found that Bacteroidales is associated with host genes *CCR2* and *FPR1* in IL-10 signaling host pathway in colorectal cancer; that Peptostreptococcaceae is associated with *MAPK3* and *VIPR1* that are part of G protein-coupled receptors pathways in inflammatory bowel disease; and that *Bacteroides massiliensis* is associated with the host gene *PLA2G4A*, a member of prostaglandin biosynthesis pathway, in irritable bowel syndrome”.

Future validation experiments using cell cultures and animal models can be designed to test these hypotheses to elucidate the causal role and mechanism of host gene-microbiome interactions. For example, using human cell culture experiments²⁹, one can spike-in specific bacteria of interest and quantify whether change in abundance of said bacteria modulates gene expression in host cells. In addition, *in vivo* studies can be used to assess the impact of specific

bacteria on host gene expression in specific disease contexts³⁰. We have added the following text in the manuscript to specify this (**Discussion, page 18**):

“For example, using human cell culture experiments²², it is possible to measure the direct effect of change in abundance of specific bacteria on host gene regulation. Moreover, *in vivo* studies using animal models of specific diseases can assess the impact of specific bacteria on host gene expression and the effect of this association on host health¹⁴⁹”.

A proof-of-principle experiment is a good idea; however, we believe it is out of scope of our current work. We believe that our work is of importance as it stands, comprising the first systems-level characterization of host gene-microbe associations across gastrointestinal diseases, and providing a machine learning framework that can be utilized by future studies for joint analyses of multi-omic datasets. We expect that future studies, focused on specific diseases, host genes, or bacterial taxa, will offer experimental validation of some of the candidate associations presented here.

Reviewer 2, Comment 3

For the analysis they “considered only associations that were found in patients and not in controls”. What about loss of function in disease?

Response:

We thank the reviewer for pointing this out. If we understand correctly, the reviewer is referring to lasso results, describing associations between specific host genes and microbial taxa. To test the effect of loss-of-function in disease (i.e., associations that are found in controls and not in patients), we performed the following analysis using a two-step approach:

i) First, we identified host pathways that are enriched only in controls, and not in cases at FDR < 0.1. In IBD and IBS, no pathways were found to be enriched in controls at FDR < 0.1, so we could not identify any functional trends in control associations for these two disease cohorts. In CRC, the top 10 control-specific pathways at FDR < 0.1 are related to transcriptional regulation, rRNA expression, DNA methylation, and other such general cellular function categories that did not provide any useful insight on loss of function in disease.

ii) Next, we identified any host pathways that were enriched in both cases and controls, and tested for differential enrichment of those pathways between cases and controls using a comparative log-odds ratio approach¹. The full details of this approach are described in our response to **Reviewer 1, Comment 4**, and updated text in the manuscript in the Methods section, **pages 29 and 34**. As mentioned above, we did not find any host pathways enriched for control associations in IBD and IBS, hence no overlapping pathways were found for cases and controls for these two diseases. In CRC, we found two pathways that overlapped between cases and controls, but the differential enrichment was not statistically significant at FDR < 0.1.

Overall, these analyses did not yield any specific functional trends in the control associations in CRC, IBD, and IBS cohorts. Hence, we only focused on case-specific associations and pathways across diseases in this study. We have now updated the manuscript to incorporate these analyses and findings (**Methods, page 35**):

“To identify any loss of function in disease (i.e., associations that are found in controls and not in cases), we performed analysis at pathway level to determine functional trends in control associations compared to case associations. To do so, we first determined any host pathways that were found enriched only in controls, and not in cases at FDR < 0.1. In IBD and IBS, no pathways were found to be enriched in controls at FDR < 0.1, so we could not identify any functional trends in control associations for these two disease cohorts. In CRC, top 10 control-specific pathways at FDR < 0.1 are related to transcriptional regulation, rRNA expression, DNA methylation, and other such general cellular function categories that did not provide any useful insight on loss of function in disease. Additionally, as mentioned above, we did not find any pathways differentially enriched in controls vs cases in any disease cohort. Hence, given that we did not find any specific functional trends in the control associations in CRC, IBD, and IBS cohorts, we only focused on case-specific associations and pathways across diseases in this study”.

Reviewer 2, Comment 4

The identification of generalisable disease-specific characteristics is challenging on such small datasets with less than 100 participants per disease. The microbiome is highly variable making the replication of disease-associated species even across large cohort studies with hundreds of samples challenging. Hence investigating functional implications of microbial shifts may be a more promising avenue. Conclusions such as “these associations are not found in IBD or IBS, and are unique to CRC” are too general/overstating the findings and it should be clearly stated that this refers to this specific cohort analysis.

Response:

We thank the reviewer for this important point, and agree with the comment about challenges associated with generalizability of disease-specific associations due to variability of microbial composition across disease cohorts. We agree that investigating the functional repertoire of microbial changes would be a promising future direction. We have now modified the statement highlighted by the reviewer as well as other statements throughout the manuscript to clarify that our conclusions are specific to the cohort used in this work:

Discussion, page 16:

“These associations are not found in IBD or IBS, and are unique to CRC in our study cohort”.

We also added a more general statement to the **Discussion**, stating that the results described in the paper refer to this specific cohort analysis (**page 19**):

“We note that it might be challenging to generalize some of the results found here, as microbiome profiles may vary across individuals and diseases. Hence, investigating the functional repertoire of microbial changes, and its association with host genomic data, would be a promising future direction”.

In addition, we added text in other locations clarifying this:

Discussion, page 16:

“For example, we found that the expression of host gene *PINK1*, which regulates the PI3-kinase/AKT signaling pathway ¹²⁵, is associated with the abundance of *Collinsella* in CRC, Peptostreptococcaceae in IBD, and *Blautia* in IBS in our study”.

Results, page 7:

“In addition, we identified 102 disease-specific host pathways that are associated with gut microbes, including 52 CRC-specific, 25 IBD-specific, and 25 IBS-specific pathways in our study cohorts”.

Results, page 13:

“*Streptococcus* also shows a disease-specific pattern of association with host gene expression in our study cohort”.

Reviewer 2, Comment 5

The authors describe the use of supercomputing nodes for their analysis. Did the analysis of these relatively small dataset require that much compute power? How will this be scaled to hundreds of samples? Or is the purely a feature provided as part of their method? Some information on how long the cohort analysis took should be provided.

Response:

Our parallel processing framework implemented on supercomputing nodes is built to overcome computational bottlenecks posed by high dimensionality, i.e. the large number of features in host transcriptome and microbiome data. We implemented a parallel framework to build gene-wise lasso models for over 12,000 host genes, where each model includes hundreds of microbial taxa as covariates. Thus, usage of supercomputing nodes here is mainly dictated by a large number of features, not by the number of samples in such datasets (although this framework can also improve scalability of models with larger sample sizes). We have clarified

this in the manuscript, and also reported the time taken for analysis across the cohorts (**Methods, page 32**):

“We leveraged supercomputing nodes to implement a parallel processing framework to allow scalable computation for the high dimensional datasets included in this analysis. We implemented a parallel framework for executing the gene-wise lasso analysis, where we parallelized execution of lasso models on host genes across multiple nodes and cores on a compute cluster from Minnesota Supercomputing Institute. Our framework scalably computes gene-wise models for over 12,000 host genes, where each model includes hundreds of microbial taxa as covariates. We used job arrays to parallelize our analysis on multiple nodes on the cluster. Additionally, we used the R packages *doParallel* (version 1.0.15) and *foreach* (version 1.4.7) to run parallel processes on multiple cores of each compute node. Our parallel processing framework using 5 compute nodes took an average of 5 hours 30 minutes per disease cohort”.

Reviewer 2, Comment 6

It's great that the authors made the code available on github, however, a description of the different scripts and a tutorial is missing.

Response:

We thank the reviewer for this suggestion. We have now added a tutorial for our pipeline in the github repository that can be accessed here:

https://github.com/blekhmanlab/host_gene_microbiome_interactions/tree/main/Tutorial. This tutorial includes code that can be used to reproduce the integration analysis in our paper, including the sparse CCA and lasso models, and will be useful for researchers interested in running the integrative machine learning framework on their data. We have also added descriptions of the different scripts.

Reviewer 2, Comment 7

What is their theory behind the disease-specific host-microbe crosstalk? Difference in host genetics?

Response:

This is an important point. A potential theory for the disease-specific pattern of host-microbe crosstalk is that gut microbes, either through direct interaction with colonic epithelium, or through indirect interaction (e.g. via production of specific metabolites), could regulate host gene expression differently in specific disease contexts. Host genetics, diet, pharmaceuticals, inter-microbial interactions, and microbial interactions with the host immune system could all potentially play a role in shaping this disease-specific host gene-microbe crosstalk. We have now added the following text to the manuscript to clarify this (**Discussion, page 16**):

“The disease-specific pattern of host gene-microbe crosstalk suggests that gut microbes, either through direct interaction with host cells, or through indirect interaction (e.g. via production of specific metabolites), may regulate host gene expression differently in specific disease contexts. Host genetics, diet, pharmaceuticals, inter-microbial interactions, and microbial interactions with the host immune system could all potentially play a role in shaping this disease-specific host gene-microbe crosstalk”.

Minor concerns:

Reviewer 2, Comment 8

The authors should clarify in the manuscript (not just the methods section) that most of the presented data is published data and also compare their analysis to the original findings.

Response:

We thank the reviewer for pointing this out, and we agree that this should have been more explicitly mentioned. Indeed, all datasets, except the host gene expression data for colorectal cancer, have been published. In addition to citing the original studies when describing the study cohorts, (**Results, page 4, lines 136–140**), we now also included a sentence to clarify this further (**Results, page 4**):

“All datasets, except the host gene expression data for CRC, have been previously published as individual studies^{3,8,25,28}. Our study performed an integrative analysis to identify host gene-microbiome associations across these datasets using a novel machine learning-based framework”.

To compare our analysis and findings with those described in the previous studies that published these datasets, we have added the following statements to the manuscript (**Discussion, page 17**):

“While the disease cohorts used in our study have been previously published^{3,8,25,28}, to the best of our knowledge, our study is the first to perform a comprehensive characterization of host gene-microbiome crosstalk within and across these disease cohorts. In the previous study that described the CRC cohort, Burns et al. characterized the tumor-associated microbiome, and how it varies in composition compared to the microbiome of adjacent matched normal colon tissue³. For example, they reported loss in abundance of multiple taxa within the order Bacteroidales in tumor-associated microbiota compared to normal samples. Here, we found that Bacteroidales is associated with host genes *CCR2* and *FPR1*, which are part of the tumor-associated IL-10 signaling pathway. The previous study that described the IBD cohort compared IBD versus non-IBD samples and found differences in transcriptional activity and abundance among taxa belonging to class Clostridia, and dysregulation of immune-related host

pathways in disease state ^{25,28}. Our integrative analysis for the same cohort revealed associations between immunoinflammatory pathways and members within class Clostridia, including Peptostreptococcaceae and *Clostridium sensu stricto 1*. In the original study describing the IBS cohort, Mars et al. found overrepresentation of *Streptococcus* species in patients with IBS compared to healthy individuals, and identified associations between fecal microbes, such as Peptostreptococcaceae, with host genes implicated in peptidoglycan binding ⁸. Our analyses revealed several novel associations between closely interacting, tissue-adherent microbiome and host genes and pathways in IBS, including associations between *Streptococcus* and host genes that modulate macrophage inflammatory response, and between Peptostreptococcaceae and host pathways that regulate intestinal homeostasis and inflammation”.

Reviewer 2, Comment 9

Different FDR threshold were used at various places.

Response:

We thank the reviewer for pointing this out. We have now fixed this to report all results at a consistent FDR threshold of 0.1 throughout the manuscript. This issue was also raised by Reviewer 1 - please see our full description of the changes above in response to **Reviewer 1, Comment 5**. Briefly, we have now updated our analysis to use a consistent cutoff of FDR = 0.1 throughout the manuscript. This has resulted in the multiple modifications throughout the manuscript, including an updated Figure 3B, and several edits in Methods and Result sections. We highlight a few changes below; please refer to the response to **Reviewer 1, Comment 5** to see all the changes.

Results, page 10:

“We identified 18 host pathways that are unique to each disease, including 4 CRC-specific, 9 IBD-specific, and 5 IBS-specific pathways that associate with unique gut bacteria (Figure 3B, Fisher’s exact test, Benjamini-Hochberg FDR < 0.1, Supplementary Table S9, see Methods)”.

We have modified the method section and Supplementary Table S9 accordingly (**Methods, page 34**):

“Using this approach, we identified 18 host pathways that are unique to each disease, including 4 CRC-specific, 9 IBD-specific, and 5 IBS-specific pathways that associate with unique gut bacteria (FDR < 0.1, Supplementary Table S9)”.

We have deleted the following sentence from the manuscript (**Methods, previously page 28**):

“Here, we used a more relaxed FDR threshold of 0.2 to present a larger number of biologically relevant host pathways”.

Reviewer 2, Comment 10

Why did the IBD dataset only have 121 taxa? Both the CRC and IBS dataset had twice as many taxa.

Response:

We thank the reviewer for pointing this out. Indeed, we found that the number of unique taxonomic groups found in the IBD dataset was 40% of those found in CRC and IBS datasets. This implies that the IBD dataset has lower bacterial diversity than the other two disease datasets. Previous studies have reported reduced bacterial diversity in gut mucosal microbiome in patients with IBD compared to individuals without this disease^{31–33}, including the HMP2 study that generated and described the IBD dataset used in our study^{19,20}. On the other hand, previous studies that generated and characterized the CRC and IBS datasets used in our work have reported increased microbial diversity in these conditions^{15,22}. We have now added the following to the manuscript to clarify this (**Methods, page 25**):

“We observed that the number of unique taxonomic groups found in the IBD dataset was 40% of those found in CRC and IBS datasets, thus implying that the IBD dataset has lower bacterial diversity than other two disease datasets. This observation is consistent with previous studies that have shown reduced bacterial diversity in gut mucosal microbiome in patients with IBD compared to individuals without this disease^{173–175}, including the HMP2 study that generated and described the IBD dataset used in our study^{25,28}. In addition, previous studies that generated and characterized the CRC and IBS datasets used in our work have reported increased microbial diversity in these conditions^{3,8}”.

Reviewer #3 (Remarks to the Author):

In this manuscript by Priya et al., the relationship between microbiome composition and host gene expression is examined in colonic mucosal biopsy samples. In particular, case control studies of three separate diseases were conducted (colorectal cancer, inflammatory bowel disease, and irritable bowel syndrome), with RNA-seq being used to measure host gene expression and 16S rRNA gene sequencing being used to measure microbiome taxonomic composition. Sparse canonical correlation analysis and clr-lasso were applied to the data to identify microbe-host gene associations, and those were compared across diseases. Overall, this is a really impressive study: methods are suitable and very well described, results are informative, and the manuscript is very well-written, with many connections to previous literature to put results in context. A study like this can serve as a framework for others interested in combining these two types of datasets while conserving power to detect associations.

Minor comments:

Reviewer 3, Comment 1

1. Results are difficult to interpret when multiple levels of bacterial taxonomy are analyzed at the same time, rather than level by level. For example, in Figure 2, both Micrococcaceae and Micrococcales are highlighted under the IBD RAC1 pathway panel in B. Is this largely the same signal, as these groups are nested, or are there distinct signals at different taxonomic levels? It may be more interpretable for the reader to focus on one level for the main text (maybe OTU or genus level), with other levels falling to the supplement.

Response:

We thank the reviewer for pointing this out. While our approach identifies distinct signals at different taxonomic levels, we agree that highlighting multiple levels of taxa at the same time can be difficult to interpret. To address this, we have now completely updated **Figures 2B, 2C, and 2D** to visually distinguish microbial taxa that are at different taxonomic levels. Specifically, taxa that belong to the same taxonomic order are now displayed as overlapping clusters in the image (see below). For example, in the CRC component for RAC1 pathway (Figure 2B), *Streptococcus*, *Leuconostoc*, and *Lactococcus*, which belong to a common taxonomic order, Lactobacillales, are shown in an overlapping cluster. Similarly, in the IBD component for RAC1 pathway, Micrococcales and Micrococcaceae, which belong to the clade Micrococcales, are shown clustered together (**Figure 2, Results, page 6**).

B

A shared host pathway

○ Host gene

△ Gut microbe

C

Disease-specific host pathways

D

We have also updated the legend for Figure 2 to indicate this modification (**Results, page 7**):

“Microbial taxa belonging to a common taxonomic order are shown as overlapping triangles”.

To explain our reasoning for performing our analysis by combining all taxonomic levels: we believe this approach allows us to identify signals found at any taxonomic level and avoid missing potentially relevant associations. At the same time, given the large number of features in high-dimensional datasets like gene expression and microbiome data, this approach circumvents the computationally time-intensive analysis that would be required if each taxonomic level was analyzed separately. Using this approach, we found that our penalized approaches using sparse CCA and lasso do not select all taxonomic levels from a given clade, rather they identify one or more microbial taxa for which abundance is associated with expression of host gene(s), thus depicting that our approach captures distinct signals at different taxonomic ranks. In theory, if we repeated our integration analyses separately at different taxonomic levels, we would likely identify the same set of taxa as our current approach that uses a combined taxonomic profile. Hence, the results would be effectively the same whether we analyze each taxonomic level separately or use multiple levels together. To clarify this in the paper, we added the following to the **Methods (page 27)**:

“We chose to combine all taxonomic levels in a single model, rather than run a separate model for each taxonomic level. Given the large number of features in high-dimensional datasets like gene expression and microbiome data, this approach circumvents the computationally time-intensive analysis that would be required if each taxonomic level was analyzed separately. Our approach allows us to identify signals found at any taxonomic level and avoid missing potentially relevant associations”.

Reviewer 3, Comment 2

2. page 4, line 136/methods: More description of the case/control design for each of the three disease types listed would be beneficial for the reader. What were the criteria used to “match” case with control? For some of the diseases (IBD and IBS), patients and controls are described as “pairs”, but there are unequal case and control sample sizes.

Response:

We agree that this description was lacking, and have now edited the text to better describe the case/control design. To clarify, we use the term “pairs” to refer to a set of two samples -- a microbiome sample and host gene expression sample -- obtained from the same individual. Our study includes 208 such pairs of microbiome and host gene expression samples (416 samples in total). These 208 paired microbiome and host gene expression samples include 88 pairs of samples in CRC, 78 pairs of samples in IBD, and 42 pairs of samples in IBS. Our usage of the term “match” is specifically with respect to the CRC cohort, where we have patient-matched tumor and normal biopsies (i.e., matched case (tumor) and control samples from the same

individual). Such patient-matched samples only exist for our CRC cohort, and not for IBD and IBS, where case and control samples were collected from different individuals. We have now modified the statement in text to clarify this and provide more description of case/control design for each disease cohort (**Results, page 4**):

“For each individual in our study, we obtained a pair of samples -- a microbiome sample and a host gene expression sample. In total, across the three disease cohorts, our study included 208 such pairs of microbiome and host gene expression samples (416 samples in total). These 208 paired microbiome and host gene expression samples include 88 pairs of samples in the CRC cohort (44 tumor and 44 patient-matched normal)³, 78 pairs of samples in the IBD cohort (56 patients and 22 controls)^{25,28}, and 42 pairs of samples in the IBS cohort (29 patients and 13 controls; see Supplementary Table S1)⁸”.

We have also edited the sentence in methods section to clarify this (**Methods, page 21**):

“We used 88 pairs of gut microbiome and host gene expression samples from 44 patients, with primary tumor and normal tissue samples taken from each individual”.

We have replaced another occurrence of the term “matched” to avoid potential confusion (**Results, page 5**):

Line 166: “matched host gene expression data and gut microbiome data” replaced with “paired host gene expression data and gut microbiome data”

Reviewer 3, Comment 3

3. Are the sample characteristics largely the same across the disease cohorts, or are there differences that might be confounded with “disease-specific configurations” of the microbiome and host gene expression? For example, if there were age differences, sex biases, different sampling locations, or treatments that might also influence gene expression or the microbiome, those would be worth clarifying for the reader and/or mentioning as a caveat to interpretation of the study.

Response:

This is an important point. We agree that some factors can be potential confounders that might influence host gene expression and/or the microbiome composition. In our lasso regression analysis, we accounted for some potential confounders, including sex and disease-subtypes, by including them as covariates in the model (**Methods, page 30**). However, some important potential confounding factors were not available for the samples used in our study; these include, for example, medication history and diet. To clarify this, as suggested by the reviewer, we now include this in the discussion of potential limitations of our study (**Discussion, page 19**):

“Lastly, there are several host and environmental variables that could potentially influence the microbiome and/or host gene expression, including age differences, sampling locations, diet, treatment, and medication history, which are not available across our disease cohorts. Thus, these factors are potential confounders that might influence our results”.

Reviewer 3, Comment 4

4. Related to sample sizes mentioned above, the three diseases have different sample sizes, which seem to match with the number of associations detected in each individual disease and the overlaps between them (CRC and IBD have the most samples and the most identified associations (including overlap), while IBS has fewer (including fewer overlaps). Some discussion of this effect of sample size would be useful for the reader, as well as if the unbalanced case/control designs has any effect on the power of applying a lasso.

Response:

We thank the reviewer for this important observation. Indeed, the number of host gene-microbe associations identified in each disease cohort is correlated with the number of samples in that cohort, which also influences the number of overlaps identified between associations across disease cohorts. One option to test for the effect of sample size is downsampling; however, since we found no overlapping associations across diseases, downsampling cohorts to a common sample size did not change the outcome of overlapping associations across cohorts. We have now included these details in the manuscript (**Methods, page 37**):

“For each disease cohort, we found that the number of host gene-microbe associations identified is correlated with the number of samples in that cohort. This also influences the number of overlaps identified across disease cohorts. One option to test for the effect of sample size is downsampling; however, since we found no overlapping associations across diseases, downsampling cohorts to a common sample size did not change the outcome of overlapping associations across diseases in our study cohort”.

To address the unevenness in case and control sample size in our study cohorts, an issue that was also raised by Reviewer 1, we have now included multiple new analyses and modifications in the manuscript. These new analyses are described in detail in response to **Reviewer 1, Comment 4**, and edits in the manuscript are on **pages 29, 33, and 34**. Briefly, to identify associations that were found only in cases and not in controls within a disease cohort, we first checked for any potential overlap between case and control host gene-microbe associations without using any p-value or FDR cutoff. We found no overlapping host gene-microbe associations between cases and controls in CRC, IBD, and IBS cohorts. Next, we accounted for this imbalance between cases and controls at the pathway level by applying a differential enrichment analysis that is more robust to the different power due to sample size. We found that most host pathways are specific to case and control groups within each disease cohort. Please refer to the response to **Reviewer 1, Comment 4** to see full details on analyses and

modifications to the text in the manuscript to address this. In addition, we have also added a discussion in the manuscript about the effect of different sample sizes across different cohorts and uneven case/control groups within each cohort (**Discussion, page 18**).

“Another potential issue is the difference in sample size across disease cohorts and between the case and control groups within each cohort. These sample size differences can result in differences in statistical power when applying lasso regression. This can impact the number of host gene-microbe associations identified in each disease group, and the number of overlapping associations identified across cohorts. We attempted to minimize this effect by applying a differential enrichment analysis that is more robust to different levels of statistical power due to sample size (see Methods)”.

Reviewer 3, Comment 5

5. In addition to sample size, the number of detected human genes and taxa differ between disease sets. For the comparisons of overlapping vs. disease specific taxa/genes, were “core” sets of taxa and gene examined that were held constant between disease set?

Response:

This is an important point. To examine this issue further, we have now added to the manuscript extensive analyses that explore whether it is possible to use “core” sets of taxa and genes in our integrative analysis. The full details of the analysis and additions to the manuscript are described below; in brief, our results indicate that although many host genes are expressed in all three disease cohorts, microbial taxa are mostly unique to each disease cohort. More specifically, we found that approximately 80% of host genes are expressed in all disease cohorts, while the remaining 20% of host genes are expressed in one or two of the disease cohorts. However, we found that only about 25% of gut microbial taxa on average are found in all three, thus indicating that a majority of taxa are specific to each disease cohort. This is consistent with previous research that has shown dissimilarity between disease-associated microbial taxa^{12,34}. Given these patterns, using “core” sets of taxa and host genes that are found in all disease cohorts may lead to missing a large number of disease-specific host gene-taxa associations. Therefore, we maintained our approach of using all the host genes and taxa that were identified after preprocessing the input dataset, which allowed us to identify disease-specific as well as overlapping host gene/taxa between associations across diseases. We have added to the manuscript a discussion of the rationale behind this approach, as well as full details of this analysis:

Overlap analysis and corresponding updates to the manuscript:

i) To examine common and distinct features in input datasets, we considered the host genes and taxa that were used as input to the lasso integration pipeline, and calculated the overlap in these host genes and taxa across cohorts. Tables S13 and S14 included below show pairwise overlaps for input host genes and input taxa across diseases, respectively. We computed the

pairwise overlap as an overlap coefficient, which is a measure of similarity between two sets, and is defined here as the number of common genes (or taxa) between the two disease datasets divided by the number of genes (or taxa) in the dataset with fewer genes (or taxa). Overall, we found that an average of 84% of host genes are commonly expressed between diseases (Table S13), while the remaining 16% of host genes might be expressed in only one disease cohort, but not in the other two. These disease-specific genes in the input datasets might have disease-specific association patterns with gut microbes, hence we used all the host genes identified after preprocessing for determining associations in each disease cohort.

On the other hand, only about 37% of input gut microbial taxa overlapped between diseases (Table S14), thus indicating that a majority of taxa were specific to each disease cohort. This is consistent with previous research that have shown high dissimilarity between disease-associated microbial communities^{12,34}. Similar to the above, we retained all gut microbial taxa identified after preprocessing and filtering in each disease cohort to allow identification of associations between gut microbes and host genes that might be specific to a given disease.

Supplementary Table S13: Overlap between input sets of host genes across CRC, IBD, and IBS. Each cell denotes an overlap coefficient between sets of genes from two diseases.

	CRC	IBD	IBS
CRC	1	0.8317063	0.8001448
IBD	0.8317063	1	0.8739257
IBS	0.8001448	0.8739257	1

Supplementary Table S14: Overlap between input sets of gut microbial taxa across CRC, IBD, and IBS. Each cell denotes an overlap coefficient between sets of taxa from two diseases.

	CRC	IBD	IBS
CRC	1	0.4214876	0.2340426
IBD	0.4214876	1	0.446281
IBS	0.2340426	0.446281	1

ii) Next, we calculated the overlap between the set of host genes found associated with gut microbes only in one of the disease cohorts (i.e. disease-specific genes in our identified associations), and the set of host genes used as input in other two diseases (Table S15). Similarly, we calculated the overlap between gut microbial taxa found associated with host genes only in one disease (i.e. disease-specific taxa in our identified associations), and the set of taxa used as input in other two diseases (Table S16). Overall, we found similar trends as described above. We found that an average of 80% of disease-specific host genes in the identified associations were also included as input in the other two diseases (Table S15). The

remaining 20% of disease-specific host genes were likely not expressed in the other two conditions.

On the contrary, we found that, on average, only 12% of disease-specific taxa in the identified associations were included as input taxa in the other two diseases (Table S16). This is in line with our observation of disease-specific trends for gut microbial taxa used as input to the integration pipeline, as described above.

Supplementary Table S15: Overlap between genes found associated with microbes in specific disease (i.e. disease-specific genes in output; rows) and input sets of host genes in other two diseases (i.e. input genes; columns). Each cell denotes an overlap coefficient between sets of genes from two diseases.

		Input host genes		
		CRC	IBD	IBS
Disease-specific host genes in output associations	CRC	1	0.7871363	0.7656968
	IBD	0.8238434	1	0.8656584
	IBS	0.7733333	0.7946667	1

Supplementary Table S16: Overlap between microbial taxa found associated with host genes in specific disease (i.e. disease-specific taxa in output; rows) and input sets of taxa in other two diseases (i.e. input taxa; columns). Each cell denotes an overlap coefficient between sets of taxa from two diseases.

		Input taxa		
		CRC	IBD	IBS
Disease-specific taxa in output associations	CRC	1	0.09615385	0.1057692
	IBD	0.1666667	1	0.1904762
	IBS	0.1011236	0.06741573	1

In addition to including the above supplementary tables in the manuscript, we have also added the following text to Methods section to add clarification regarding overlapping and disease-specific features across diseases (**Methods, page 35**):

“Given the difference between number of host genes and gut microbial taxa identified across diseases, we first determined overlaps in host genes and microbial taxa in input datasets and gene-taxa associations identified across disease cohorts. To do this, we used a two-step process:

i) To examine common and distinct features in input datasets, we considered the host genes and taxa that were used as input to the lasso integration pipeline, and calculated the overlap in these host genes and taxa across cohorts (Supplementary Tables S13 and S14). We computed the pairwise overlap as an overlap coefficient, which is a measure of similarity between two sets, and is defined here as the number of common genes (or taxa) between the two disease datasets divided by the number of genes (or taxa) in the dataset with fewer genes (or taxa). Overall, we found that an average of 84% of host genes are commonly expressed between diseases (Supplementary Table S13), while the remaining 16% of host genes might be expressed in one disease cohort, but not in the other two. On the other hand, only about 37% of input gut microbial taxa overlapped between diseases (Supplementary Table S14), thus indicating that a majority of taxa were specific to each disease cohort. This is consistent with previous research that have shown high dissimilarity between disease-associated microbial communities^{154,194}.

ii) Next, we calculated the overlap between the set of host genes found associated with gut microbes only in one of the disease cohorts (i.e. disease-specific genes in our identified associations), and the set of host genes used as input in other two diseases (Supplementary Table S15). Similarly, we calculated the overlap between gut microbial taxa found associated with host genes only in one disease (i.e. disease-specific taxa in our identified associations), and the set of taxa used as input in other two diseases (Supplementary Table S16). Overall, we found similar trends as described above. We found that an average of 80% of disease-specific host genes in the identified associations were also included as input in the other two diseases (Supplementary Table S15). The remaining 20% of disease-specific host genes were likely not expressed in the other two conditions. On the contrary, we found that, on average, only 12% of disease-specific taxa in the identified associations were included as input taxa in the other two diseases (Supplementary Table S16). This is in line with our observation of disease-specific trends for gut microbial taxa used as input to the integration pipeline, as described above.

Given these patterns, we considered all the host genes and taxa that were identified after preprocessing the input dataset, which allowed us to identify disease-specific as well as overlapping host gene/taxa between associations across diseases”.

Reviewer 3, Comment 6

6. p8, line 264: I suggest using “associate” rather than “interact”, as “interact” could be interpreted by some as a direct, physical interaction (which it might not be).

Response:

We thank the reviewer for this suggestion. We agree that the term “interact” or “interaction” can be misleading, and hence we have replaced the usage of these terms with “associate” and “association”, respectively, throughout the manuscript. As mentioned previously in responses to **Reviewer 1, Comment 1**, and **Reviewer 1, Comment 7**, this resulted in a total of **83 changes** throughout the manuscript. Our edit for this specific sentence is as follows (**Results, page 9**):

“Specific gut microbes interact with individual host genes ...” changed to “Specific gut microbes associate with individual host genes ...”

Reviewer 3, Comment 7

7. p28: How many host pathways were filtered out for being too small, too large, or having not many genes that overlap? Are there potential pathways of interest that aren't being examined here, but could be interesting?

Response:

This is an important point. To avoid pathways that were too large to provide any specific biological insights or too small to provide adequate statistical power, we excluded from our analysis any pathway that included more than 85 genes, fewer than 10 genes, or fewer than 5 genes that overlapped between the pathway and the list of genes identified by our analysis. Out of 1881 host pathways, this criteria filtered out 297 pathways for being too large, 299 pathways for being too small, and an average of 1186 pathways for not having sufficient overlap with the genes of interest in each disease cohort. This resulted in an average of 99 pathways that were tested for enrichment in each disease cohort. We are aware that this approach may filter out some potential pathways of interest from our analysis. However, even if included, these pathways are unlikely to yield significant associations due to lack of statistical power. Thus, we only examine pathways for which associations can be detected as described above. We have now added these details to the manuscript (**Methods, page 33**):

“To avoid pathways that were too large to provide any specific biological insights or too small to provide adequate statistical power, we excluded any pathways from our analysis with more than 85 genes, fewer than 10 genes, or fewer than 5 genes that overlapped between the pathway and the genes of interest. Out of 1881 host pathways, these criteria filtered out 297 pathways for being too large, 299 pathways for being too small, and an average of 1186 pathways for not having sufficient overlap with the genes of interest, resulting in an average of 99 pathways that were tested for enrichment in each disease cohort. We are aware that this approach may filter out some potential pathways of interest from our analysis. However, even if included, these pathways are unlikely to yield significant associations due to lack of statistical power”.

Reviewer 3, Comment 8

8. Figure 2/p50: Overall, I find the visualizations extremely effective for this manuscript, especially considering the many dimensions this data is explored in. The one exception is with the sCCA result visualizations in parts B/C of Figure 2 and the supplemental figure on page 50. It's hard to tell which genes are shared across disease subtypes, because host genes are listed in a different orientation in each subtype. I suggest having the entire host pathway illustrated (maybe in gray) and then coloring in the significant host genes within disease-subtype. That would preserve the order, allowing for better highlighting of the similarities and differences for the reader.

Response:

We thank the reviewer for their positive feedback on the visualizations in our paper. We agree with the suggestion regarding the visualization in **Figure 2** to distinguish host genes that are common or distinct across diseases, and have now updated **figures 2B and 2C** to depict this. In our updated figure, we show genes that are common between pathways or components across at least two diseases in grey, and the disease-specific genes are shown in the color corresponding to the disease (see **Figures 2B and 2D** below). In Figure 2C, we focused on three separate, disease-specific host pathways (i.e. host pathways for which gene expression correlates with gut microbes in only one of the disease cohorts), where by definition, the top 10 genes representing each disease-specific pathway do not have any overlaps, and hence are depicted in disease-specific colors. We have edited the legend for Figure 2 to clarify the updates to the figure (**Results, page 7**):

“Genes that are common between pathways or components across at least two disease cohorts are shown in grey”.

References:

1. Waardenberg, A. J., Basset, S. D., Bouveret, R. & Harvey, R. P. CompGO: an R package for comparing and visualizing Gene Ontology enrichment differences between DNA binding experiments. *BMC Bioinformatics* **16**, 275 (2015).
2. Morris, J. A. & Gardner, M. J. Calculating confidence intervals for relative risks (odds ratios) and standardised ratios and rates. *Br. Med. J.* **296**, 1313–1316 (1988).
3. Lim, M. & Hastie, T. Learning interactions via hierarchical group-lasso regularization. *J. Comput. Graph. Stat.* **24**, 627–654 (2015).
4. Tsilimigras, M. C. B. & Fodor, A. A. Compositional data analysis of the microbiome: fundamentals, tools, and challenges. *Ann. Epidemiol.* **26**, 330–335 (2016).
5. Kaul, A., Mandal, S., Davidov, O. & Peddada, S. D. Analysis of Microbiome Data in the Presence of Excess Zeros. *Front. Microbiol.* **8**, 2114 (2017).
6. Gloor, G. B., Macklaim, J. M., Pawlowsky-Glahn, V. & Egozcue, J. J. Microbiome Datasets Are Compositional: And This Is Not Optional. *Front. Microbiol.* **8**, 2224 (2017).
7. Witten, D. M., Tibshirani, R. & Hastie, T. A penalized matrix decomposition, with applications to sparse principal components and canonical correlation analysis. *Biostatistics* **10**, 515–534 (2009).
8. Wang, Y. & LêCao, K.-A. Managing batch effects in microbiome data. *Brief. Bioinform.* **21**, 1954–1970 (2020).
9. Schloss, P. D., Gevers, D. & Westcott, S. L. Reducing the effects of PCR amplification and sequencing artifacts on 16S rRNA-based studies. *PLoS One* **6**, e27310 (2011).
10. Gibbons, S. M., Duvall, C. & Alm, E. J. Correcting for batch effects in case-control microbiome studies. *PLoS Comput. Biol.* **14**, e1006102 (2018).
11. Leek, J. T. *et al.* Tackling the widespread and critical impact of batch effects in high-throughput data. *Nat. Rev. Genet.* **11**, 733–739 (2010).
12. Tierney, B. T., Tan, Y., Kostic, A. D. & Patel, C. J. Gene-level metagenomic architectures

- across diseases yield high-resolution microbiome diagnostic indicators. *Nat. Commun.* **12**, 2907 (2021).
13. Duvallet, C., Gibbons, S. M., Gurry, T., Irizarry, R. A. & Alm, E. J. Meta-analysis of gut microbiome studies identifies disease-specific and shared responses. *Nat. Commun.* **8**, 1784 (2017).
 14. Abellan-Schneyder, I. *et al.* Primer, Pipelines, Parameters: Issues in 16S rRNA Gene Sequencing. *mSphere* **6**, (2021).
 15. Burns, M. B., Lynch, J., Starr, T. K., Knights, D. & Blekhman, R. Virulence genes are a signature of the microbiome in the colorectal tumor microenvironment. *Genome Med.* **7**, 55 (2015).
 16. Burns, M. B. *et al.* APOBEC3B is an enzymatic source of mutation in breast cancer. *Nature* **494**, 366–370 (2013).
 17. Edgar, R. C. Search and clustering orders of magnitude faster than BLAST. *Bioinformatics* **26**, 2460–2461 (2010).
 18. Caporaso, J. G. *et al.* QIIME allows analysis of high-throughput community sequencing data. *Nat. Methods* **7**, 335–336 (2010).
 19. Lloyd-Price, J. *et al.* Multi-omics of the gut microbial ecosystem in inflammatory bowel diseases. *Nature* **569**, 655–662 (2019).
 20. Integrative HMP (iHMP) Research Network Consortium. The Integrative Human Microbiome Project. *Nature* **569**, 641–648 (2019).
 21. Edgar, R. C. UPARSE: highly accurate OTU sequences from microbial amplicon reads. *Nat. Methods* **10**, 996–998 (2013).
 22. Mars, R. A. T. *et al.* Longitudinal Multi-omics Reveals Subset-Specific Mechanisms Underlying Irritable Bowel Syndrome. *Cell* (2020) doi:10.1016/j.cell.2020.08.007.
 23. Al-Ghalith, G. A., Hillmann, B., Ang, K., Shields-Cutler, R. & Knights, D. SHI7 Is a Self-Learning Pipeline for Multipurpose Short-Read DNA Quality Control. *mSystems* **3**, e00202–

- 17 (2018).
24. Al-Ghalith, G. & Knights, D. BURST enables mathematically optimal short-read alignment for big data. *bioRxiv* 2020.09.08.287128 (2020) doi:10.1101/2020.09.08.287128.
 25. Kalari, K. R. *et al.* MAP-RSeq: Mayo Analysis Pipeline for RNA sequencing. *BMC Bioinformatics* **15**, 224 (2014).
 26. Kim, D. *et al.* TopHat2: accurate alignment of transcriptomes in the presence of insertions, deletions and gene fusions. *Genome Biol.* **14**, R36 (2013).
 27. Liao, Y., Smyth, G. K. & Shi, W. The R package Rsubread is easier, faster, cheaper and better for alignment and quantification of RNA sequencing reads. *Nucleic Acids Res.* **47**, e47 (2019).
 28. Poyet, M. *et al.* A library of human gut bacterial isolates paired with longitudinal multiomics data enables mechanistic microbiome research. *Nat. Med.* (2019) doi:10.1038/s41591-019-0559-3.
 29. Richards, A. L. *et al.* Gut Microbiota Has a Widespread and Modifiable Effect on Host Gene Regulation. *mSystems* **4**, (2019).
 30. Luca, F., Kupfer, S. S., Knights, D., Khoruts, A. & Blekhman, R. Functional Genomics of Host-Microbiome Interactions in Humans. *Trends Genet.* **34**, 30–40 (2018).
 31. Qin, J. *et al.* A human gut microbial gene catalogue established by metagenomic sequencing. *Nature* **464**, 59–65 (2010).
 32. Lepage, P. *et al.* Twin study indicates loss of interaction between microbiota and mucosa of patients with ulcerative colitis. *Gastroenterology* **141**, 227–236 (2011).
 33. Kostic, A. D., Xavier, R. J. & Gevers, D. The microbiome in inflammatory bowel disease: current status and the future ahead. *Gastroenterology* **146**, 1489–1499 (2014).
 34. Ma, Z. S. Testing the Anna Karenina Principle in Human Microbiome-Associated Diseases. *iScience* **23**, 101007 (2020).

Decision Letter, first revision:

19th January 2022

Dear Ran,

Thank you for your patience while your manuscript "Shared and disease-specific host gene-microbiome associations across human diseases" was under peer-review at Nature Microbiology. It has now been seen by 3 referees, whose expertise and comments you will find at the of this email. You will see from their comments below that while they find your work of interest, some important points are raised. We are very interested in the possibility of publishing your study in Nature Microbiology, but would like to consider your response to these concerns in the form of a revised manuscript before we make a final decision on publication.

In particular, you will see that referees #2 and #3 both have ongoing concerns with the fact that the datasets have been processed using different computational pipelines. In line with referee #3's suggestion, please add a clear discussion of the limitations of this approach to the discussion section. Please also clarify the remaining concerns from the referees, which are clear and the remaining issues should be straightforward to address.

If you have not done so already please begin to revise your manuscript so that it conforms to our Article format instructions at <http://www.nature.com/nmicrobiol/info/final-submission/>

The usual length limit for a Nature Microbiology Article is six display items (figures or tables) and 3,000 words. We have some flexibility, and can allow a revised manuscript at 3,500 words, but please consider this a firm upper limit. There is a trade-off of ~250 words per display item, so if you need more space, you could move a Figure or Table to Supplementary Information.

Some reduction could be achieved by focusing any introductory material and moving it to the start of your opening 'bold' paragraph, whose function is to outline the background to your work, describe in a sentence your new observations, and explain your main conclusions. The discussion should also be limited. Methods should be described in a separate section following the discussion, we do not place a word limit on Methods.

Nature Microbiology titles should give a sense of the main new findings of a manuscript, and should not contain punctuation. Please keep in mind that we strongly discourage active verbs in titles, and that they should ideally fit within 90 characters each (including spaces).

We strongly support public availability of data. Please place the data used in your paper into a public data repository, if one exists, or alternatively, present the data as Source Data or Supplementary

Information. If data can only be shared on request, please explain why in your Data Availability Statement, and also in the correspondence with your editor. For some data types, deposition in a public repository is mandatory - more information on our data deposition policies and available repositories can be found at <https://www.nature.com/nature-research/editorial-policies/reporting-standards#availability-of-data>.

Please include a data availability statement as a separate section after Methods but before references, under the heading "Data Availability". This section should inform readers about the availability of the data used to support the conclusions of your study. This information includes accession codes to public repositories (data banks for protein, DNA or RNA sequences, microarray, proteomics data etc...), references to source data published alongside the paper, unique identifiers such as URLs to data repository entries, or data set DOIs, and any other statement about data availability. At a minimum, you should include the following statement: "The data that support the findings of this study are available from the corresponding author upon request", mentioning any restrictions on availability. If DOIs are provided, we also strongly encourage including these in the Reference list (authors, title, publisher (repository name), identifier, year). For more guidance on how to write this section please see:

<http://www.nature.com/authors/policies/data/data-availability-statements-data-citations.pdf>

To improve the accessibility of your paper to readers from other research areas, please pay particular attention to the wording of the paper's opening bold paragraph, which serves both as an introduction and as a brief, non-technical summary in about 150 words. If, however, you require one or two extra sentences to explain your work clearly, please include them even if the paragraph is over-length as a result. The opening paragraph should not contain references. Because scientists from other sub-disciplines will be interested in your results and their implications, it is important to explain essential but specialised terms concisely. We suggest you show your summary paragraph to colleagues in other fields to uncover any problematic concepts.

If your paper is accepted for publication, we will edit your display items electronically so they conform to our house style and will reproduce clearly in print. If necessary, we will re-size figures to fit single or double column width. If your figures contain several parts, the parts should form a neat rectangle when assembled. Choosing the right electronic format at this stage will speed up the processing of your paper and give the best possible results in print. We would like the figures to be supplied as vector files - EPS, PDF, AI or postscript (PS) file formats (not raster or bitmap files), preferably generated with vector-graphics software (Adobe Illustrator for example). Please try to ensure that all figures are non-flattened and fully editable. All images should be at least 300 dpi resolution (when figures are scaled to approximately the size that they are to be printed at) and in RGB colour format. Please do not submit Jpeg or flattened TIFF files. Please see also 'Guidelines for Electronic Submission of Figures' at the end of this letter for further detail.

Figure legends must provide a brief description of the figure and the symbols used, within 350 words, including definitions of any error bars employed in the figures.

When submitting the revised version of your manuscript, please pay close attention to our [href="https://www.nature.com/nature-research/editorial-policies/image-integrity">Digital Image Integrity Guidelines.](https://www.nature.com/nature-research/editorial-policies/image-integrity) and to the following points below:

Please include a statement before the acknowledgements naming the author to whom correspondence and requests for materials should be addressed.

Finally, we require authors to include a statement of their individual contributions to the paper -- such as experimental work, project planning, data analysis, etc. -- immediately after the acknowledgements. The statement should be short, and refer to authors by their initials. For details please see the Authorship section of our joint Editorial policies at http://www.nature.com/authors/editorial_policies/authorship.html

- * include a point-by-point response to any editorial suggestions and to our referees. Please include your response to the editorial suggestions in your cover letter, and please upload your response to the referees as a separate document.

- * ensure it complies with our format requirements for Letters as set out in our guide to authors at www.nature.com/nmicrobiol/info/gta/

- * state in a cover note the length of the text, methods and legends; the number of references; number and estimated final size of figures and tables

- * resubmit electronically if possible using the link below to access your home page:

[Redacted]

- *This url links to your confidential homepage and associated information about manuscripts you may have submitted or be reviewing for us. If you wish to forward this e-mail to co-authors, please delete this link to your homepage first.

Please ensure that all correspondence is marked with your Nature Microbiology reference number in the subject line.

Nature Microbiology is committed to improving transparency in authorship. As part of our efforts in this direction, we are now requesting that all authors identified as 'corresponding author' on published papers create and link their Open Researcher and Contributor Identifier (ORCID) with their account on the Manuscript Tracking System (MTS), prior to acceptance. This applies to primary research papers only. ORCID helps the scientific community achieve unambiguous attribution of all scholarly contributions. You can create and link your ORCID from the home page of the MTS by clicking on 'Modify my Springer Nature account'. For more information please visit www.springernature.com/orcid.

We hope to receive your revised paper within three weeks. If you cannot send it within this time, please let us know.

Yours sincerely,

[Redacted]

Reviewer Expertise:

Referee #1: gut microbiome, computational biology
Referee #2: gut microbiome, machine learning, GI disease
Referee #3: microbiome, computational biology

Reviewers Comments:

Reviewer #1 (Remarks to the Author):

The authors have done a thorough job of addressing reviewer comments and concerns.

I only have a minor point to bring up about the authors' response to my 8th comment on CLR transforms. I agree with the authors that it's not a great idea to CLR each taxonomic rank profile first and then combine across ranks. Now that I think about it, my main point was that the combining of different taxonomic ranks gives rise to a multiple-counting problem. That is, read abundances that contribute to the relative abundance of the genus *Bacteroides* will also contribute to the family-level relative abundance of *Bacteroidaceae*. I have a feeling that this kind of multi-counting could introduce artifacts, where certain taxonomic groups with a large number of clades (e.g. the *Bacteroides* genus has a much larger number of species in it than the *Prevotella* genus) swamp the abundance matrix with quasi-redundant count data (i.e. a bunch of correlated features that are non-independent due to taxonomic binning of counts). I doubt this will have a huge impact on the results, but I have a feeling that restricting your analysis to a particular taxonomic stratum (i.e. OTU/ASV, species, or genus) might be most appropriate. This kind of restriction in the number of features might actually help you see more hits, as it might reduce the multi-test correction penalty.

Anyway, this is just a minor point, which I think is partially addressed by the additional analyses the authors outline in their revision. I commend them on a highly responsive revision and a great paper.

Reviewer #2 (Remarks to the Author):

While I agree with the authors that uniform bioinformatic processing won't counteract the experimental biases introduced by different in sample collection, DNA extraction, etc, there is really not need to introduce additional biases with different bioinformatic processing pipelines. It's fine to analyse each dataset separately, rather than combining the data, but again this should be done after uniformly processing the reads. The two microbiome studies cited, did employ the same bioinformatic pipeline for data processing in each study, respectively. The study by Duvallet et al. used an in-house pipeline to re-process the reads, while Tierney et al. used profiles generated by the package `curatedMetagenomicData` which supplies microbiome profiles of datasets which were all re-processed using their pipeline. So, my concerns (previous comment 1) remain.

All other concerns were sufficiently addressed.

Reviewer #3 (Remarks to the Author):

The authors should be commended for their extensive revisions based on the previous reviewer comments. I remain highly enthusiastic about this work, as I think it serves as a novel framework for the integration of -omics data in the microbiome context.

Most of my comments have been completely addressed, but two remain:

1. The analysis comparing taxa calling methodology raised more questions in my mind than it did to quell them. A spearman rho of 0.6 at the genus does seem drastic when comparing compositional differences between the same data processed different ways. A more robust analysis would be to process the data as similarly as possible, even though there will be batch effects due to primer choice, facility, etc. Barring that (which would require a complete redo of the whole paper), I think the wording around the extent this could influence results should be strengthened in the discussion on page 18. It's not appropriate to say the different analysis methodologies play a minor role compared to other technical variables, given those variables can't be quantified here.
2. I still have concerns about un-equal sample sizes raised in my original Comment 4. The response from the authors is confusing to me, as they state there weren't overlaps across diseases in the study. In my original comment I was referring to the overlaps of shared pathways in the CCA (lines 212, figure 2A) and lasso results (figure 4A), where there do appear to be pathways and associations shared between diseases (and in such a way that lines up with sample size). I find the addition of the paragraph on lines 1292 – 1297 to be confusing, as it's not clear what this is referring to. This should be clarified for the reader.

Author Rebuttal, first revision:

Shared and disease-specific host gene-microbiome associations across human diseases

Editor comments:

Thank you for your patience while your manuscript "Shared and disease-specific host gene-microbiome associations across human diseases" was under peer-review at Nature Microbiology. It has now been seen by 3 referees, whose expertise and comments you will find at the of this email. You will see from their comments below that while they find your work of interest, some important points are raised. We are very interested in the possibility of publishing your study in Nature Microbiology, but would like to consider your response to these concerns in the form of a revised manuscript before we make a final decision on publication.

In particular, you will see that referees #2 and #3 both have ongoing concerns with the fact that the datasets have been processed using different computational pipelines. In line with referee #3's suggestion, please add a clear discussion of the limitations of this approach to the discussion section. Please also clarify the remaining concerns from the referees, which are clear and the remaining issues should be straightforward to address.

Response to editor:

We thank the editor and reviewers for these important comments and suggestions. We agree that differences in data processing pipelines could potentially contribute to differences across disease cohorts, and we have now updated the manuscript with a discussion on the limitations of this approach and potential impact on downstream results. Specifically, we acknowledge that it is hard to disentangle and compare the bias due to differences in data processing pipelines with those due to differences in experimental factors, and that these biases could potentially affect the integrative analysis of taxonomic composition and gene expression profiles across disease cohorts.

Additionally, to address the comment from Reviewer 1 regarding the issue of multi-counting reads when using combined taxa matrix, we have updated the manuscript to clarify how our approach overcomes this issue and allows us to identify signals found at any taxonomic level without missing potentially relevant associations. We have also addressed the comment from Reviewer 3 regarding unequal sample sizes across cohorts by updating discussion text to clarify the potential impact of sample size on identification of associations and their overlaps across diseases.

Reviewer #1 (Remarks to the Author):

The authors have done a thorough job of addressing reviewer comments and concerns.

Reviewer 1, Comment 1

I only have a minor point to bring up about the authors' response to my 8th comment on CLR transforms. I agree with the authors that it's not a great idea to CLR each taxonomic rank profile first and then combine across ranks. Now that I think about it, my main point was that the combining of different taxonomic ranks gives rise to a multiple-counting problem. That is, read abundances that contribute to the relative abundance of the genus *Bacteroides* will also contribute to the family-level relative abundance of *Bacteroidaceae*. I have a feeling that this kind of multi-counting could introduce artifacts, where certain taxonomic groups with a large number of clades (e.g. the *Bacteroides* genus has a much larger number of species in it than the *Prevotella* genus) swamp the abundance matrix with quasi-redundant count data (i.e. a bunch of correlated features that are non-independent due to taxonomic binning of counts). I doubt this will have a huge impact on the results, but I have a feeling that restricting your analysis to a particular taxonomic stratum (i.e. OTU/ASV, species, or genus) might be most appropriate. This kind of restriction in the number of features might actually help you see more hits, as it might reduce the multi-test correction penalty.

Anyway, this is just a minor point, which I think is partially addressed by the additional analyses the authors outline in their revision. I commend them on a highly responsive revision and a great paper.

Response:

We thank the reviewer for their positive feedback and constructive suggestions. We agree that combining taxonomic ranks introduces the issue of multi-counting reads within a taxonomic group, leading to addition of correlated features in the taxa abundance matrix. We note that this issue is mitigated by our penalization approach using lasso with stability selection, which, instead of picking multiple correlated taxa from a given taxonomic clade, only selects the microbial taxon from a group of correlated taxa for which the abundance is most robustly associated with the expression of a host gene^{1,2}. This approach allows us to identify signals found at any taxonomic level and avoid missing potentially relevant associations by limiting the analysis to a single taxonomic level. At the same time, given the large number of features in high-dimensional datasets like gene expression and microbiome data, this approach circumvents the computationally intensive analysis that would be required if each taxonomic level was analyzed separately. Our assumption is that if we repeated our integration analyses separately at different taxonomic levels, we would likely identify the same set of taxa as our current approach that uses a combined taxonomic profile. We have added a new paragraph that clarifies this in the manuscript (**Methods, page 24**):

“To allow for identification of associations at any taxonomic level without repeating the analyses at each taxonomic rank, we combined summarized taxa matrices at different ranks into a combined taxa matrix. This approach could potentially lead to multi-counting of reads within a taxonomic group, leading to addition of correlated features in the taxa abundance matrix. This issue is mitigated by our penalization approach using lasso with stability selection (see Method sections on “Lasso regression analysis” and “Stability selection for lasso model”). Specifically, instead of picking multiple correlated microbial taxa from a given taxonomic clade, this approach only selects the microbial taxon out of a group of correlated taxa for which the abundance is most robustly associated with the expression of a host gene^{31,172}. This approach allows us to identify signals found at any taxonomic level and avoid missing potentially relevant associations by limiting the analysis to a single taxonomic level. At the same time, given the large number of features in high-dimensional datasets like gene expression and microbiome data, our approach circumvents the computationally intensive analysis that would be required if each taxonomic level was analyzed separately.”

Reviewer #2 (Remarks to the Author):

Reviewer 2, Comment 1

While I agree with the authors that uniform bioinformatic processing won't counteract the experimental biases introduced by different in sample collection, DNA extraction, etc, there is really not need to introduce additional biases with different bioinformatic processing pipelines. It's fine to analyse each dataset separately, rather than combining the data, but again this should be done after uniformly processing the reads. The two microbiome studies cited, did employ the same bioinformatic pipeline for data processing in each study, respectively. The study by Duvallet et al. used an in-house pipeline to re-process the reads, while Tierney et al. used profiles generated by the package curatedMetagenomicData which supplies microbiome profiles of datasets which were all re-processed using their pipeline. So, my concerns (previous comment 1) remain.

All other concerns were sufficiently addressed.

Response:

We agree with the reviewer that differences in data processing pipelines could potentially add to the biases introduced by differences in experimental protocols across the disease cohorts. While a re-analysis of a few representative samples in our dataset using different processing pipelines showed correlated taxonomic profiles (Methods, page 26), it is hard to quantify the overall influence of these differences on downstream analyses and disentangle these biases from batch effects due to experimental factors. We have now updated our manuscript with discussion on limitations of our approach (**Discussion, page 18**; new and edited text in green):

“Another caveat of our study is that it includes three different disease cohorts with disparate protocols for sample collection, preparation, sequencing, and data processing, which may lead to batch effects^{150–152}. While a re-analysis of a few representative samples in our dataset using different data processing pipelines showed fairly correlated taxonomic profiles (see Methods), it is hard to assess the overall bias due to differences in data processing on the downstream analyses. Additionally, it is difficult to disentangle the biases introduced due to differences in data analysis factors from those introduced due to differences in experimental factors, as the latter variables cannot be quantified in our study. These combined differences could potentially influence the assignment of taxonomic composition and gene expression profiles across disease cohorts, which, in turn, may impact downstream integration analyses. While it is difficult to fully eliminate these biases, we tried to minimize the overall batch effect across disease cohorts in our study by adopting a meta-analysis approach, where we performed our integration analysis and compared disease to control samples within each cohort separately, and combined the results across cohorts at the last analysis step. While meta-analysis approaches have disadvantages, like reduced statistical power, they have been extensively used to minimize batch effects when integrating genomic data from multiple studies¹⁵², and have recently proven useful in microbiome studies^{153,154}”.

Reviewer #3 (Remarks to the Author):

The authors should be commended for their extensive revisions based on the previous reviewer comments. I remain highly enthusiastic about this work, as I think it serves as a novel framework for the integration of -omics data in the microbiome context.

Most of my comments have been completely addressed, but two remain:

Reviewer 3, Comment 1

1. The analysis comparing taxa calling methodology raised more questions in my mind than it did to quell them. A spearman rho of 0.6 at the genus does seem drastic when comparing compositional differences between the same data processed different ways. A more robust analysis would be to process the data as similarly as possible, even though there will be batch effects due to primer choice, facility, etc. Barring that (which would require a complete redo of the whole paper), I think the wording around the extent this could influence results should be strengthened in the discussion on page 18. It's not appropriate to say the different analysis methodologies play a minor role compared to other technical variables, given those variables can't be quantified here.

Response:

We thank the reviewer for their positive feedback and constructive comments. We agree that differences in data processing pipelines could potentially introduce additional biases contributing to batch effects due to differences in experimental protocols. We also agree that it is hard to

disentangle and compare the bias due to differences in data processing pipelines with those due to differences in experimental factors, as the latter cannot be quantified in our study. Thus, these combined differences could potentially influence the assignment of taxonomic composition and gene expression profiles across disease cohorts, which in turn may impact downstream integration analyses. We have now updated our discussion text to clarify these limitations of our approach and how it impacts our results – please refer to our response to **Reviewer 2's comment** above to see the updated text (**Discussion, page 18**). We have also edited the wording in the section in Methods describing correlations between different processing pipelines to reflect this potential issue (**Methods, page 26**).

Reviewer 3, Comment 2

2. I still have concerns about un-equal sample sizes raised in my original Comment 4. The response from the authors is confusing to me, as they state there weren't overlaps across diseases in the study. In my original comment I was referring to the overlaps of shared pathways in the CCA (lines 212, figure 2A) and lasso results (figure 4A), where there do appear to be pathways and associations shared between diseases (and in such a way that lines up with sample size). I find the addition of the paragraph on lines 1292 – 1297 to be confusing, as it's not clear what this is referring to. This should be clarified for the reader.

Response:

We apologize for the confusion, which stemmed from our misunderstanding of the reviewer's original comment. Indeed, the differences in sample size across disease cohorts could potentially influence the overlapping associations and pathways we identified across diseases using the sparse CCA and lasso approaches. We have now removed the text in lines 1292 – 1297 from the manuscript, which was added as a result of our misunderstanding of the reviewer's original comment. The discussion section already included a discussion on unequal sample sizes, which we edited further to address this comment and clarify this as a limitation of our study (**Discussion, page 18**):

“Another potential issue is the difference in sample size across disease cohorts and between the case and control groups within each cohort. These sample size differences can result in differences in statistical power when applying sparse CCA and lasso regression. This can impact the number of host gene-microbe associations and pathways identified in each disease cohort, and the number of overlapping associations and pathways identified across cohorts. We attempted to minimize this effect by applying a differential enrichment analysis that is more robust to different levels of statistical power due to sample size (see Methods)”.

References:

1. Tibshirani, R. Regression Shrinkage and Selection via the Lasso. *J. R. Stat. Soc. Series B*

Stat. Methodol. **58**, 267–288 (1996).

2. Meinshausen, N. & Bühlmann, P. Stability selection. *J. R. Stat. Soc. Series B Stat. Methodol.* **72**, 417–473 (2010).

Decision Letter, second revision:

Our ref: NMICROBIOL-21030565B

16th February 2022

Dear Ran,

Thank you for submitting your revised manuscript "Shared and disease-specific host gene-microbiome associations across human diseases" (NMICROBIOL-21030565B). It has now been seen by the original referees and their comments are below. The reviewers find that the paper has improved in revision, and therefore we'll be happy in principle to publish it in Nature Microbiology, pending minor revisions to satisfy the referees' final requests and to comply with our editorial and formatting guidelines.

Thank you again for your interest in Nature Microbiology Please do not hesitate to contact me if you have any questions.

Sincerely,

[Redacted]

Decision Letter, final checks:

Our ref: NMICROBIOL-21030565B

8th March 2022

Dear Ran,

Thank you for your patience as we've prepared the guidelines for final submission of your Nature Microbiology manuscript, "Shared and disease-specific host gene-microbiome associations across human diseases" (NMICROBIOL-21030565B). Please carefully follow the step-by-step instructions provided in the attached file, and add a response in each row of the table to indicate the changes that you have made. Please also check and comment on any additional marked-up edits we have proposed within the text. Ensuring that each point is addressed will help to ensure that your revised manuscript can be swiftly handed over to our production team.

In recognition of the time and expertise our reviewers provide to Nature Microbiology's editorial process, we would like to formally acknowledge their contribution to the external peer review of your manuscript entitled "Shared and disease-specific host gene-microbiome associations across human diseases". For those reviewers who give their assent, we will be publishing their names alongside the published article.

Nature Microbiology offers a Transparent Peer Review option for new original research manuscripts submitted after December 1st, 2019. As part of this initiative, we encourage our authors to support increased transparency into the peer review process by agreeing to have the reviewer comments, author rebuttal letters, and editorial decision letters published as a Supplementary item. When you submit your final files please clearly state in your cover letter whether or not you would like to participate in this initiative. Please note that failure to state your preference will result in delays in accepting your manuscript for publication.

Cover suggestions

As you prepare your final files we encourage you to consider whether you have any images or illustrations that may be appropriate for use on the cover of Nature Microbiology.

Nature Microbiology has now transitioned to a unified Rights Collection system which will allow our Author Services team to quickly and easily collect the rights and permissions required to publish your work. Approximately 10 days after your paper is formally accepted, you will receive an email in providing you with a link to complete the grant of rights. If your paper is eligible for Open Access, our

Author Services team will also be in touch regarding any additional information that may be required to arrange payment for your article.

Please note that *Nature Microbiology* is a Transformative Journal (TJ). Authors may publish their research with us through the traditional subscription access route or make their paper immediately open access through payment of an article-processing charge (APC). Authors will not be required to make a final decision about access to their article until it has been accepted. [Find out more about Transformative Journals](https://www.springernature.com/gp/open-research/transformative-journals)

[Redacted]

With best wishes,

[Redacted]

Final Decision Letter:

6th April 2022

Dear Ran,

I am pleased to accept your Article "Identification of shared and disease-specific host gene-microbiome associations across human diseases using multi-omic integration" for publication in Nature Microbiology. Thank you for having chosen to submit your work to us and many congratulations.

Acceptance of your manuscript is conditional on all authors' agreement with our publication policies (see <https://www.nature.com/nmicrobiol/editorial-policies>). In particular your manuscript must not be published elsewhere and there must be no announcement of the work to any media outlet until the publication date (the day on which it is uploaded onto our website).

Please note that *Nature Microbiology* is a Transformative Journal (TJ). Authors may publish their research with us through the traditional subscription access route or make their paper immediately open access through payment of an article-processing charge (APC). Authors will not be required to make a final decision about access to their article until it has been accepted. [Find out more about Transformative Journals](https://www.springernature.com/gp/open-research/transformative-journals)

Authors may need to take specific actions to achieve [compliance with funder and institutional open access mandates](https://www.springernature.com/gp/open-research/funding/policy-compliance-faqs). If your research is supported by a funder that requires immediate open access (e.g. according to [Plan S principles](https://www.springernature.com/gp/open-research/plan-s-compliance)) then you should select the gold OA route, and we will direct you to the compliant route where possible. For authors selecting the subscription publication route, the journal's standard licensing terms will need to be accepted, including [self-archiving policies](https://www.springernature.com/gp/open-research/policies/journal-policies). Those licensing terms will supersede any other terms that the author or any third party may assert apply to any version of the manuscript.

An online order form for reprints of your paper is available at a

href="https://www.nature.com/reprints/author-reprints.html">https://www.nature.com/reprints/author-reprints.html. All co-authors, authors' institutions and authors' funding agencies can order reprints using the form appropriate to their geographical region.

With kind regards,

[Redacted]

P.S. Click on the following link if you would like to recommend Nature Microbiology to your librarian <http://www.nature.com/subscriptions/recommend.html#forms>

** Visit the Springer Nature Editorial and Publishing website at http://editorial-jobs.springernature.com?utm_source=ejp_NMicro_email&utm_medium=ejp_NMicro_email&utm_campaign=ejp_NMicro for more information about our career opportunities. If you have any questions please click [here](mailto:editorial.publishing.jobs@springernature.com). **